# CREB1-driven CXCR4[hi] neutrophils promote skin inflammation in mouse models and human patients

Jiaoling Chen [1,4], Yaxing Bai [1,4], Ke Xue [1], Zhiguo Li [1], Zhenlai Zhu [1], Qingyang Li [1], Chen Yu [1], Bing Li[1], Shengxian Shen[1], Pei Qiao [1], Caixia Li[1], Yixin Luo[1], Hongjiang Qiao [1], Erle Dang [1], Wen Yin [2], Johann E. Gudjonsson [3] ✉, Gang Wang [1] ✉ & Shuai Shao[1] ✉

Neutrophils have a pathogenic function in inflammation via releasing pro-inflammatory mediators or neutrophil extracellular traps (NETs). However, their heterogeneity and pro-inflammatory mechanisms remain unclear. Here, we demonstrate that CXCR4[hi] neutrophils accumulate in the blood and inflamed skin in human psoriasis, and correlate with disease severity. Compared to CXCR4[lo] neutrophils, CXCR4[hi] neutrophils have enhanced NETs formation, phagocytic function, neutrophil degranulation, and overexpression of pro-inflammatory cytokines and chemokines in vitro. This is accompanied by a metabolic shift in CXCR4[hi] neutrophils toward glycolysis and lactate release, thereby promoting vascular permeability and remodeling. CXCR4 expression in neutrophils is dependent on CREB1, a transcription factor activated by TNF and CXCL12, and regulated by de novo synthesis. In vivo, CXCR4[hi] neutrophil infiltration amplifies skin inflammation, whereas blockade of CXCR4[hi] neutrophils through CXCR4 or CXCL12 inhibition leads to suppression of immune responses. In this work, our study identifies CREB1 as a critical regulator of CXCR4[hi] neutrophil development and characterizes the contribution of CXCR4[hi] neutrophils to vascular remodeling and inflammatory responses in skin.

Neutrophils are the most abundant immune cells in the human body, participating in innate immune responses[1]. Although short-lived, neutrophils realize their pathogenic role via phagocytosis, degranulation, and release of neutrophil extracellular traps (NETs)[2], modulating inflammatory responses in various immune-mediated diseases such as psoriasis[3], systemic lupus erythematosus[4], and rheumatoid arthritis[5]. In psoriasis, neutrophils interact with vascular endothelial cells to amplify inflammatory trafficking[6,7]. Furthermore, the release of NETs may activate keratinocytes[8], dendritic cells[3], and T helper 17 (Th17) cells[9], initiating the adaptive immune system and amplifying

inflammatory responses in diseases such as psoriasis. However, the steps involved in this neutrophil activation, and the identity of the neutrophil subset involved are still poorly understood. Hence, a better understanding of the interplay between neutrophils and stromal cells, and the upstream regulators involved may provide important insights into disease biology and opportunities for therapeutic development.

Emerging evidence has shown that neutrophils exhibit considerable plasticity with heterogeneous phenotypes and subpopulations under certain physiologic and pathologic conditions[10,11]. These

[1]Department of Dermatology, Xijing Hospital, Fourth Military Medical University, Xi'an, Shaanxi 710032, China. [2]Department of Transfusion Medicine, Xijing Hospital, Fourth Military Medical University, Xi'an, Shaanxi 710032, China. [3]Department of Dermatology, University of Michigan, Ann Arbor, MI 48109, USA. [4]These authors contributed equally: Jiaoling Chen, Yaxing Bai. ✉e-mail: johanng@med.umich.edu; wanggangxjyy@163.com; shaoshuai19900728@qq.com

neutrophil subtypes likely reflect differences in density, surface markers, and maturity, but consensus criteria are still lacking. Neutrophil subsets have been defined by the expression of molecules including CD177[12], ofactomedin-4 (OLFM4)[13], and CD63[14]. Although neutrophils are conspicuously poor in RNA content, recent single-cell sequencing approaches have advanced our understanding of neutrophil plasticity, identifying functional subtypes such as Fth1[hi] neutrophils, TGFβ1+CCR5+ neutrophils, and various tumor-associated neutrophil (TAN) populations[15–18], providing a comprehensive transcriptional landscape of neutrophils during their lifetime[15]. Moreover, neutrophils may adopt variable phenotypic and functional properties in different tissues[19]. In the lung, C-X-C chemokine ligand 12 (CXCL12)-producing vessels play a critical role in the retention of neutrophils in specific perivascular areas, where they undergo reprogramming to support vascular growth[19]. One recent study reveals that immunotherapy for cancers expands a neutrophil state with an interferon-stimulated gene signature[20], highlighting the dynamic nature of neutrophil phenotypes. Short-lived neutrophils progressively upregulate C-X-C chemokine receptor 4 (CXCR4) during their lifetime, along with the loss of L-selectin (CD62L), and represent an overly active group with enhanced NETs formation[21]. Exposure of neutrophils to a low dose of lipopolysaccharide promotes upregulation of CXCR4 in the lung, which then facilitates the release of NETs to mediate allergic asthma[22]. However, their role in inflammatory responses in skin has remained unclear.

In this work, we investigate the function and contribution of CXCR4[hi] neutrophils to skin inflammation and demonstrate the role of the transcription factor cAMP response element binding protein 1 (CREB1) in the induction and pro-inflammatory function of CXCR4[hi] neutrophils. Our findings provide insights into the role of neutrophils in amplifying skin inflammation and identify CXCR4[hi] neutrophils as a potential therapeutic target in inflammatory skin diseases such as psoriasis.

## Results

### CXCR4[hi] neutrophils are increased in peripheral blood and inflamed psoriatic skin

Firstly, we observed increased surface expression of CXCR4 on peripheral psoriatic neutrophils compared with healthy controls as measured by mean fluorescence intensity (MFI) of CXCR4 (Fig. 1a, Supplementary Fig. 1a), which was positively correlated with Psoriasis Area Severity Index (PASI) (R = 0.37, p = 0.0012, n = 25) (Fig. 1b). Furthermore, the frequency of CXCR4[hi] neutrophil was also elevated in psoriatic patients (Fig. 1c, Supplementary Fig. 1b) and correlated positively with PASI score (Fig. 1d), but not with disease duration (Supplementary Fig. 1c). Quantitative real-time PCR (qRT-PCR) (Supplementary Fig. 1d) and Western blot (Fig. 1e) confirmed the increased expression of CXCR4 in circulating psoriatic neutrophils compared to healthy controls. Moreover, CXCR4[hi] neutrophils were increased in inflamed psoriatic skin with immunofluorescence (IF) showing a marked increase and co-localization of CXCR4 with the neutrophil marker CD15, but both were undetectable in healthy control skin (Fig. 1f). The proportion of CXCR4[hi] neutrophils in inflamed psoriatic skin was 5-fold greater than that of CXCR4[lo] neutrophils (Fig. 1g). Notably, in psoriasis patients treated with an anti-interleukin (IL)-17A biologic (secukinumab), CXCR4 expression on circulating neutrophils was reduced over a 12-weeks period treatment (Fig. 1h, Supplementary Fig. 1e) and was concomitant with a decrease in overall disease activity (Supplementary Fig. 1f). In addition, ELISA assay detected that serum protein levels of CXCL12, IL-17A, and myeloperoxidase, also showed decrease with treatment (Fig. 1i).

### CXCR4[hi] neutrophils exhibit enhanced pro-inflammatory functions

Morphologically, CXCR4[hi] neutrophils, particularly psoriatic CXCR4[hi] neutrophils, displayed a higher nucleus/cytoplasm ratio and a hypersegmented nucleus compared to CXCR4[lo] neutrophils (Fig. 2a). Compared with CXCR4[lo] neutrophils and controls, psoriatic CXCR4[hi] neutrophils had a higher surface protein expression of activation indicators including CD66b, CD11b, and CD44 (a leukocyte adhesion molecule) and maturation indicators[23] such as CD101 and CD10, but lower expression of CXCR2, a negative indicator of neutrophil activation[24] (Fig. 2b, Supplementary Fig. 2a). CXCR4[lo] neutrophils in healthy controls expressed relatively lower, but not absent level of CD10 (Supplementary Fig. 2a). Then, neutrophil survival rate after 24 h (h) of culture in vitro was evaluated via Annexin V-7-AAD staining and showed an increased early apoptotic rate of psoriatic CXCR4[hi] neutrophils compared to the other two groups (Fig. 2c, Supplementary Fig. 2b). We also observed elevated ROS levels in CXCR4[hi] neutrophils compared to CXCR4[lo] neutrophils, as indicated by the dihydroethidium (DHE) probe (Fig. 2d, Supplementary Fig. 2c). Generation of ROS is a key step to form NETs, and consistent with that we observed increased NETs formation by IF staining and quantification, which was most pronounced in psoriatic CXCR4[hi] neutrophils (Fig. 2e), and further enhanced in PMA-treated groups (Supplementary Fig. 3a, b). In addition, phagocytic capacity, as measured by the ability to take up FITC-labeled *Escherichia coli* (Fig. 2f, g, Supplementary Fig. 3c, d), and degranulation capacity, as measured by CD63 expression (Fig. 2h), were all higher in CXCR4[hi] compared to CXCR4[lo] neutrophils. Correspondingly, ELISA and Western blot of the neutrophil supernatant demonstrated that CXCR4[hi] neutrophils released greater amount of matrix metalloprotein 9 (MMP-9) than CXCR4[lo] neutrophils (Fig. 2i, j). QRT-PCR further confirmed that pro-inflammatory factors, including *IL17A*, *tumor necrosis factor (TNF)*, *CXCL8 (IL8)*, *IL18*, *high mobility group box protein 1 (HMGB1)*, *cathelicidin (LL37)*, *S100A8*, and *S100A9*, were all highly expressed in CXCR4[hi] neutrophils, compared to CXCR4[lo] neutrophils, with expression levels being overall higher in psoriasis CXCR4[hi] neutrophils than healthy controls (Fig. 2k). Collectively, these data support a pro-inflammatory role of CXCR4[hi] neutrophils in skin inflammation.

### CXCR4[hi] neutrophils show a metabolic shift towards glycolysis along with heightened pro-inflammatory functions

To further investigate the function of CXCR4[hi] neutrophils, RNA sequencing (RNA-seq) was performed on paired peripheral CXCR4[lo] and CXCR4[hi] neutrophils isolated from psoriasis patients (n = 6) and healthy controls (n = 7) (Supplementary Fig. 4a). A total of 1699 and 319 genes were upregulated (FC > 1, FDR < 0.05) in CXCR4[hi] *vs.* CXCR4[lo] neutrophils in psoriasis patients and healthy controls, respectively (Fig. 3a). Psoriatic CXCR4[hi] neutrophils had increased expression of genes involved in differentiation (i.e., *CREB1*), activation (i.e., *LCN2*, *TNF*), metabolism (i.e., *lactate dehydrogenase A (LDHA)* and *CD36*), adherence (i.e., *ICAM2*), and chemotaxis (i.e., *CCR7* and *CXCL1*), compared to CXCR4[lo] neutrophils (Fig. 3b). Enriched biological processes in psoriatic CXCR4[hi] neutrophils included metabolic processes and cellular/immune responses, with glycolysis being the most significantly enriched pathway (Fig. 3c). This was further confirmed by gene-set enrichment analysis (Fig. 3d, Supplementary Fig. 4b). QRT-PCR confirmed increased mRNA expression of several key glycolytic genes, including *hypoxia inducible factor-1 (HIF1A)*, *glucose transporter glucose transporter 1 (GLUT1)*, *hexokinase2 (HK2)*, and *LDHA*, in CXCR4[hi] neutrophils compared to CXCR4[lo] neutrophils (Supplementary Fig. 4c). Cell IF demonstrated that LDHA, a critical molecule in the glycolytic pathway, observably co-localized with CXCR4 in psoriatic neutrophils (Supplementary Fig. 4d). Flow cytometry analysis confirmed increased expression of glycolytic proteins in CXCR4[hi] neutrophils including HIF1A, HK2, and GLUT1 (Fig. 3e, Supplementary Fig. 4e). Correspondingly, glucose uptake and lactate production were markedly increased in CXCR4[hi] neutrophils (Fig. 3f, g, Supplementary Fig. 4f). These data reveal

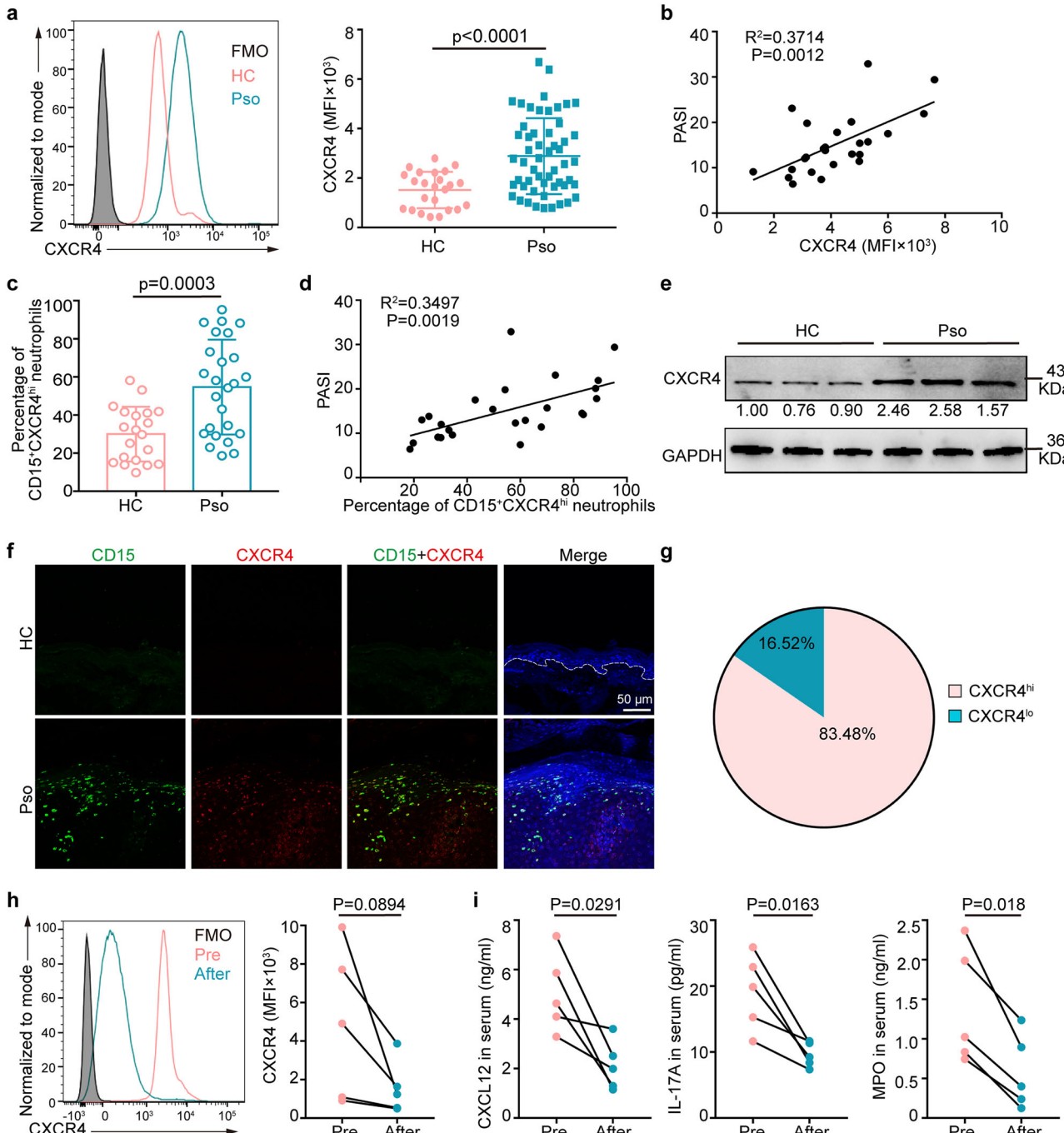

**Fig. 1 | CD15+CXCR4hi neutrophils are increased in the peripheral blood and inflamed psoriatic lesions. a** The mean fluorescence intensity (MFI) of CXCR4 in peripheral neutrophils from healthy controls (n = 24) and psoriasis patients (n = 54). **b** Correlation of the CXCR4 MFI on peripheral neutrophils with PASI in psoriasis patients (n = 25). **c** The proportion of circulating CD15+CXCR4hi neutrophils in healthy controls (n = 20) and psoriasis patients (n = 25). **d** Correlation of the proportions of CXCR4hi neutrophils with PASI in psoriasis patients. The adjusted $R^2$ and $P$-values were plotted in the graph. **e** Representative immunoblots of total CXCR4 in circulating neutrophils from healthy controls and psoriasis patients. The relative multiple expression was counted. Blots for each antigen were processed in the same experiment in parallel. The result was repeated twice independently with similar results. **f** Immunofluorescence staining of CD15 (green) and CXCR4 (red) in normal and inflamed psoriatic skin. Scale bar = 50 μm. n = 10 biologically independent samples. **g** The proportion of CXCR4lo *vs.* CXCR4hi neutrophils in inflamed psoriatic skin (n = 30 fields from 10 patient samples). MFI of CXCR4 on peripheral neutrophils (**h**) and serum protein levels of CXCL12, IL-17A, and MPO (**i**) before and after treatment with anti-IL-17 inhibitor for 12 weeks. n = 5 biologically independent samples. Data are mean ± SD. Analyses: unpaired Student's t-test in (**a**) and (**c**); The Spearman method in (**b**) and (**d**); Paired Student's t-test in (**h**) and (**i**). The paired and unpaired Student's t-test were conducted as two-sided tests. FMO, Fluorescence Minus One; HC, healthy control; MFI, mean fluorescence intensity; Pre, pre-treatment; Pso, psoriasis patients. Source data are provided as a Source Data file.

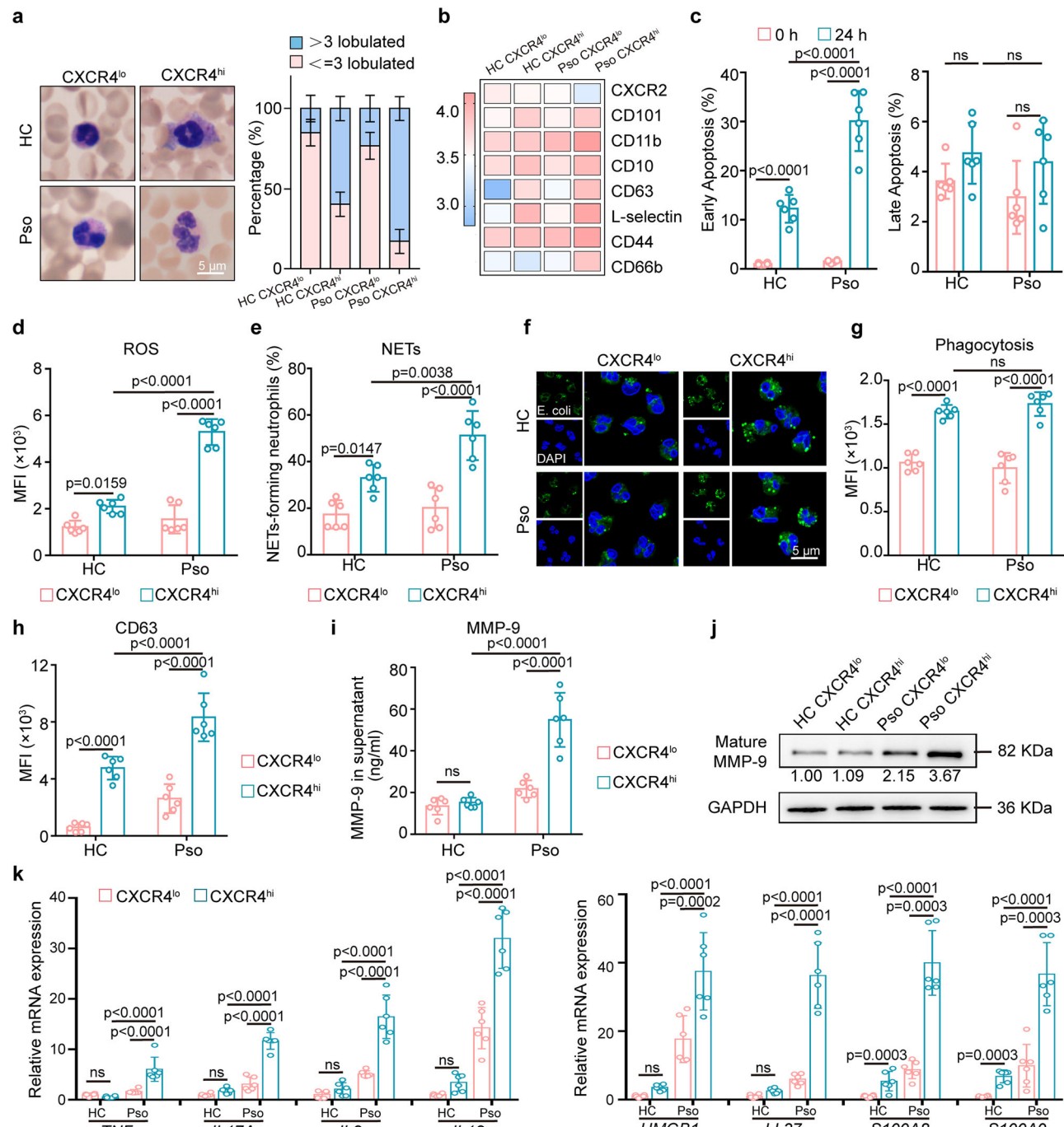

**Fig. 2 | CXCR4^hi neutrophils display enhanced pro-inflammatory functions.**
**a** Representative images and quantification of different nucleus morphology of CXCR4^lo and CXCR4^hi neutrophils from healthy controls and psoriasis patients. Scale bar = 5 μm. **b** Flow cytometry analysis of key immune markers between peripheral CXCR4^lo and CXCR4^hi neutrophils from healthy controls and psoriasis patients (scale: log-transformed MFI). **c** Assessment of CXCR4^hi neutrophil survival by Annexin V and 7-AAD staining after 24 h of culture. **d** Measurement of ROS production by DHE fluorescence. **e** Quantification of NETs in CXCR4^lo and CXCR4^hi neutrophils from healthy controls and psoriasis patients. Phagocytosis of pHrodo Green *E. coli* by neutrophils was evaluated by immunofluorescence staining (**f**) and flow cytometry (**g**). Scale bar = 5 μm. **h** Degranulation of CXCR4^lo or CXCR4^hi neutrophils as assessed by CD63 expression. ELISA quantification (**i**) and representative immunoblots (**j**) for mature MMP-9 in the supernatant of CXCR4^lo and CXCR4^hi neutrophils from healthy controls and psoriasis patients that cultured for 24 h. GAPDH was analyzed from corresponding neutrophils. Blots for each antigen were processed in the same experiment in parallel. **k** Relative mRNA expression of pro-inflammatory mediators in CXCR4^lo and CXCR4^hi neutrophils from healthy controls and psoriasis patients. Data are mean ± SD (n = 6 biologically independent samples). The immunofluorescence staining was repeated three times independently with similar results. Two-way ANOVA with Tukey's post hoc test was performed as two-sided analyses and adjusted for multiple comparisons in the statistical analyses. ns, not significant. HC, healthy control; MFI, mean fluorescence intensity; MMP-9, matrix metalloprotein 9; NETs; neutrophil extracellular traps; neu, neutrophils; Pso, psoriasis patients. Source data are provided as a Source Data file.

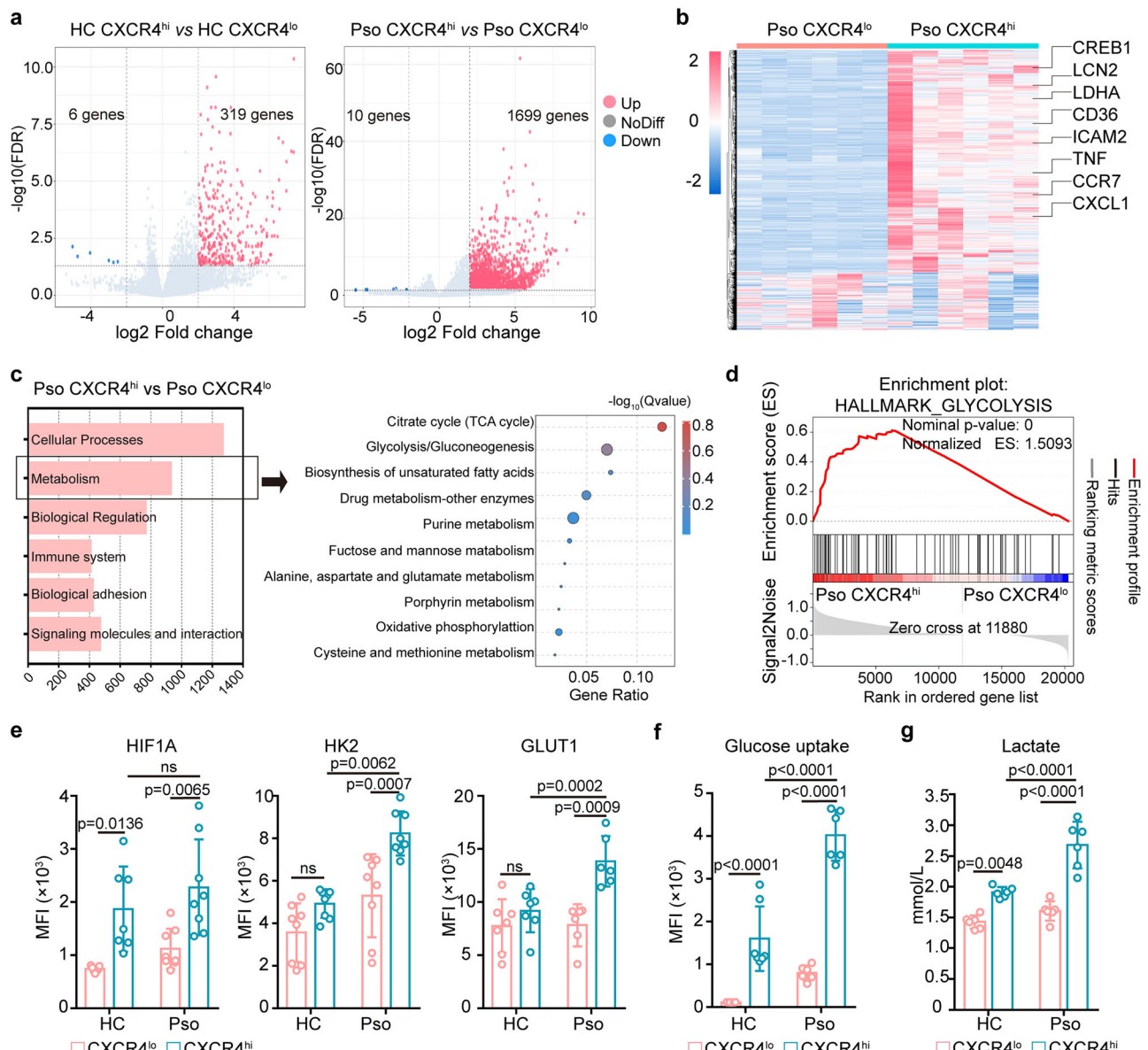

**Fig. 3 | RNA sequencing reveals the immune signature of CXCR4[hi] neutrophils.** **a** Volcano plot showing the number of differentially expressed genes (DEGs) between CXCR4[lo] and CXCR4[hi] neutrophils from healthy controls (n = 7) and psoriasis patients (n = 6). **b** Heatmap of DEGs between psoriatic CXCR4[lo] and CXCR4[hi] neutrophils related to neutrophil and immune function. **c** Enriched GO terms between CXCR4[lo] and CXCR4[hi] psoriatic neutrophils. **d** Gene set enrichment analysis showing enrichment of genes involved in glycolysis in psoriatic CXCR4[hi] neutrophils. **e** Flow cytometry of glycolytic markers in CXCR4[lo] and CXCR4[hi] neutrophils from healthy controls and psoriasis patients. **f** Uptake of glucose (2-NBDG)

in CXCR4[lo] and CXCR4[hi] neutrophils from healthy controls and psoriasis patients. **g** Extracellular lactate production in CXCR4[lo] and CXCR4[hi] neutrophils from healthy controls and psoriasis patients. Mean ± SD (n = 6 biologically independent samples/group). Two-way ANOVA with Tukey's post hoc test was performed as two-sided analyses and adjusted for multiple comparisons in the statistical analyses. ns, not significant. HC, healthy control; MFI, mean fluorescence intensity; Pso, psoriasis patients; RNA-seq, RNA sequencing. Source data are provided as a Source Data file.

upregulation of glycolytic activity in both healthy and psoriatic-derived CXCR4[hi] neutrophils.

## CXCR4[hi] neutrophils modulate vascular permeability via the lactate-GPR81 axis

Endothelial cell (EC) barrier damage contributes to inflammatory trafficking[6]. We, therefore, explored if CXCR4[hi] neutrophils affect vascular remodeling. Psoriatic CXCR4[hi] neutrophils showed prominent adhesions to human microvascular endothelial cells (HMEC-1) in a co-culture system (Fig. 4a, Supplementary Fig. 5a), suggesting that CXCR4[hi] neutrophils have enhanced adhesion ability. Psoriatic CXCR4[hi] neutrophils induced the expression of the adhesion molecules *ICAM-1* and

*VCAM-1* in HMEC-1 cells (Fig. 4b). Furthermore, psoriatic CXCR4[hi] neutrophils affected vascular permeability, with Western blot (Fig. 4c) and qRT-PCR (Fig. 4d) showing reduced expression of junction proteins in HMEC-1 cells following exposure to psoriatic CXCR4[hi] neutrophils, including zonula occudens-1 (ZO-1), VE-Cadherin, and Occludin. These findings were validated by cell IF (Supplementary Fig. 5b). Lastly, by using a Transwell culture system and measuring FITC-dextran leakage between the two chambers, we confirmed the increased permeability of HMEC-1 monolayer caused by psoriatic CXCR4[hi] neutrophils, compared to CXCR4[lo] and CXCR4[hi] controls (Fig. 4e).

Lactate, a major by-product of glycolytic cells, is an important regulator of EC activation[25]. Inhibition of LDHA partially reversed the

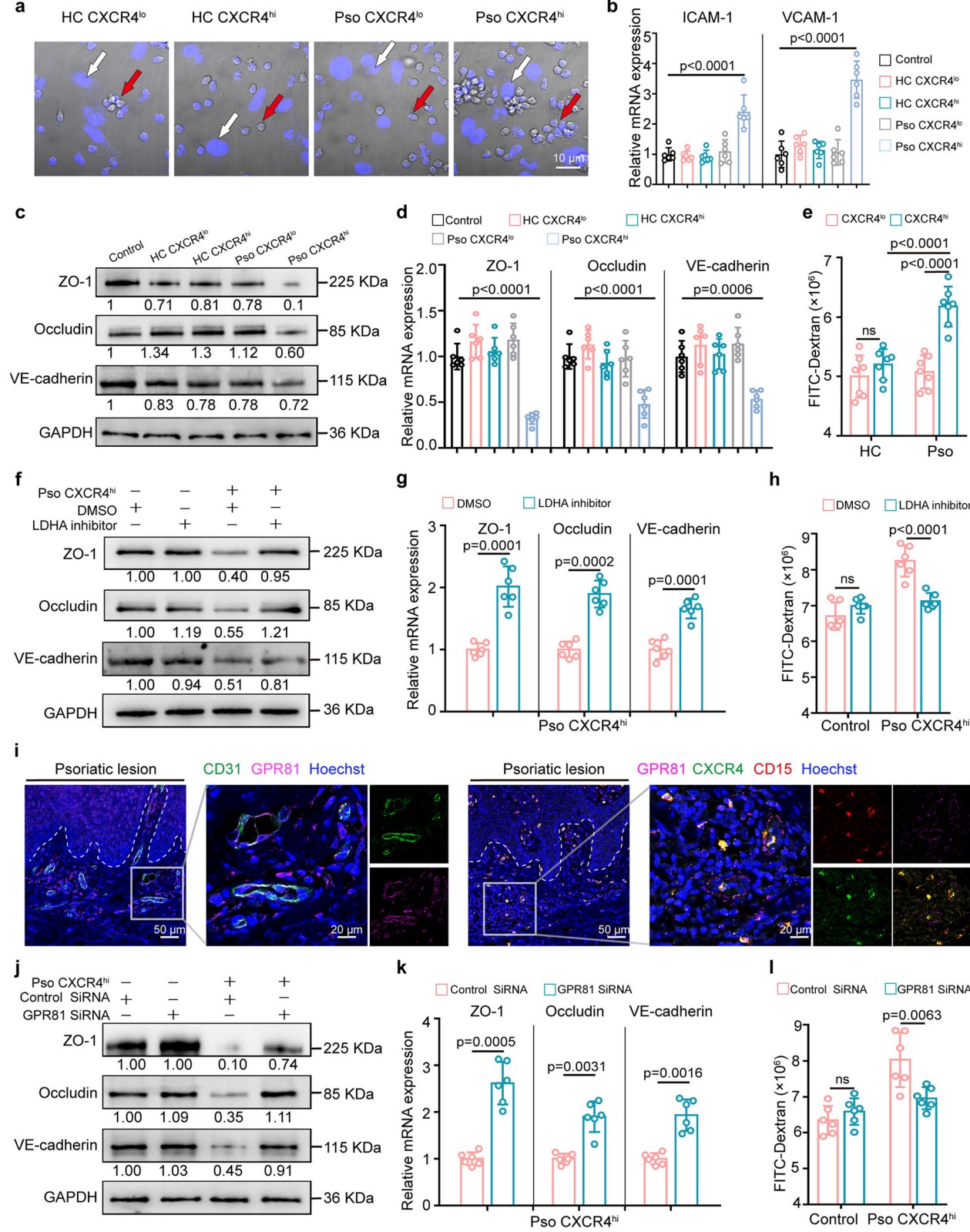

suppression of ZO-1, Occludin, and VE-cadherin expression induced by psoriatic CXCR4[hi] neutrophils, both at the protein (Fig. 4f, Supplementary Fig. 6a) and mRNA levels (Fig. 4g). In addition, LDHA inhibition partially reversed the increased permeability in HMEC-1 cells induced by CXCR4[hi] psoriatic neutrophils (Fig. 4h). GPR81 is a receptor for lactate[25], and we found that GPR81 was mainly expressed by CD31-positive ECs in psoriatic skin with CXCR4[hi] neutrophils in close

proximity (Fig. 4i). Moreover, co-culture with psoriatic CXCR4[hi] neutrophils led to approximately 2-fold increase in GPR81 expression in HMEC-1 cells (Supplementary Fig. 6b), which was reversed by LDHA inhibition (Supplementary Fig. 6c). Furthermore, siRNA knockdown of GPR81 (Supplementary Fig. 6d, e) recovered the reduction of ZO-1, Occludin, and VE-cadherin in psoriatic CXCR4[hi] neutrophils-treated HMEC-1 cells, both at the protein (Fig. 4j) and mRNA levels (Fig. 4k).

**Fig. 4 | Lactate released by CXCR4^hi neutrophils induces vascular remodeling and permeability. a** Representative images of adherent CXCR4^lo and CXCR4^hi neutrophils co-cultured with HMEC-1 cells. White arrows indicate HMEC-1 cells; red arrows indicate adherent neutrophils. Scale bar = 10 μm. **b** Relative mRNA expression of adhesion molecules in HMEC-1 cells with indicated treatment. Representative immunoblots (**c**) and qRT-PCR analysis (**d**) of tight junctions in HMEC-1 cells co-cultured with indicated treatment. **e** Stimulation of HMEC-1 cells with CXCR4^lo and CXCR4^hi neutrophils for 6 h, followed by analysis of leaked fluorescence intensity of FITC-dextran in a Transwell system. Representative immunoblots of HMEC-1 cells pre-incubated with LDHA inhibitor for 30 min following co-culture with psoriatic CXCR4^hi neutrophils (**f**) and qRT-PCR assessment of tight junction genes (**g**). **h** Analysis of released FITC-dextran in a Transwell system with HMEC-1 cells, pre-incubated with LDHA inhibitor for 30 min, followed by co-culture with psoriatic CXCR4^hi neutrophils. **i** Confocal images of CD31 (green) and GPR81 (purple) in psoriatic lesions (n = 6) with CD15^+CXCR4^hi neutrophils adjacent to GPR81^+ vascular ECs. Scale bar = 50 μm, 20 μm. The result was repeated three times independently with similar results. HMEC-1 cells were transfected with GPR81 siRNA and subjected to indicated treatment, followed by Western blot (**j**), qRT-PCR (**k**), and FITC-dextran leakage (**l**). The immunoblotting samples shown are from the same experiment (**c**, **f** and **j**) and blots were processed in parallel. Mean ± SD (n = 6 biologically independent samples/group). Analyses: two-way ANOVA with Tukey's post hoc test in (**b**), (**d**), (**e**), (**h**) and (**l**); One-way ANOVA with Tukey's post hoc test in (**g**) and (**k**). One or two-way ANOVA tests were performed as two-sided analyses and adjusted for multiple comparisons in the statistical analyses. ns, not significant. HC, healthy control; HMEC-1 cells, human microvascular endothelial cells; Pso, psoriasis patients. Source data are provided as a Source Data file.

In parallel, GPR81 knockdown with siRNA attenuated HMEC-1 permeability induced by psoriatic CXCR4^hi neutrophils (Fig. 4l). Collectively, these findings demonstrate that CXCR4^hi neutrophils modulate vascular permeability via lactate-GPR81 axis, likely facilitating immune cells trafficking into inflamed skin.

## CXCR4 expression in neutrophils is regulated by psoriasis-related mediators

To identify the regulators of CXCR4 expression in neutrophils, we stimulated neutrophils from healthy controls with a panel of pro-inflammatory cytokines including IL-17A, IL-23, IL-22, IL-36γ, IL-25, TNF, and CXCL12. IL-25, CXCL12, and TNF all increased the surface expression of CXCR4 in neutrophils at 2 h (Fig. 5a, Supplementary Fig. 7a). We further observed the elevation of serum CXCL12 in psoriasis patients compared to healthy controls (Fig. 5b). Moreover, increased CXCR4 expression on neutrophils was induced by psoriatic serum treatment at 2 h, which was attenuated by a CXCL12 neutralizing antibody (Fig. 5c, Supplementary Fig. 7b). We then re-analyzed publicly available single cell RNA sequencing (scRNA seq) data from psoriatic lesions and healthy controls, as previously reported[26]. This demonstrated that CXCL12^+ cells are increased in psoriatic lesions, among which ECs (12.37%), fibroblasts (73.87%), and pericytes (7%) were the main sources of CXCL12 in psoriatic lesions (Supplementary Fig. 7c, d). Multiple-color immunofluorescence also revealed that CXCL12 immunoreactivity co-localized prominently with ECs marked by CD31 and fibroblasts marked by vimentin (Fig. 5d) and demonstrated the colocalization of CXCL12 and CD31^+ ECs with CXCR4^hi neutrophils in close proximity in psoriatic lesions (Fig. 5e). We further showed induction of CXCL12 in HMEC-1 cells by either mixture of pro-inflammatory cytokines (IL-17A, IL-22, TNF, IL-1α, and oncostatin M) (Supplementary Fig. 7e), or serum from active psoriasis patients (Supplementary Fig. 7f).

Next, the kinetics of CXCR4 upregulation at the mRNA and protein levels were evaluated in response to TNF, IL-25, and CXCL12 for the indicated times (2, 4, 8, 12, 24 h). QRT-PCR showed increased mRNA expression of CXCR4 in human neutrophils at 2 h with stimulation of TNF, IL-25, or CXCL12, reaching peak level at 24 h (Fig. 5f). Flow cytometry analysis showed time-dependent overexpression of CXCR4 on neutrophil cell surface upon TNF, IL-25, or CXCL12 stimulation at early timepoints (<8 h), although lower in the CXCL12-treated group beyond 8 h (Fig. 5g), possibly due to CXCR4 internalization. Cell IF showed colocalization between CXCR4 and lysosome-associated membrane protein 1 (LAMP1) in CXCL12-treated neutrophils at 12 h (Fig. 5h). In addition, pre-treatment with brefeldin A, which reversibly blocks protein translocation from the endoplasmic reticulum to the Golgi apparatus, almost completely abrogated upregulation of CXCR4 on neutrophil membranes in the IL-25-, TNF-, or CXCL12-treated groups (Supplementary Fig. 7g). These results indicate that CXCR4 upregulation depends on de novo mRNA and protein synthesis, and on intracellular protein transport.

Imaging flow cytometry was employed to quantify the intensity and distribution of CXCR4 as previously reported[27]. CXCR4 did not co-localize with neutrophil granule markers (CD63 for azurophil granules, LCN2 for specific granules) in psoriatic or healthy CXCR4^lo neutrophils (Fig. 5i, Supplementary Fig. 8a), which was consistent with our cell IF data (Supplementary Fig. 8b). In addition, short (30 min) TNF stimulation, the degranulation trigger[28], did not induce an increase in the mean fluorescence intensity of CXCR4 on neutrophils (Supplementary Fig. 8c).

Overall, these results indicate that the immune microenvironment that includes TNF, IL-25, or CXCL12 may increase the expression of CXCR4 in neutrophils and that CXCR4 expression on neutrophils is regulated by de novo protein synthesis and involved in internalization and intracellular trafficking.

## CREB1 drives neutrophils towards pro-inflammatory CXCR4^hi phenotype

To identify the factors responsible for inducing CXCR4^hi neutrophils, we analyzed our RNA-seq data, and identified CREB1 as the most highly enriched transcriptional factor in CXCR4^hi neutrophils compared to controls (Fig. 6a). CREB1 is a known regulator of neutrophil activation under both acute and chronic inflammatory conditions[29]. QRT-PCR (Fig. 6b) and flow cytometry analysis (Fig. 6c, Supplementary Fig. 9a) showed that the mRNA level of CREB1 and protein level of p-CREB1, respectively, were elevated in CXCR4^hi neutrophils, and further enhanced in psoriatic patients. Both CXCL12 and TNF induced the phosphorylation of CREB1 in CXCR4^hi neutrophils (Fig. 6d, Supplementary Fig. 9b). Cell IF showed nearly undetectable p-CREB1 in CXCR4^lo neutrophils, but prominent cytoplasmic and focal nuclear localization in CXCR4^hi neutrophils (Fig. 6e). In addition, p-CREB1 was increased and co-localized with CD15 and CXCR4 in inflamed psoriatic skin (Supplementary Fig. 9c, d).

Next, we analyzed the binding of activated CREB to its coactivator protein, CREB-binding protein (CBP)[30]. CBP was constitutively expressed and co-localized with phosphorylation of CREB1 (S133) in TNF- or CXCL12-treated human neutrophils, but undetectable in normal controls (Fig. 6f). In accordance with this, the Western blot data demonstrated phosphorylation of CREB1 and increased CBP expression in TNF- or CXCL12-treated neutrophils (Fig. 6g). Therefore, we consider that the CREB-CBP complex contributes to CXCR4 expression and CXCR4^hi neutrophils in skin inflammation.

To explore the role of CREB1 in driving the pro-inflammatory effects of CXCR4^hi neutrophils, KG-501, a specific inhibitor that disrupts the CREB-CBP complex and inhibits CREB-target gene induction was used. The dose-response curves for KG-501 showed that the optimal inhibitory effects were reached at 300 μM (Supplementary Fig. 10a), without general adverse effects on cell viability (Supplementary Fig. 10b). Pre-treatment of psoriatic neutrophils with KG-501 (300 μM) for 60 min in vitro suppressed surface expression of CXCR4, maturation/activation marker including CD10, CD101, degranulation marker CD63 in CXCR4^hi neutrophils (Fig. 6h, Supplementary Fig. 10c), and the glycolytic proteins including HIF1A, HK2, GLUT1, and LDHA (Fig. 6i, Supplementary Fig. 11a), and decreased

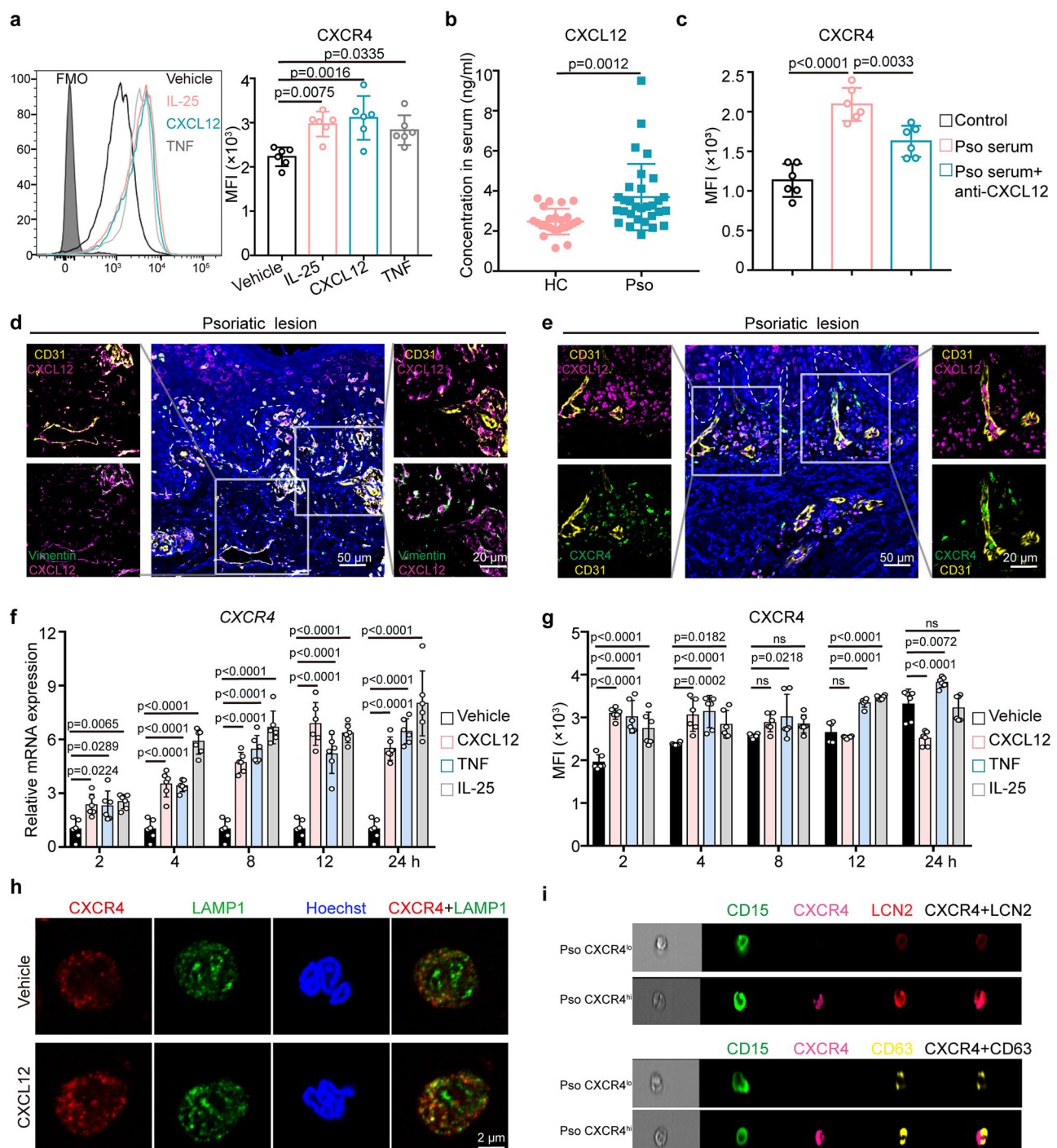

**Fig. 5 | CXCR4 expression in neutrophils is regulated by psoriasis-related mediators. a** Expression of CXCR4 on neutrophils stimulated with IL-25, CXCL12, and TNF for 2 h was measured by flow cytometry. **b** Serum level of CXCL12 in healthy controls (n = 24) and psoriasis patients (n = 30) was detected by ELISA. **c** Expression of CXCR4 on neutrophils with indicated treatment was measured by flow cytometry. **d** Representative immunofluorescence staining of CD31 (yellow), Vimentin (green), and CXCL12 (purple) in inflamed psoriatic skin (n = 6). Scale bar = 50 μm, 20 μm. **e** Representative immunofluorescence staining of CD31 (yellow), CXCR4 (green), and CXCL12 (purple) in inflamed psoriatic skin (n = 6). Scale bar = 50 μm, 20 μm. QRT-PCR analysis (**f**) and flow cytometry analysis (**g**) of CXCR4 in healthy neutrophils with indicated treatments. **h** Representative immuno-fluorescence co-staining of CXCR4 (red) and LAMP1 (green) in neutrophils. Scale bar = 2 μm. **i** Examples of low colocalization values in a sample of CXCR4[hi] and

CXCR4[lo] neutrophils of psoriasis patients stained for LCN2 (red) and CD63 (yellow) was visualized by ImageStream analysis. Images are from one representative experiment out of six. The scale bar indicates 7 μm. Isolated neutrophils were used for (**a**), (**c**), (**f**), and (**g**). Mean ± SD (n = 6 biologically independent samples/group). The immunofluorescence staining was repeated three times independently with similar results. Analyses: one-way ANOVA with Tukey's post hoc test in (**a**) and (**c**); two-way ANOVA with Tukey's post hoc test in (**f**) and (**g**); Unpaired Student's t-test in (**b**). The unpaired Student's t-test was conducted as two-sided tests. One or two-way ANOVA tests were performed as two-sided analyses and adjusted for multiple comparisons in the statistical analyses. ns, not significant. FMO, Fluorescence Minus One; HC, healthy control; LCN2, lipocalin 2; MFI, mean fluorescence intensity; Pso, psoriasis patients. Source data are provided as a Source Data file.

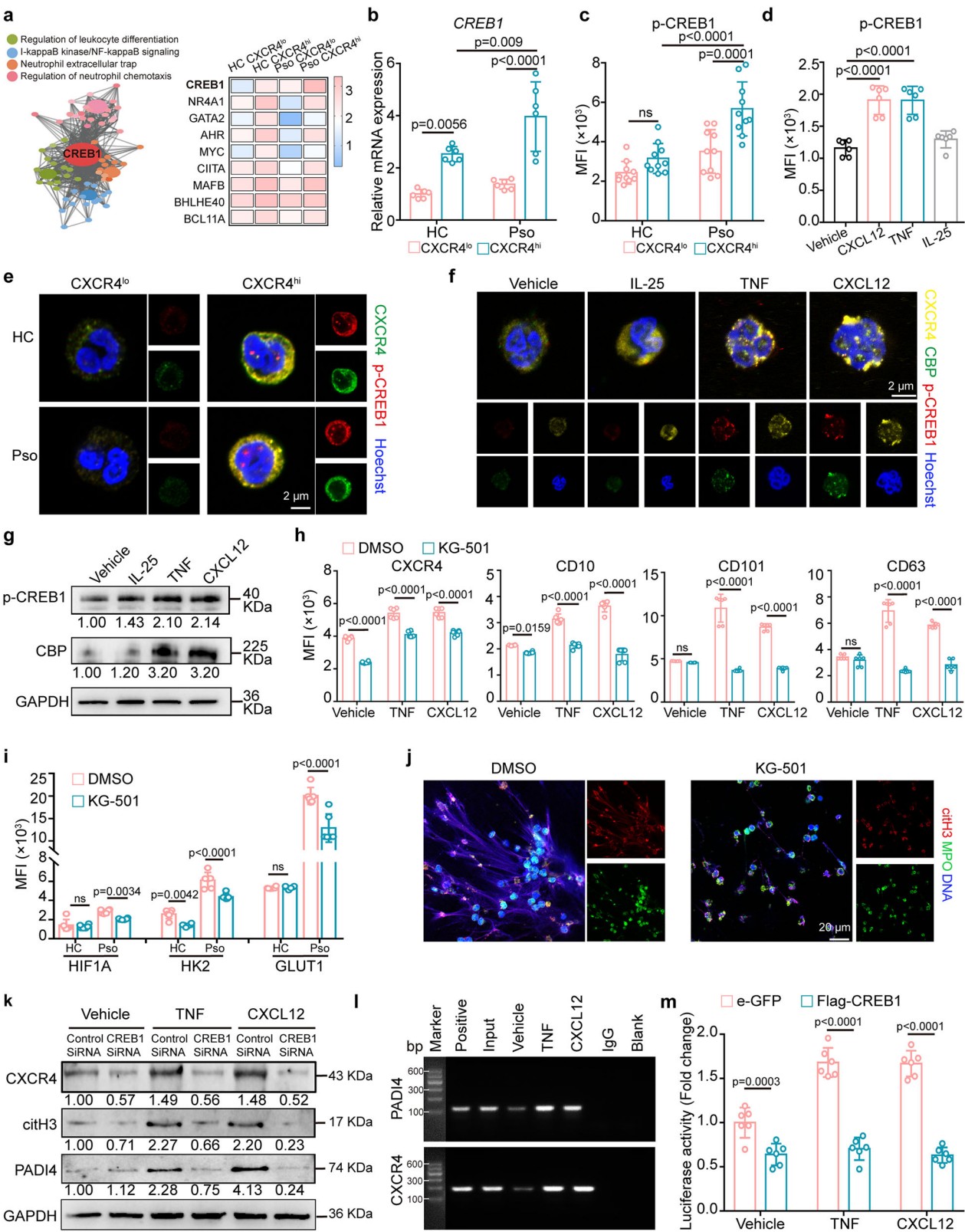

NETs formation (Fig. 6j, Supplementary Fig. 11b, c). Using siRNA-mediated knockdown of CREB1 in dHL-60 cells (Supplementary Fig. 11d, e), we observed reduced expression of CXCR4 in CXCL12- or TNF-treated dHL-60 cells by Western blot, accompanied by reduced citrullinated Histone3 (citH3) and protein-arginine deiminase type-4 (PADI4), both indicators of NETs formation (Fig. 6k). Moreover, chromatin immunoprecipitation (ChIP) assay confirmed the

recruitment of CREB1 to the promoter regions of CXCR4 and PADI4 in dHL-60 cells (Fig. 6l). To further determine this, we cloned the promoter region of CXCR4 into a luciferase construct and generated a deletion construct that lacked the predicted N-CREB1 binding site. Treatment with CXCL12 or TNF resulted in a significant increase in CXCR4 promoter expression, which was found to be inhibited by the deletion of the CREB1 binding site (Fig. 6m). These above results are

**Fig. 6 | CREB1 promotes the development of CXCR4$^{hi}$ neutrophils. a** Conceptual network diagram of genes showing the relationship between CREB1 and enrichment pathways, and heatmap of top transcription factors in CXCR4$^{hi}$ neutrophils. The mRNA expression of CREB1 (**b**) and protein levels of p-CREB1 (s133) (**c**) in CXCR4$^{lo}$ and CXCR4$^{hi}$ neutrophils. **d** Flow cytometry analysis of p-CREB1 (s133) in neutrophils treated with indicated stimulation. **e** Co-localization of CXCR4 (green) and p-CREB1 (red) in neutrophils from healthy controls and psoriasis patients. Scale bar = 2 μm. Representative staining (**f**) and Western blot (**g**) of CBP (green) and p-CREB1 (red) in neutrophils treated with indicated stimulation for 2 h. **h, i** Healthy neutrophils were pre-treated with CREB1 inhibitor (KG-501, 300 μM) for 1 h, followed by indicated stimulation for 2 h. Flow cytometry analysis of the expression of membrane molecules (**h**), as well as glycolysis (**i**). **j** Representative co-staining of DNA (Hoechst), Cit-H3 (citrullinated histone-3, red), and MPO (myeloperoxidase, green) to assess NETs formation in psoriatic neutrophils in vitro after pretreatment with KG-501 (300 μM) for 1 h. Scale bar = 20 μm. **k** Western blot of dHL-60 cells transfected with CREB1 siRNA and subjected to the indicated treatment. **l** Depiction of the CREB1 binding site in the *CXCR4* promoter for ChIP assay. **m** The luciferase activities of wild type CXCR4 and CXCR4 with the CREB1-binding site mutant were determined by luciferase reporter gene assays in dHL-60 cells. Isolated neutrophils were used for (**b**), (**d**), (**h**), and (**i**), and whole blood was used for (**c**). Mean ± SD (n = 6 biologically independent samples). The immunofluorescence staining was repeated three times independently with similar results. Blots for each antigen were processed in the same experiment in parallel. Two-way ANOVA with Tukey's post hoc test (**b, c, h, i**, and **m**) and one-way ANOVA with Tukey's post hoc test (**d**) were performed as two-sided analyses and adjusted for multiple comparisons in the statistical analyses. ns, not significant. HC, healthy control; MFI, mean fluorescence intensity; NETs, neutrophil extracellular traps; Pso, psoriasis patients. Source data are provided as a Source Data file.

consistent with CREB1 being a key driver for development of inflammatory CXCR4$^{hi}$ neutrophils.

## Targeting CXCL12/CXCR4 signaling alleviates skin inflammation

To address the contribution of CXCR4$^{hi}$ neutrophils to skin inflammation, we used the acute imiquimod (IMQ)-induced inflammatory skin model. The expression of both CXCL12 and CXCR4 was upregulated in inflamed skin of IMQ-induced skin inflammation (Fig. 7a), with CXCR4 being primarily found on infiltrating neutrophils (Fig. 7b). Serum CXCL12 level was also elevated in IMQ mice compared to control (Fig. 7c).

To deplete neutrophils, mice were injected intraperitoneally with an anti-Ly6G antibody (Supplementary Fig. 12a) as we previously reported[6], and then injected subcutaneously with fresh isolated homologous Ly6G$^+$CXCR4$^{lo}$ or Ly6G$^+$CXCR4$^{hi}$ neutrophils (Fig. 7d). Parameters of inflammation, including erythema scaling, acanthosis (epidermal thickening), and inflammatory infiltrates were suppressed by anti-Ly6G antibody treatment, whereas injection of Ly6G$^+$CXCR4$^{hi}$ neutrophils increased inflammation, as observed by visual inspection and H&E-staining (Fig. 7e). Epidermal thickness (Fig. 7f), immune cell infiltration (Supplementary Fig. 12b), and dermal vascular area (Fig. 7g, h) were increased in the Ly6G$^+$CXCR4$^{hi}$ neutrophil-treated group, whereas only a slight increase was seen in the Ly6G$^+$CXCR4$^{lo}$ neutrophil-treated group. A concurrent increase in mRNA expression of *Il17a*, *Tnf*, *S100a8*, and *S100a9* was noted in the Ly6G$^+$CXCR4$^{hi}$ neutrophil-treated IMQ group, compared with Ly6G$^+$CXCR4$^{lo}$ neutrophils-treated group (Fig. 7i).

To address whether therapeutic targeting of CXCR4 or its ligand CXCL12 improved skin inflammation, we used a neutralizing antibody against CXCL12, and in parallel an inhibitor against CXCR4 (AMD3100) as previously reported[31]. Concentration gradients of CXCL12 neutralizing antibody or AMD3100 were used along with assessment of skin inflammation (Supplementary Fig. 13a–f, Supplementary Fig. 14a–e). No side-effects or organ damage, including the kidney and liver, were observed with the treatments (Supplementary Fig. 13g, Supplementary Fig. 14f). Both CXCL12 inhibition and CXCR4 antagonism ameliorated psoriasis-like lesions in IMQ-treated mice (Fig. 8a), including epidermal thickness (Fig. 8b), and proportion of infiltrating Ly6G$^+$CXCR4$^{hi}$ neutrophils (Fig. 8c). Immunostaining of vascular cells by CD31 showed a reduction of vascular area in the dermis in both treatment groups (Fig. 8d), which was consistent with the quantification based on analysis of H&E images (Fig. 8e). Concurrent reduction of Evans blue dye leakage was noted in both anti-CXCL12 and AMD3100-treated groups, suggesting recovery of vascular permeability (Fig. 8f). Similarly, mRNA expression of pro-inflammatory mediators in skin lesions was suppressed by CXCL12 or CXCR4 inhibition, including *Il17a*, *Il1β*, *Il36*, *Il18*, and *S100a8* (Fig. 8g).

Notably, subcutaneous injection of recombinant murine CXCL12 in IMQ-treated mice over a 5-day period resulted in a severe inflammatory response, acanthosis, increased immune cell trafficking into skin, enhanced vascular area, and overexpression of inflammatory mediators (Supplementary Fig. 15a–e). Furthermore, rmCXCL12-treated IMQ mice had a greater proportion of Ly6G$^+$CXCR4$^{hi}$ neutrophils in both peripheral blood and inflamed skin, as demonstrated by flow cytometry and IF staining (Supplementary Fig. 15f, g).

In summary, these results demonstrate a pivotal role of CXCR4$^{hi}$ neutrophils in promoting skin inflammation and suggest targeting of CXCL12/CXCR4 axis may have a role in treatment of inflammatory skin diseases.

## Discussion

In this study, we present insights on the development and contribution of CXCR4$^{hi}$ neutrophils to inflammatory responses in skin. Thus, our data highlight the importance of CXCR4$^{hi}$ neutrophils in inducing vascular remodeling and permeability, facilitating infiltration of immune cells into tissues, as well as promoting heightened inflammatory responses. In addition, we identify the CXCR4/CXCL12 axis as a potential therapeutic target in inflammatory skin diseases (Fig. 9).

Neutrophils are the most abundant white blood cell in humans, and have a short lifespan, typically surviving for less than 24 h in the bloodstream but have been reported to last up to 5.4 days in some reports[32,33]. Although traditionally viewed as poorly plastic, neutrophils are now increasingly recognized as functionally diverse[34]. High-dimensional single-cell transcriptomic approaches have provided insights into their transcriptional heterogeneity during health and various disease states, revealing subtypes such as TGFβ1$^+$CCR5$^+$ neutrophils, Fth1$^{hi}$ neutrophils, and various TAN populations[15–18]. Cell-surface markers have also helped to identify circulating populations of neutrophils with variable functions, such as VEGFR1$^+$ neutrophils that promote angiogenesis in hypoxic tissues[35]. Similarly, studies in circulating human neutrophils have characterized discrete phenotypic subsets, including a population of CD177$^+$ neutrophils in patients with bacterial infections[28,36] and in a variety of autoimmune diseases including systemic lupus erythematosus (SLE)[37], or OLFM4 observed in sepsis[38]. These findings demonstrate that neutrophils in circulation exhibit a heterogeneous mixture of cells with diverse phenotypic and functional states[34]. However, the frequency of CD177$^+$ and OLFM4$^+$ neutrophils is not elevated in psoriasis patients compared to healthy controls (Supplementary Fig. 16a, b), and the actual degree of heterogeneity of neutrophils in psoriasis and the underlying mechanisms remain unclear.

Another model for studying neutrophil heterogeneity is the low-density neutrophils (LDNs) and normal-density neutrophils (NDNs) dichotomy, with pro-inflammatory properties of LDNs have been described in some inflammatory conditions[39]. Importantly, low-density granulocytes (LDGs) demonstrate further heterogeneity and can be divided into mature CD10$^+$ and immature CD10$^-$ neutrophils, each with different biologic function[24]. Previous studies have shown that the number of circulating LDNs and NDGs is higher in psoriasis patients compared to healthy controls[40]. LDNs are more efficient at

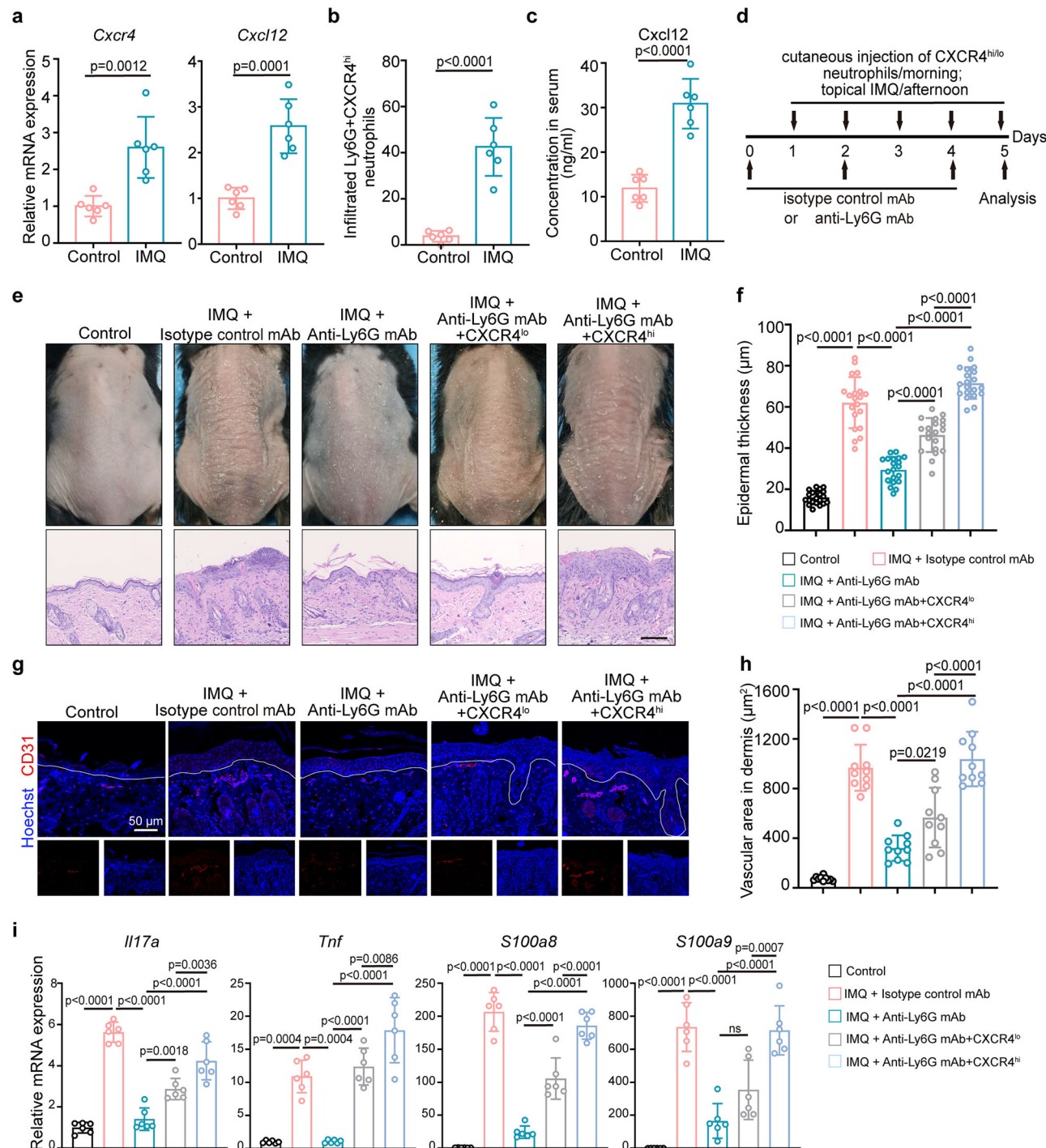

**Fig. 7 | CXCR4hi neutrophils contribute to psoriasis-like inflammation in vivo.** **a** Relative mRNA expressions of CXCR4 and CXL12 in mice tissues was evaluated by qRT-PCR. n = 6 mice. **b** Proportion of Ly6G+CXCR4hi neutrophils in mice skin was determined by flow cytometry. n = 6 mice. **c** Serum level of CXCL12 in control and IMQ-treated mice was detected by ELISA. n = 6 mice. **d** Schematic diagram of mouse experimental protocol. IMQ mice were injected intraperitoneally with an anti-Ly6G antibody every other day and then injected subcutaneously with fresh isolated homologous Ly6G+CXCR4lo or Ly6G+CXCR4hi neutrophils daily. **e** Phenotype and representative H&E staining of IMQ-treated mice in indicated groups on day 5. Images are representative of six individual mouse per group. Control group was topically applied with Vaseline cream. Bar = 200 μm. n = 6 mice. **f** Epidermal

thickness was assessed by H&E. n = 20 vision fields from 6 mice. **g** Representative immunofluorescence staining of CD31 (red) in inflamed skin. Scale bar = 50 μm. n = 6 mice. **h** Quantification of the dermal vascular area in the H&E-stained sections. n = 10 vision fields from 6 mice. **i** Relative mRNA expressions of inflammatory cytokines in mice tissues. n = 6 mice. The immunofluorescence staining was repeated three times independently with similar results. Mean ± SD. Analyses: unpaired Student's t-test in (**a**), (**b**), and (**c**); One-way ANOVA with Tukey's post hoc test in (**f**), (**h**), and (**i**). The unpaired Student's t-test was conducted as two-sided tests. One-way ANOVA test was performed as two-sided analyses and adjusted for multiple comparisons in the statistical analyses. ns, not significant; IMQ, imiquimod. Source data are provided as a Source Data file.

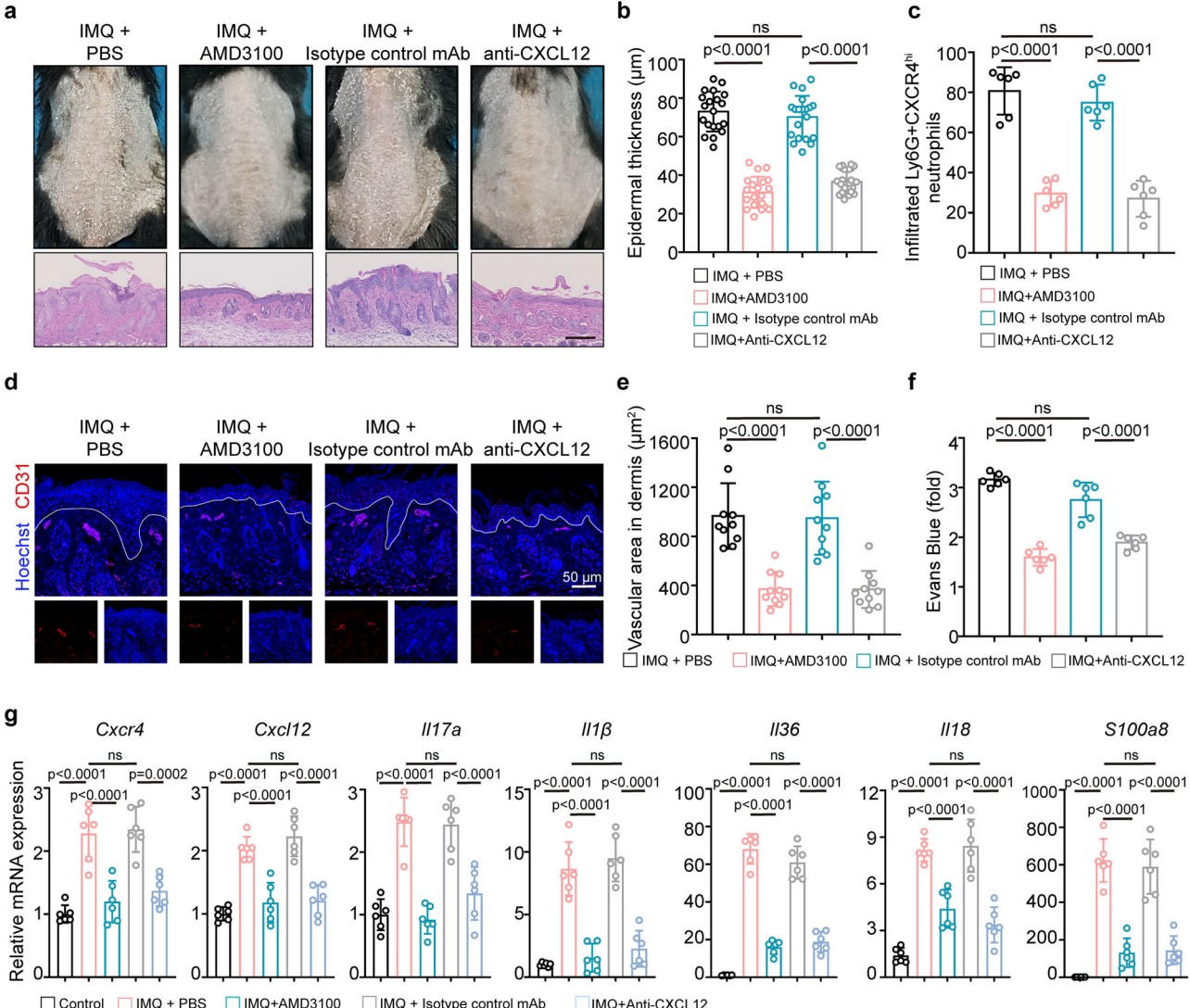

**Fig. 8 | Targeting CXCR4/CXCL12 axis alleviates IMQ-induced psoriasiform inflammation. a** Phenotype and H&E staining of IMQ-treated mice in different treatment groups. Images are representative of six individual mice per group. The mice in the control group was topically applied with vaseline cream. Bar = 200 μm. n = 6 mice. **b** Epidermal thickness as assessed by H&E staining. n = 20 vision fields from 6 mice. **c** Proportion of Ly6G$^+$CXCR4$^{hi}$ neutrophils in inflamed skin was determined by flow cytometry. n = 6 mice. **d** Representative immunofluorescence staining of CD31 (red) in lesional skin of indicated groups. Scale bar = 50 μm. n = 6 mice. **e** Quantification of the dermal vascular area in the H&E-stained sections. n = 10 vision fields from 6 mice. **f** Quantification of extracted Evans blue dye in inflamed skin in indicated groups. n = 6 mice. **g** Relative mRNA expressions of inflammatory cytokines in mice tissues. n = 6 mice. The immunofluorescence staining was repeated three times independently with similar results. Mean ± SD. One-way ANOVA with Tukey's post hoc test was performed as two-sided analyses and adjusted for multiple comparisons in the statistical analyses. ns, not significant; IMQ, imiquimod. Source data are provided as a Source Data file.

generating NETs[41]. Although it is reported that there are no significant differences in the proportion of aged neutrophils (CXCR4$^+$CD62L$^{low}$) between LDNs and NDNs[42], the CXCR4$^{hi}$ neutrophils identified in our study appear to have some overlap with mature LDNs[40], on account of similar functional characteristics, including enhanced leukocyte activation, increasing vascular permeability, pro-inflammatory effects, NF-kappa B signaling, NETs formation, enhanced phagocytosis, etc.

Neutrophils progressively upregulate CXCR4 during their lifetime, along with loss of CD62L, and represent an overly active neutrophil phenotype with enhanced NETs formation[21]. Our data are consistent with this scenario, but further demonstrate that CXCR4$^{hi}$ neutrophils are significantly increased at baseline during systemic inflammatory states, such as in patients with active psoriasis. Strikingly, our data show that CXCR4$^{hi}$ neutrophils are not just a marker of this heightened inflammatory state but also play a highly active role in amplifying this inflammatory state, through increasing vascular permeability and expression of vascular adhesions molecules to facilitate

the influx of inflammatory cells into psoriatic skin. It is likely that the contribution of neutrophils may differ in different inflammatory skin diseases, and one limitation of the data presented here is that it is not currently feasible to directly study neutrophil interactions within tissues. However, our data provide a fundamental shift in our view of the contribution of neutrophils to skin inflammation and highlight CXCR4$^{hi}$ neutrophils and the CXCR4/CXCL12 axis as a potential therapeutic target.

Neutrophils display complex gene expression patterns depending on their developmental stage, activation state, and tissue microenvironment and this is closely regulated by different transcription factors[43]. CREB1 has been implicated in neutrophil biology and shown to be activated by TNF stimulation, and to regulate pro-inflammatory chemokines and cytokines including CXCL8, CCL3, CCL4, and TNF[44]. CREB1 has also been shown to contribute to neutrophil degranulation[45], and NADPH oxidase activity mediating formation of NETs[29]. In other cell types, such as pancreatic islet cells, CREB1

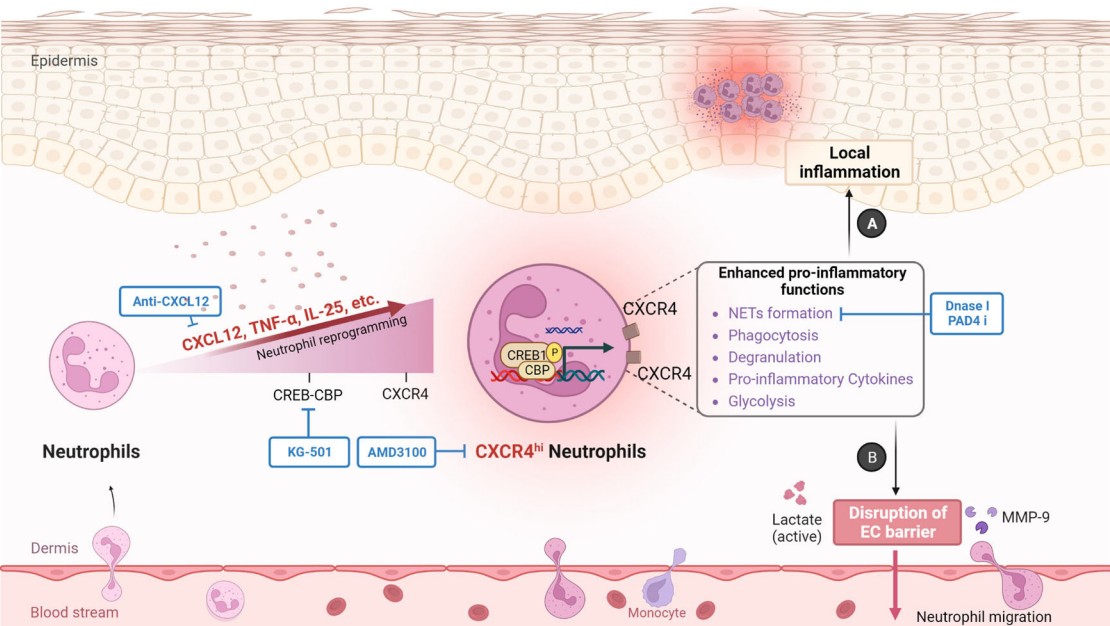

**Fig. 9 | Proposed mechanisms of CXCR4ʰⁱ neutrophil phenotype in skin inflammation.** This study provides insights into the development and role of CXCR4ʰⁱ neutrophils in skin inflammation. Compared to CXCR4ˡᵒ neutrophils, CXCR4ʰⁱ neutrophils exhibit increased capacity for NETs formation, phagocytosis, and degranulation, as well as higher expression of pro-inflammatory mediators, supported by activated glycolysis. CXCR4 expression on neutrophils is induced by CXCL12, IL-25, and TNF, which requires de novo mRNA and protein synthesis and intracellular protein transport, but is not stored in neutrophil granules and regulated by degranulation. CREB1 is activated by phosphorylation on Ser-133 upon stimulation, enabling interaction with its coactivator protein CBP to initiate transcription of CREB-responsive genes, thereby contributing to CXCR4 expression and CXCR4ʰⁱ neutrophils in skin inflammation. Our data further emphasize the crucial role of CXCR4ʰⁱ neutrophils in promoting vascular remodeling through lactate and MMP-9 release (B), facilitating immune cell infiltration into tissues, and increasing inflammatory responses (A). Moreover, we identify the CXCR4/CXCL12 axis as a potential therapeutic target for inflammatory skin diseases, with a focus on CREB1, CXCR4/CXCL12, or the formation of NETs. CBP, CREB-binding protein; EC, endothelial cell; MMP9, matrix metallopeptidase 9; NETs; neutrophil extracellular traps; PAD4 i, PAD4 inhibitor. Created with BioRender.com.

regulates metabolic shifts towards glycolysis[46]. An important finding presented here is the role of CREB1 in promoting CXCR4 expression in neutrophils, and it sheds light on the key role of this transcription factor in neutrophil plasticity and inflammatory responses. As demonstrated here, through both knockdown and chromatin immunoprecipitation approaches, CREB1 is an essential regulator of both CXCR4 and PADI4, an enzyme that converts arginine into citrulline and promotes the formation of NETs[47]. Upon stimulation, CREB is phosphorylated on Ser-133, allowing it to interact with its coactivator protein, CBP, to initiate transcription of CREB-responsive genes[30]. KG-501 directly targets the KIX domain of CBP, resulting in a disrupted CREB-CBP complex, inhibiting CREB-targeted gene induction. We further demonstrate that the CREB-CBP complex contributes to CXCR4 expression and CXCR4ʰⁱ neutrophils in skin inflammation. While other transcriptional factors have not been excluded, especially c-Jun and CCAAT enhancer binding protein β, C/EBPβ[48]. These data establish CREB1 as a critical regulator for the generation and function of this pro-inflammatory CXCR4ʰⁱ neutrophil phenotype.

Neutrophils exhibit dynamic metabolic adaptations to exert specific functions[49]. Neutrophils must rapidly access ATP to enable various cellular responses and therefore have a dependence upon glycolysis for ATP production[49]. We found that CXCR4ʰⁱ neutrophils demonstrate a shift towards glycolytic metabolism, and consistent with this a prior report has demonstrated that CXCL12 may play a role in this shift and promote glycolytic reprogramming, as shown in acute myeloid leukemia cells[50]. This increase in glycolytic metabolism is accompanied by increased lactate release from neutrophils. Lactate exerts important immunomodulating effects, including amplification of IL-17A production and retention of CD4⁺ T cells in tissues including rheumatoid arthritis synovium[51]. In addition, lactate may increase vascular permeability in bone marrow vascular cells via reducing VE-cadherin

expression, a mechanism that may play a critical role in sepsis[25]. Neutrophils can contribute to vascular inflammation through secretion of MMP-9[52], NETs[53], extracellular vesicles[54], or direct platelet interactions[40]. For instance, neutrophils and NETs[53], as well as MMP-9[52], can degrade glycocalyx on the EC surface to expose adhesion molecules and increase endothelial permeability. As we have recently demonstrated, endothelial glycocalyx destruction is a major feature of EC dysfunction in psoriasis and a driving force facilitating immune cell extravasation[55]. We propose that CXCR4ʰⁱ neutrophils exert their pro-inflammatory effects on ECs via different molecules and mechanisms. This neutrophil-vascular crosstalk has previously been appreciated and is likely to have a significant impact on how neutrophils facilitate the entry of other inflammatory cells into tissues.

Our data further show the important role of CXCL12, IL-25, and TNF as inducers of CXCR4 expression on neutrophils. TNF is one of the key pro-inflammatory mediators in psoriasis[56] and is also increased in multiple other inflammatory skin diseases[57]. TNF has also been shown to regulate CXCR4 expression in other inflammatory diseases[58]. In this study, we have demonstrated that CXCL12⁺ cells are increased in psoriatic skin, with ECs, fibroblasts, and pericytes being the main sources of CXCL12. CXCL12 is increased in postcapillary venular cells (vessel endothelial cell cluster 3) involved in leukocyte adhesion and migration[26]. Furthermore, CXCL12 is induced in ECs in response to psoriasis-related stimuli including IL-17A (Supplementary Fig. 7e, f). This aligns with the normalization of CXCL12 serum level after therapeutic targeting of IL-17A and may provide an explanation for the decreased level of CXCR4ʰⁱ neutrophils following effective psoriasis treatment. The clinical relevance of the CXCR4/CXCL12 axis has been demonstrated in rheumatoid arthritis and multiple sclerosis[59]. Our findings extend this to inflammatory skin diseases, and in particular psoriasis. However, CXCR4 is expressed by other cell types beyond

neutrophils, including B cells, T cells, endothelial, and epithelial cells[60], and the function of CXCR4 on different cell types may need further study, but are beyond the scope of this current study.

Notably, CXCR4 upregulation depends on de novo mRNA and protein synthesis and on intracellular protein transport but is not stored inside neutrophil granules and mobilized to the surface upon degranulation. CXCR1, 2, and 4 have been described to undergo receptor internalization, modifying the interaction and activation potential of different cells[61]. It is further reported that internalization, as well as lysosomal degradation of CXCR4, is regulated by complicated posttranslational modifications, including phosphorylation and ubiquitination[62]. The rare autoimmune disease WHIMS is caused by impaired desensitization and internalization of CXCR4, leading to enhanced chemotactic responsiveness to CXCL12[63]. However, the mechanisms that regulate cell surface expression, membrane trafficking, and recycling of CXCR4 in neutrophils in skin inflammation are largely unknown and warrant further in-depth investigation.

Recent data suggest that the trafficking and recruitment of leukocytes are rhythmic during light-dark cycles, both under physiological conditions and during inflammation[64]. In healthy states, CD62L$^{lo}$CXCR4$^{hi}$ neutrophils that have "aged" in the circulation are eliminated at the end of the resting period in mice[65]. CXCR4 expression is relatively low at 7:00 - 8:00 am in the morning compared to different times of the day, and all blood samples in our study were collected between 7:00 - 8:00 am and immediately processed for cell isolation and serum collection.

Based on the results described above, we have performed a variety of interventions centered on CREB1-CXCR4$^{hi}$ neutrophils. KG-501 inhibits CREB1 activation in psoriatic neutrophils, AMD3100 or CXCL12 neutralizing antibody blocks the CXCL12-CXCR4 axis and accumulation of CXCR4$^{hi}$ neutrophils, and DNaseI removes the formed NETs as we previously reported[8] (Fig. 9); all of these agents prevent skin inflammation to varying degrees. Greater in-depth exploration and validation of the contribution of the CREB1-CXCR4$^{hi}$ neutrophils axis will require the use of complex conditional knockout mice and will need to be addressed in future research. Another limitation is the potential of non-neutrophil contaminants in vitro. To minimize this, we set the purity of isolated neutrophils, including CXCR4$^{hi}$ neutrophils, at > 95%, although we cannot completely exclude the possibility that a small number of monocytes/macrophages are included among the neutrophils, this would have been unlikely to have a major impact on our results. Furthermore, the specificity of CXCR4$^{hi}$ neutrophils to specific types of skin inflammation remains to be addressed. We have found that CXCR4$^{hi}$ neutrophils accumulate in skin lesions of psoriasis vulgaris but not in generalized pustular psoriasis. Further study is needed to fully figure out the neutrophil heterogeneity and phenotypes in skin inflammation.

A major challenge in studying neutrophil subsets is the lack of genetic and molecular tracers to accurately distinguish between heterogeneous populations beyond surface markers[66]. Unlike T cell subsets, which can be categorized based on their origin, transcriptional profile, and genetic drivers[67], neutrophil heterogeneity remains poorly understood due to a lack of molecular resolution. Though single-cell sequencing has revealed a range of transcriptional states for neutrophils, the absence of reliable markers makes it difficult to formally classify them into distinct subsets[68]. Additionally, these states exist as a transcriptional continuum rather than independent clusters, further complicating their classification. As discussed recently[66], despite recent advances, our understanding of neutrophil heterogeneity, adaptability, and contributions to diseases remains limited. To address this, advanced barcoding tools coupled with scRNA seq or genetic studies in vitro and in vivo may be required to fully explore the global architecture of the neutrophil compartment, the specific tissue niches that enable functional reprogramming, and potential clinical applications.

In summary, the present study demonstrates the development and pro-inflammatory role of CXCR4$^{hi}$ neutrophils in both promoting vascular permeability, as well as amplifying inflammatory responses in skin. These findings gain a deeper insight into the diversity of neutrophils and highlight the pathogenic and potential therapeutic value of a critical neutrophil CXCR4/CXCL12 axis in inflammatory skin diseases.

## Methods

### Research compliances
All analyses of human materials were done in full agreement with our institutional guidelines, with the approval of the Ethical committee of the Xijing Hospital, the Fourth Military Medical University (KY20203171-1). Written informed consent was obtained from each participant. This study is compliant with the "Guidance of the Ministry of Science and Technology (MOST) for the Review and Approval of Human Genetic Resources." All animal procedures complied with the National Institutes of Health Guide for the Care and Use of Laboratory Animals and with Institutional Animal Care and Use Committee approval at the Fourth Military Medical University. Animal Experimental Ethical Inspection was approved Laboratory Animal Welfare and Ethics Committee of Fourth Military Medical University.

### Patient Selection
Patients enrolled in our study fulfilled the diagnostic criteria for psoriasis and disease activity was scored by the PSAI. Patients were randomly recruited from outpatient, inpatient and were eligible to participate if they: were ≥18 years of age; reported regular work schedules and sleep-wake patterns in the preceding 4 weeks; without other autoimmune or systemic diseases and were not receiving systemic treatment in the recent 4 weeks. Controls were collected from sex-, and age-matched healthy volunteers. The patients were recruited according to the objective criteria of age, gender, and health status and no self-selection bias affected the recruitment. Demographic information of patients is provided in Supplementary Table 1.

### Experimental Animals
C57BL/6 J mice (8–10 weeks old) were purchased from Department of Laboratory Animal Medicine of the Fourth Military Medical University (Xian, Shaanxi, China; permit number: 2019-001). Both male and female were used for all experiments. Mice were randomly assigned to groups of 6 mice, then bred and maintained in a specific pathogen-free barrier facility. For in vivo experiments, researchers were blinded to the treatment of each animal received until data were analyzed. After the experiments, mice were euthanized by an overdose of sodium pentobarbital.

For psoriasis-like mouse model, mice received daily topical applications of 62.5 mg IMQ cream (5% IMQ, INova Pharmaceuticals, 3 M Health Care) on the shaved dorsal skin for consecutive 5 days. Neutrophil depletion was done using intraperitoneally injection of purified anti-Ly6G antibody (127649, Biolegend, USA) *vs.* isotype control antibody (400565, Biolegend) every other day. The dose of the first injection was 100 μg, and the subsequent injection dose was 50 μg. IMQ mice were subcutaneously injected with purified Ly6G$^+$CXCR4$^{lo}$ or Ly6G$^+$CXCR4$^{hi}$ neutrophils (6×10$^5$/mouse) isolated from peripheral blood of homologous mice. Schematic diagram of experimental protocol is shown in Fig. 7d. Peripheral blood of C57BL/6 wild-type mice was collected, and mouse neutrophils were isolated using a magnetic-activated cell sorting method by mouse Ly6G MicroBeads (130-120-337, Miltenyi Biotec Inc., USA). CXCR4$^{hi}$ neutrophils were also obtained by positive selection from total neutrophils using mouse CXCR4 MicroBeads (130-118-682, Miltenyi Biotec Inc.) according to the manufacturer's protocol, as depicted below.

In our in vivo experiment, 10 mg/kg of AMD3100 hydrate (A5602, Sigma-Aldrich, USA), a selective CXCR4 antagonist was administered

intraperitoneally every day to inhibit CXCR4 function. AMD3100 was dissolved in PBS (PC-00003, PlantChemMed, Shanghai, China) and PBS served as vehicle control. To neutralize CXCL12, IMQ mice were administered intraperitoneally with anti-mouse CXCL12 monoclonal antibody (1 mg/kg, MAB310, R&D Systems, USA) or control mouse IgG (1 mg/kg, MAB002, R&D Systems) every other day. Moreover, a bolus injection of AMD3100 (0.1, 1, or 10 mg/kg) and CXCL12 monoclonal antibody (0.1, 0.5, or 1 mg/kg) was administered intraperitoneally at various concentrations to explore the optimal inhibitory concentration (Supplementary Fig. 13, 14).

To investigate cutaneous vascular permeability, Evans blue dye (50 mg/kg, E8010, Solarbio technology, Beijing, China) was injected by tail vein. 2 h later, the shaved back was resected (10 mm in diameter) and incubated in 1 ml formamide (V900064, Sigma-Aldrich) at 56 °C for 48 h to extract the dye. Absorbance of extravasated Evans blue dye was measured at 610 nm wavelength and the relative absorbance of extravasated dye was normalized to that of PBS.

### Blood sampling handling and neutrophil isolation

To eliminate the influence of circadian rhythm, all blood samples in this study were collected at 7:00 - 8:00 am and immediately processed for cell isolation and serum collection. Then isolated neutrophils were processed immediately for the next experimental steps and serum was frozen at -80 °C. Moreover, all selected participants in this study reported both regular work schedules and sleep-wake patterns in the preceding 4 weeks. As neutrophil stimulation assays were performed at different times of the day, the data regarding percentages of CXCR4$^{hi}$ neutrophils or CXCR4 expression were normalized by calculating the ratio between the values of samples containing stimulus and the values of samples with corresponding controls in each test.

4 ml blood was layered on top of 4 ml of Polymorphprep (1114683, Axis-Shield, Norway) in a 15 mL centrifugation tube. The tube was centrifuged at 500 g at 20 °C for 30 min. The polymorphonuclear cell layer was collected and red blood cells were removed using Red Blood Lysing Buffer (FXP001, 4 A Biotech Co., Ltd, Beijing, China). Freshly isolated neutrophils were suspended at $1\times10^7$/ml in PBS. For separation of CXCR4$^{hi}$ neutrophils, cells labeled with MACS beads are captured by the magnetic field of the separator (Miltenyi Biotec Inc., 130-090-312), whereas unlabeled cells pass the magnetic field and end up in the flow-through fraction. In short, freshly isolated neutrophils were re-suspended in 100 μL MACS Separation Buffer (130-091-221, Miltenyi Biotec Inc.) and stained with APC-conjugated anti-human CXCR4 antibody (2 μL/10$^7$ cells) for 10 min, cells were washed and then incubated with anti-APC microbeads (20 μL/10$^7$ cells, Miltenyi Biotec Inc., 130-100-070) in 80 μL MACS Separation Buffer (130-091-221, Miltenyi Biotec Inc.) for 15 min in the dark. Subsequently, the CXCR4$^{hi}$ neutrophils were separated on an MS column (130-042-201, Miltenyi Biotec Inc.) on a MACS Separator (130-090-312, Miltenyi Biotec Inc.) and washed again to detach from the antibody-magnetic bead. Isolation efficiencies were analyzed by incubating cells with FITC conjugated anti-human CD15 (301904, 1:100, BioLegend) and PE-Cy7 conjugated anti-human CXCR4 (306514, 1:100, BioLegend), for 30 min at 4 °C (Supplementary Fig. 17a).

Mouse neutrophils were isolated from the peripheral blood of healthy 8-week-old C57Bl6/J mice. Mice were anesthetized via intraperitoneal injection of 1% sodium pentobarbital (4579, 100-150 μL/mouse, R&D Systems), and the eyeball was removed to collect blood samples. Peripheral blood was taken from 10 mice each time to achieve neutrophil counts and erythrocytes were removed using Red Blood Lysing Buffer (FXP001, 4 A Biotech Co., Ltd). Up to 10$^8$ cells were resuspended in 200 μL of MACS buffer (130-091-221, Miltenyi Biotec Inc.) and incubated with 50 μL of anti-Ly6G biotin beads per sample (130-120-337, Miltenyi Biotec Inc.) for 15 min in the dark at 4 °C. The cells were then washed and centrifuged, resuspended in MACS buffer, and passed through a LS column (130-042-401, Miltenyi Biotec Inc.) on a MACS

Separator (130-042-303, Miltenyi Biotec Inc.). The magnetically labeled Ly6G$^+$ cells were retained on the column. The labeled cells were collected, washed once with MACS buffer, and centrifuged at 300 g for 10 min. The isolated neutrophils were resuspended in PBS for subsequent experiments.

To separate mouse CXCR4$^{hi}$ and CXCR4$^{lo}$ neutrophils, freshly isolated neutrophils were re-suspended in 100 μL MACS Separation Buffer (130-091-221, Miltenyi Biotec Inc.) and stained with PE-conjugated anti-mouse CXCR4 antibody (2 μL/10$^7$ cells, 130-118-682, Miltenyi Biotec Inc.) for 10 min, cells were washed and then incubated with anti-PE microbeads (20 μL/10$^7$ cells, 130-048-801, Miltenyi Biotec Inc.) in 80 μL MACS Separation Buffer (130-091-221, Miltenyi Biotec Inc.) for 15 min in the dark. Then, mouse CXCR4$^{hi}$ neutrophils were separated on LS column (130-042-401, Miltenyi Biotec Inc.) on a MACS Separator (130-042-303, Miltenyi Biotec Inc.) and washed again. Isolation efficiencies were analyzed by incubating cells with FITC anti-mouse Ly6G (127606, 1:100, BioLegend) and PE anti-mouse CXCR4 (146506, 1:100, BioLegend) for 30 min at 4 °C. The purity of the cells was routinely 90 to 95%, (Supplementary Fig. 17b).

### Flow cytometry analysis

For the analysis of neutrophils phenotypes, total blood leukocytes from healthy controls and psoriasis patients were washed and incubated with the following primary antibodies: FITC conjugated anti-human CD15 (301904, 1:100), PE-Cy7 conjugated anti-human CD15 (301924, 1:100), PE conjugated anti-human CXCR4 (306506, 1:100), PE-Cy7 conjugated anti-human CXCR4 (306514, 1:100), PE-Cy5 conjugated anti-human CD62L (304808,1:100), PE conjugated anti-human CD11b (393112, 1:100), Pacific/Blue conjugated anti-human CD11b (301315, 1:100), PE anti-human CD44 (338808, 1:100), PerCP/Cy5.5 conjugated anti-human CD101 (331016, 1:100), PE-Cy5 anti-human CD10 (312206, 1:100), APC/Cy7 conjugated anti-human CD10 (312212,1:100) (all from BioLegend) for 30 min at 4 °C. For intracellular staining, neutrophils were isolated and stained with FITC anti-human CD15 (301904, 1:100, BioLegend) and PE-Cy7 anti-human CXCR4 (306514, 1:100, BioLegend) for 30 min at 4 °C in the dark; after washing with PBS, cells were incubated with fixation/permeabilization buffer (562574, BD Pharmingen) at 4 °C for 50 min and then incubated with Rabbit monoclonal to phospho-CREB1 (9198 S, 1:800, Cell Signaling Technology (CST), USA), Rabbit monoclonal to HK2 (209847, 1:60, Abcam, USA), Rabbit monoclonal to GLUT1 (115730,1:40, Abcam), and PE-anti-human HIF1A (359704, 1:100, BioLegend). After centrifuging at 350 g for 5 min, cells were incubated with PE Donkey anti-Rabbit IgG (406421, 1:100, BioLegend) for 30 min at 4 °C in dark. After washing with PBS, the cells were analyzed by flow cytometry (649225, BD LSRFortessa Cell Analyzer), and data were analyzed with Flowjo v10.8.1 (Tree Star). We used the fluorescence minus one (FMO) as a negative control, which contains all the fluorochromes except for the one that is being measured.

### Imaging Flow Cytometry

CXCR4$^{hi}$ and CXCR4$^{lo}$ neutrophils isolated from healthy controls and psoriasis patients were stained with FITC anti-human CD15 (301904, 1:100, BioLegend) and PE-Cy7 anti-human CXCR4 (306514, 1:100, BioLegend) for 30 min at 4 °C in the dark. After washing, cells were fixed and permeabilized with a fixation/permeabilization kit (562574, BD Pharmingen, USA) according to the manufacturer's instructions and incubated with the antibodies, PE conjugated anti-human CD63 (353004, 1:100, BioLegend), Rabbit monoclonal to Lipocalin-2 (125075, 1:400, Abcam), and Rabbit monoclonal to MMP9 (76003, 1:500, Abcam) for 40 min at 4 °C in the dark. Samples were then washed with PBS and incubated with secondary antibodies conjugated with APC anti-Mouse IgG Antibody (406610, 1:100, Biolegend) or Brilliant Violet 421 Donkey anti-Rabbit IgG (minimal x-reactivity) Antibody (406410, 1:100, Biolegend) for 40 min at 4 °C in the dark. The cells were analyzed using an imaging flow cytometer (ImageStream Mark II, Luminex) at a

magnification of 40X. The IDEAS 6.2 software (Amnis) was used to visualize and analyze samples for marker expression. Single stained control cells were used to compensate for fluorescence between channel images, to prevent overlap of emission spectra. Cells were gated for single cells based on the area and aspect ratio features, and for focused cells using the Gradient RMS feature. Finally, cells were gated for positive staining based on their pixel intensity.

## Phagocytosis

To assess the phagocytic capacity, neutrophils from healthy controls and psoriasis patients were isolated and incubated with pHrodo Green *E coli* (2 mg/mL; P35366, Thermo Fisher Scientific, USA) at 37 °C for 30 min. Incubation was stopped by the addition of 2 mL of PBS, and cells were then washed 3 times. Cells were divided into two parts, the first of which was used for flow staining. After incubated with PE-Cy7 conjugated anti-human CD15 (301924, 1:100, BioLegend) and PE conjugated anti-human CXCR4 (306506, 1:100, BioLegend) at 4 °C for 30 min. Phagocytic uptake was analyzed by flow cytometry and expressed as median fluorescence intensity (MFI). The remaining cells, which were not used for flow staining, were incubated with Hoechst 33258 (C0021, 1:1000, Solarbio technology) for 15 min for following confocal microscope (LSM880, Carl Zeiss, Germany).

## Glycolytic activity analysis

Supernatants of CXCR4$^{lo}$ and CXCR4$^{hi}$ neutrophils were collected after 6-hour cell culture, and lactate was assessed via a lactate assay kit (A019-2-1, Nanjing Jiancheng Bioengineering Institute, Nanjing, China) according to the manufacturer's instructions. For glucose uptake detection, isolated neutrophils were incubated with 100 mM 2-NBDG (N13195, Invitrogen) for 2 h and then stained with PE-Cy7 conjugated anti-human CD15 (301924, 1:100, BioLegend) and PE conjugated anti-human CXCR4 (306506, 1:100, BioLegend) at 4 °C for 30 min before measuring fluorescence by flow cytometry.

## Cell transfection

Synthetic small interfering RNA (siRNA) duplexes against human CREB1 and GPR81 were purchased from Beijing Baiaopuke Biotechnology. The siRNA sequences used were as follows: CREB1, GCCUGCAAACAUUAACCAUTT (forward), AUGGUUAAUGUUUGCAG GCCC (reverse); GPR81, GCGUGUCUGC- UAGACUCUATT (forward), UAGAGUCUAGCAGACACGCTG (reverse). dHL60 cells were seeded in the 6-well plates at the concentration of 5×10$^5$ cell/well and transfected with CREB1 siRNAs using lipofectamine 2000 transfection reagent (MF135-01, Mei5 biotechnology, Beijing, China) following manufacturer's protocols. For transient knockdown of GPR81 in HMEC-1 cells, siRNAs were transfected using lipofectamine 3000 transfection reagent (L3000008, Invitrogen, USA) according to the manufacturer's instruction.

## ChIP (chromatin immunoprecipitation) assays

CHIP assays were performed with Simple CHIP Plus Sonication Chromatin IP Kit (56383, CST) with Rabbit monoclonal to CREB1 (9197 S, 1:50, CST). Briefly, dHL60 cells (1×10$^7$ cells) were treated with TNF or CXCL12 for 2 h at 37 °C, when the cells were collected for subsequent steps according to the manufacturer's instructions. CHIP signals were quantified by quantitative PCR analysis. The specific primers pair for the promoter regions were described below: CXCR4, GGGCCTCAGT GTCTCTACTGT (forward), GTTTGAGGGAAGCGGGATGC (reverse). PADI4, ACGGGTTTGTCGTAATGAGC (forward), TGGGACAAGTCTC TCCACCT (reverse).

## RNA sequencing and transcriptomics analysis

Total RNA of CXCR4$^{lo}$ and CXCR4$^{hi}$ neutrophils isolated from healthy controls and psoriasis patients was extracted with TRIzol (15596018CN, Invitrogen), analyzed with an Agilent 2100 Bioanalyzer

(Agilent Technologies), and then quantified using Qubit 2.0 (N12391, Thermo Fisher Scientific). Sequencing libraries were generated and sequenced by GENE DENOVO (Guangzhou, China). Expression profiles of candidate genes were analyzed and visualized using Omicsmart tools. Differentially expressed genes (DEGs) from the RNA-seq data were analyzed using DESeq2 based on the criteria of false discovery rate (FDR) ≤ 0.05. Enrichment analyses for DEGs were conducted using Kyoto Encyclopedia of Genes and Genomes (KEGG) and Gene Ontology (GO) using R packages clusterProfiler, org.Hs.eg.db, enrichplot, and ggplot2. GSEA analysis was performed using GSEA 4.3.2 (http://www.broadinstitute.org/gsea/). The enriched pathways with *P* values were finally visualized with bubble chart and bar chart by the R language package. Correlation heatmaps were analyzed and plotted using the R programming language (version 4.2.3; heatmap package).

## Single-cell RNA-seq analysis

Single-cell dataset was downloaded from https://developmental.cellatlas.io/diseased-skin[26]. All clusters from normal controls and psoriasis were extracted and imported to R environment (v4.0.5) through Scanpy[69] (v1.9.1) and Seurat package[70] (v4.1.0) using pipelines described in https://mojaveazure.github.io/seurat-disk/articles/convert-anndata.html. Then cells were calculated to acquire the percentage of CXCL12$^+$ cells. GraphPad Prism 9.5.0 was used for data visualization.

## Flow cytometry analysis of mouse skin

For the analysis of mouse skin, 1 cm × 1 cm dorsal skin was cut off and transferred to an EP tube containing 1 mL Hank's Balanced Saline Solution (HBSS, H4641, Sigma-Aldrich). The skin was washed rigorously by quickly shaking up and down by hand for 15 s × 3 times. The skin was cut into pieces (<0.5 mm in size) in a 6-well plate placed on ice with dulbecco's modified eagle medium (DMEM, 11885-084, Gibco) (not supplemented with FBS) containing 1 mg/mL Collagenase P (11213857001, Roche, USA) and 0.2 mg/mL DNase I (AMPD1, Sigma-Aldrich). The samples were incubated in a 37 °C cell culture incubator for 60 min and pipetted every 20 min to gently mix the cells. Cell suspensions were filtered through 40 μm cell strainer. Cells were stained with PerCP/Cy5.5 anti-mouse CD45 (103132, 1:100, BioLegend), FITC anti-mouse Ly6G (127606, 1:100, BioLegend), PE anti-mouse CXCR4 (146506, 1:100, BioLegend), and Zombie UV dye (423102, 1:500, BioLegend) in FACS buffer for 30 min and then analyzed by flow cytometry (649225, BD LSRFortessa Cell Analyzer). Data were analyzed with Flowjo v10.8.1 (Tree Star). Background fluorescence levels were determined by Fluorescence Minus One (FMO).

## Analysis of cell viability and apoptosis

Viability assays were performed using the Annexin V-PE/7-AAD apoptosis kit (AP104, MultiSciences Biotech Co., Ltd.) following the manufacturer's instructions. Neutrophils were resuspended in RPMI Medium 1640 (C11875500BT, Gibco, USA) containing 10% FBS at a concentration of 5 × 10$^5$ cells/mL and incubated for 24 h at 37 °C. Before and after incubation, cells were co-stained with FITC anti-human CD15 (301904, 1:100, BioLegend) and PE-Cy7 anti-human CXCR4 (306514, 1:100, BioLegend) for 30 min. After washing, cells were incubated with Annexin V (5 μL) and 7-AAD antibodies (10 μL) in 1 × Annexin V binding buffer for 5 min at room temperature and immediately analyzed by flow cytometry. Four cellular populations were distinguished: viable cells (Annexin V-PE and 7-AAD double-negative), early apoptotic cells (Annexin V-PE positive and 7-AAD negative), late apoptotic cells (Annexin V-PE and 7-AAD double-positive) and necrotic cells/cellular debris (Annexin V-PE negative and 7-AAD positive).

## Cell transfection and dual-luciferase assay

The dHL60 cells were seeded in 6-well plates at a concentration of 5×10$^5$ cells per well and transfected with CREB1 using the lipofectamine

2000 transfection reagent (MF135-01, Mei5 biotechnology) following the manufacturer's protocols. The CXCR4 promoter (cxcr-4-p) construct contained the CREB1 binding site 1 (cxcr-4-p site 1: -150 ~ -138). Sequences for the CXCR4 promoter containing the binding site were designed as follows: mutant type (MT) CXCR4 promoter: 5′-GTGGAAGACGCC-3′. The dual-luciferase reporter assay was carried out using the Dual-luciferase Reporter assay kit (D0010, Solarbio technology). Luciferase activity was measured as the ratio of firefly luciferase signal to Renilla luciferase signal.

## Cell culture and treatment

Human microvascular endothelial cell line (HMEC-1 cell, CRL-3243) was purchased from American Type Culture Collection (ATCC, USA) and cultured as required. Cells were seeded in 6-well plates and cultured until they reached 80% - 90% confluence, followed by co-incubation with human CXCR4[lo] and CXCR4[hi] neutrophils ($2\times10^5$/well), or LDHA inhibitor (LDHA-IN-3, 50 μM, MCE) at 37 °C for 6 h, DMSO alone served as vehicle control. For the expression of CXCL12 in HMEC-1 cells, cells were seeded in 6-well plates and cultured until they reached 60% - 70% confluence, followed by co-incubation with pro inflammatory cytokines (IL17A for 50 ng/ml, IL22 for 20 ng/ml, oncostatin M for 20 ng/ml, TNF 50 ng/ml, and IL1α for 20 ng/ml), healthy serum and psoriatic serum (20%) at 37 °C for 24 h. Cells were washed with PBS and harvested for further experiments.

HL-60 cell lines (CL-0110) were obtained from Procell (Wuhan, China) and cultured as required. Differentiation of HL60 cells into neutrophil-like cells (dHL-60) was induced by culturing in a $CO_2$ incubator for 6 days in the presence of 1.25% DMSO. Morphology analysis with microscopy was used for cell line authentication. All cell lines were tested negative for Mycoplasma contamination.

## ROS detection

For ROS analysis, $5 \times 10^5$ neutrophils were isolated from healthy controls and psoriasis patients and treated with phorbol 12-myristate 13-acetate (PMA, 50 nM, P1585, Sigma-Aldrich) at 37 °C for 30 min, then cells stained with FITC anti-human CD15 (301904, 1:100, BioLegend) and PE-Cy7 anti-human CXCR4 (306514, 1:100, BioLegend), and intracellular ROS production was measured by a dihydroethidium probe kit (BB-47051, DHE, BestBio, Beijing, China) at 37 °C for 30 min. After washing, the cells were immediately analyzed by flow cytometry.

## Degranulation

Degranulation of neutrophils was assessed by monitoring the cell surface expression of CD63. Blood samples obtained from healthy controls and psoriasis patients were collected and red blood cells were removed using Red Blood Lysing Buffer (FXP001, 4 A Biotech Co., Ltd). Then cells were labeled with FITC anti-human CD15 (301904, 1:100, BioLegend), PE-Cy7 anti-human CXCR4 (306514, 1:100, BioLegend) and PE anti-human CD63 (353004, 1:100, BioLegend) at 4 °C for 30 min. After three washes, cells were resuspended in PBS and analyzed using flow cytometry (649225, BD LSRFortessa Cell Analyzer), Background fluorescence levels were determined by Fluorescence Minus One (FMO).

## Wright-Giemsa Staining

Wright-Giemsa staining was performed to analyze the morphology of freshly isolated peripheral CXCR4[lo] and CXCR4[hi] neutrophils obtained from healthy controls and psoriasis patients. The cells were resuspended in an autologous red blood cell suspension with plasma and stained using the Wright-Giemsa (G5637, Sigma-Aldrich) following the manufacturer's instructions. The resulting smears were examined under a microscope to assess the morphological features of the neutrophils.

## Endothelial permeability measurements

Endothelial permeability measurements were performed using a Transwell system with HMEC-1 cells seeded in the upper chamber with a 5.0 μm pore size (CLS3421, Corning, USA) until they reached 90% confluence. Freshly isolated CXCR4[lo] and CXCR4[hi] neutrophils from healthy controls and psoriasis patients ($1\times10^5$/well) were added to the upper chamber and co-incubated at 37 °C for 6 h. LDHA-IN-3 (50 μM, MCE) was added to block lactate activity. Transwell inserts without stimulation or with DMSO were used as controls. After 6 h, the culture medium was removed and washed with free-cell medium three times. Next, 50 μl of FITC-labeled dextran (D1844, 2.5 mg/mL, 40 kDa, Invitrogen) was added to the upper chamber as a tracer. After 2 h, 100 μl samples were collected from the lower chamber and fluorescence spectrophotometry was performed with an excitation wavelength of 494 nm and an emission wavelength of 521 nm (spectrofluorometer, Varioskan LUX 3020-265, Thermo Scientific) to measure the permeability of the endothelial monolayer.

## Skin histopathology

Fixed examples were embedded in paraffin and the 4 μm sections were stained with hematoxylin and eosin (H&E) for histological analysis. Slides were scanned into digital section by slide scanner and analyzed by NDP2 viewer software (HAMAMATSU Photonics). The dermal vascular area in H&E-stained sections was quantified.

## Immunofluorescence staining

For tissue specimens, paraffin-embedded sections (4 μm) were deparaffinized and rehydrated. For cultured cells (HMEC-1 cells) in coverslips, they were fixed with 4% paraformaldehyde for 15 min, and permeabilized with 0.2% Triton X-100 (93443, Sigma-Aldrich) for 10 min. After incubation in goat serum for 1 h at room temperature, the skin sections or the cells were incubated with primary antibodies overnight at 4 °C. The following antibodies were used: Rat monoclonal to Ly6G (sc-53515,1:100, Santa), Rabbit monoclonal to phospho-CREB1 (9198 S, 1:400, CST), Mouse monoclonal to CD15 (241552, 1:100, Abcam), Rabbit monoclonal to CD15 (135377, 1:100, Abcam), Rabbit monoclonal to CXCR4 (181020, 1:100, Abcam), Mouse monoclonal to CXCR4 (60042-1-1 g,1:100, Proteintech, Wuhan, China), Goat polyclonal to CXCR4 (GTX21671, 1:100, GenTex, USA), Mouse monoclonal to CD31 (199012, 1:200, Abcam), Rabbit polyclonal to GPR81 (PA5-114741, 1:100, Invitrogen), Rabbit monoclonal to Vimentin (16700, 1:500, Abcam), Rabbit polyclonal to CXCL12 (17402-1-AP, 1:100, Proteintech), Rabbit polyclonal to ZO-1 (96587, 1:100, Abcam), Rabbit polyclonal to VE-cadherin (33168, 1:100, Abcam), and Rabbit monoclonal to Occludin (216327, 1:100, Abcam). The samples were washed with PBS buffer for three times. The corresponding fluorescent-labeled secondary antibody was further incubated at room temperature for 1 h, then washed with PBS buffer for 10 min × 3 times. After incubation with Hoechst 33258 (C0021, 1:1000, Solarbio technology), the samples were observed with a confocal microscope (LSM880, Carl Zeiss).

For immunofluorescence in neutrophils, cells were fixed, permeabilized and incubated as described above. The following primary antibodies were used: Rabbit monoclonal to phospho-CREB1 (9198 S, 1:400, CST), Rabbit monoclonal to LDHA (3582 S,1:200, CST), Mouse monoclonal to CXCR4 (60042-1-1 g, 1:100, Proteintech), Goat polyclonal to CXCR4 (GTX21671, 1:100, GenTex), Mouse monoclonal to LAMP1 (25630, 1:400, Abcam), Mouse monoclonal to CBP (MA5-13634, 1:500, Thermo Fisher Scientific), Mouse monoclonal to MMP-9 (58803,1:100, Abcam) and Mouse monoclonal to LCN2 (23477, 1:400, Abcam). After incubation with the corresponding secondary antibodies and Hoechst 33258 (C0021, 1:1000, Solarbio technology), the cells were resuspended in 100 μl PBS and placed overnight in coverslips for following confocal microscope (LSM880, Carl Zeiss).

For multiplex fluorescence staining, the staining kit was purchased from four-color multiple fluorescent immunohistochemical staining kit (abs50012, Shanghai, China). Paraffin sections were heated at 80 °C for 10 min and dewaxed in xylene, gradient alcohol dehydration, 10% neutral formalin immersion for 10 min; antigen was repaired

in EDTA solution using microwave repair and cooled to room temperature. Endogenous peroxidase activity was first blocked with 0.3% hydrogen peroxide for 10–15 min, followed by blocking of non-specific sites with 10% goat serum for 30 min. Mouse monoclonal to CD15 (241552, 1:100, Abcam) was incubated overnight as an antibody; the secondary antibody was incubated for 10 min and then incubated with fluorescent dye for 10 min. After the microwave repair was cooled to room temperature, the above steps were repeated again to complete the staining of Mouse monoclonal to CD31 (199012, 1:200, Abcam), Rabbit polyclonal to GPR81 (PA5-114741, 1:100, Invitrogen), Rabbit monoclonal to Vimentin (16700, 1:500, Abcam), Rabbit polyclonal to CXCL12 (17402-1-AP, 1:100, Proteintech), or Rabbit monoclonal to CXCR4 (181020, 1:100, Abcam), and finally incubated with DAPI for 5 min to stain the nucleus and anti-fluorescence quencher seal. The expression was observed under fluorescence microscope (LSM880, Carl Zeiss).

## Visualization and quantification of NETs
CD15$^+$CXCR4$^{lo}$ and CD15$^+$CXCR4$^{hi}$ neutrophils from healthy controls and psoriasis patients were re-suspended in RPMI Medium 1640 (C11875500BT, Gibco) and were seeded on poly-Llysine (P4832, Sigma-Aldrich)-coated coverslips (10$^5$ cells/well). Cells were incubated for 4 h to assess their ability to form NETs at 37 °C. To block p-CREB1 signaling pathway, psoriatic CD15$^+$CXCR4$^{hi}$ neutrophils were treated with DMSO vehicle or 300 μM KG-501 (HY-103299, MCE). Neutrophils were fixed with 4% paraformaldehyde and permeabilized with 0.2% Triton X-100 (93443, Sigma-Aldrich). Cells were blocked with 3% BSA in PBS for 2 h at room temperature, and then cells were incubated with Rabbit polyclonal to citrullinated histone-3 (5103, 1:200, Abcam) and Mouse monoclonal to MPO (25989, 1:100, Abcam) at 4 °C overnight. Secondary antibody conjugated with Goat anti-Mouse IgG cy3 (97035, 1:1000, Abcam) or Goat anti-Rabbit IgG Alexa Fluor 488 (150077, 1:1000, Abcam) were then used. Cells were washed and stained with Hoechst 33258 (C0021, 1:1000, Solarbio technology) and subsequently analyzed by a confocal microscope (LSM880, Carl Zeiss). NETs quantification in immunofluorescence referred to previous study[71] in which the NETs were counted as extracellular citrullinated histone-3-positive cells at least three representative immunofluorescence images (from two neighboring sections) per sample. Then neutrophils were counted as MPO-positive cells at the same images. The percentage of NETs-forming neutrophils was calculated using the formula: (number of NETs-forming neutrophils/number of neutrophils) ×100. All values were determined by two pathologists who were blind to clinical or experimental information.

## Western Blot analysis
Human neutrophils and cultured cells treated with various reagents were washed and lysed in RIPA buffer (P0013C, Beyotime, Shanghai, China). After incubation at 4 °C for 30 min, protein lysates were centrifuged at 12,000 × g for 15 min and supernatants was collected for concentration determination with the BCA Protein Assay Kit (PA115-02, TIANGEN, Beijing, China). The culture supernatants of CXCR4$^{lo}$ and CXCR4$^{hi}$ neutrophils were collected and purified using the ammonium sulfate method. Salt (0.431 g/ml) was added slowly while stirring the supernatant, and precipitation was performed at 4 °C overnight. The protein precipitates were obtained by centrifugation at 13,000 g for 15 min at 4 °C, dissolved in PBS, and dialyzed in deionized water. The corresponding CXCR4$^{lo}$ and CXCR4$^{hi}$ neutrophils were also extracted for the detection of GAPDH.

Briefly, equivalent amounts of protein were separated on 10% SDS–PAGE and transferred to PVDF membranes. Then the membranes were blocked with blocking buffer for 1 h, and incubated with primary antibody: Rabbit polyclonal to ZO-1 (96587, 1:1000, Abcam), Rabbit polyclonal to VE-cadherin (33168, 1:1000, Abcam), Rabbit monoclonal to Occludin (216327, 1:1000, Abcam), Mouse monoclonal to GAPDH

(60004-1-Ig, 1:5000, Proteintch), Mouse monoclonal to PADI4 (128086, 1:1000, Abcam), Rabbit polyclonal to citrullinated histone-3 (5103, 1:1000, Abcam), Rabbit monoclonal to CREB1 (9197 S, 1:1000, CST), Goat polyclonal to GPR81 (106942, 1:500, Abcam), Rabbit monoclonal to phospho-CREB1 (9198 S, 1:400, CST), Mouse monoclonal to CBP (MA5-13634, 1:1000, Thermo Fisher Scientific), Mouse monoclonal to MMP-9 (58803,1:100, Abcam) and Mouse monoclonal to CXCR4 (60042-1-Ig, 1:1000, Proteintech) at 4 °C overnight. Then membranes were incubated with secondary antibody conjugated with anti-rabbit or anti-mouse horseradish peroxidase, for 1 h at room temperature. Blots were detected using an enhanced chemiluminescence detection kit (GTX14698, GeneTex). Intensities of the bands were quantified by Image Lab (Bio-Rad Laboratories, Inc., version 5.2.1). The uncropped and unprocessed scans were supplied in the Source Data file and Supplementary Information.

## Cell adhesion assay
HMEC-1 cells were seeded in a 15-mm glass bottom cell culture dish (801002, NEST, Wuxi, China) and cultured until they reached 80% confluence. Freshly isolated CXCR4$^{lo}$ and CXCR4$^{hi}$ neutrophils (10$^5$ cells/well) from healthy controls and patients with psoriasis were then added into the culture dish. Cells were incubated at 37 °C for 2 h and then incubated with Hoechst 33258 (C0021, 1:1000, Solarbio technology) for 15 min in dark. After washing twice with PBS and subsequently analyzed by a confocal microscope (LSM880, Carl Zeiss).

## Enzyme-linked immunosorbent assay (ELISA)
CXCL12 (E-EL-H0052c, ElabScience, Wuhan, China), IL17A (E-EL-H5812c, ElabScience), MPO (ab119605, Abcam), and MMP-9 (E-EL-H6075, ElabScience) levels in blood plasma samples were measured using ELISA kits according to the manufacturer's instructions. In brief, samples or standards were added to the wells and incubated for 90 min at room temperature. After washing, a working detector was added to each well, followed by the addition of the substrate solution. The reaction was stopped, and the absorbance was read at 450 nm. The amount was calculated using a standard curve and GraphPad Prism 9.5.0.

## RNA isolation and quantitative RT-PCR
RNA was extracted by standard procedure with TRIzol reagent (15596018CN, Invitrogen). The RNA concentrations and purity were measured spectrophotometrically (N12391, Thermo Fisher Scientific, Inc.) and qRT-PCR was performed with SYBR Green Master Mix (RR820A, TaKaRa, Japan) in 384-well plates according to the manufacturer's instructions. Data analysis of mRNA expression was normalized to the internal control β-actin and quantified by the $2^{-\Delta\Delta Ct}$ method. All primers used for RT-PCR are listed in Supplementary Table 2.

## Statistics and Reproducibility
Statistical analyses were performed using GraphPad Prism 9.5.0 (GraphPad Software, Inc., USA) and R software (R Statistical Software, version 4.2.3). Experimental data were analyzed using two-tailed paired and unpaired Student t test, one-way or two-way ANOVA. Two-sided Tukey's multiple comparison test was used for multiple comparisons. Differentially expressed genes (DEGs) from the RNA-seq analyses were identified using the R "DEseq2" package with a threshold of |logFC | >1 and false-discovery rate (FDR) < 0.05. The Gene Ontology (GO) and Kyoto Encyclopedia of Genes and Genomes (KEGG) enrichment analyses were carried out using R "clusterProfiler", "enrichplot", and "ggplot2" packages. Bar chart, bubble chart, and correlation heatmaps were mapped using R programming language. $P < 0.05$ considered statistically significant. All in vitro and in vivo experiments were repeated at least twice independently with similar results.

**Reporting summary**

Further information on research design is available in the Nature Portfolio Reporting Summary linked to this article.

## Data availability

The raw sequence data generated in this study has been deposited in the Genome Sequence Archive in National Genomics Data Center, Beijing Institute of Genomics (BIG), Chinese Academy of Sciences under accession number HRA005230. The human publicly available data used in this study are available in the 10X genomics database [https://developmental.cellatlas.io/diseased-skin]. The remaining data are available within the Article, Supplementary Information or from the corresponding author upon request. Source data are provided with this paper.

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

## Acknowledgements

This study was supported by the National Natural Science Foundation of China (Grant No. 82230105 to G.W., 82003339 to J.L.C., and 82273520 to S.S.) and Shaanxi scientific research grant (Grant No. 2022ZDLSF03-14 to G.W.).

## Author contributions

J.L.C., Y.X.B. and K.X. contributed to the study design, data collection, and paper preparation. Z.G.L., Z.L.Z., C.Y., B.L. and Q.Y.L. performed scRNA-seq analysis and animal studies. P.Q., C.X.L., Y.X.L., S.X.S. and H.J.Q. contributed to the patient sample collection, in vitro experiments, and formal analysis. E.L.D. and W.Y. revised the article. S.S., G.W. and J.E.G. developed the concept, supervised the research, and critically revised the paper. All authors read and approved the final manuscript.

## Competing interests

The authors declare no competing interests.
