## [Peer Review File · Nature Communications]

CREB1-driven CXCR4hi neutrophils promote skin inflammation in mouse models and human patientsREVIEWER COMMENTS

Reviewer #1 (Remarks to the Author):

Heterogeneity of neutrophils, and the separation of neutrophils in distinct subsets, is a fascinating new concept in innate immunity although as of yet there is lack of clear consensus re how to properly identify and define different subsets. Multiple different 'types' of neutrophils have been described in recent years, but as far as I am aware only two truly distinct subtype markers (that are either on or off independent of maturity/maturation stage), CD177 and OLFM4. In contrast to such bona fide subset markers, CXCR4 is a marker that may be induced in all (?) neutrophils. The manuscript describes the abundance of CXCR5hi neutrophils in psoriasis and claim that these cells are equipped with some peculiar features (e.g., hyperactivated glycolysis) that drive skin inflammation in this disease. The underlying observation, that psoriasis patients have an abundance of non-standard neutrophils CXCR4hi, but also high in a variety of additional markers) in circulation, is exciting. However, the presented data fail to convince that a) CXCR4 per se is of importance or even a suitable marker, and b) that the CXCR4hi cells are truly distinct functionally.

Specific comments:

1) The isolation procedure on which much data rely is problematic and insufficiently described (as is unfortunately a lot of the experimental details). Flow cytometry data on the isolated (by CXCR4-magnetic beads) subtypes is completely lacking, as is information on purity of isolated cells. Are the isolated CXCR4hi cells used for functional analyses still bound to the antibody-magnetic bead? If so, this fact alone could well explain at least some of the increased activation seen in this subtype. Antibody-bead isolation is likely perfectly fine for analyzing cell content, but not for functional studies of viable cells. More specifically:

- Is the cut-off (re CXCR4) consistent with the gates shown in other data sets (e.g., Fig. 1)?
- Are there any non-neutrophil contaminants (enriched in either of the fractions)?
- Why was this isolation method not used for the bulk RNAseq (explained in suppl fig. 3)?

2) CXCR4 cannot be key to the various effects seen by psoriasis CXCR4hi neutrophils. The pronounced difference in effect by Pso CXCR4hi and HC CXCR4hi cells clearly demonstrates that there is something else going on in the former. For example, >10x more genes are

upregulated in CXCR4^{hi} vs CXCR4^{lo} neutrophils from psoriasis patients, as compared to the situation in HC. There are multiple markers (and effects) shown that does not follow CXCR4 expression. Focusing on CXCR4 as the pivotal player is misleading since, according to the data presented, the subset might as well be described as CREB1^{hi}, CD101^{hi}, or CD10^{hi}, especially the latter seems almost an ON/OFF marker (suppl fig. 2). Especially for CD10, an isotype control should be shown –are the CXCR4^{lo} samples (suppl fig 2a) in fact CD10-? If so, they might be regarded as immature, making the situation even more complex to decipher (since they would be overrepresented in HC blood).

3) CXCR4 is a chemokine receptor, heavily expressed by various lymphocytes, but also on the surface of some, presumably aged and/or activated neutrophils. Most neutrophil receptors, the surface expression of which can be substantially increased, are stored within mobilizable granules and does not require de novo transcription/protein synthesis in order to be upregulated. I lack information on the mechanism underlying CXCR4 upregulation:

- is it stored inside granules in CXCR4^{lo} cells?
- Is de novo protein synthesis required?
- What is the kinetic of CXCR4 upregulation (the shortest time shown is 2 hrs) and would is this limited time sufficient for the gene and protein to be expressed?

4) Although mentioned in the introduction, the possible impact of LDNs to the results is neither addressed nor discussed. LDNs (it is not unlikely that they are rather abundant in psoriasis patients) would be present in all experiments using whole blood, but not in the experiments using isolated neutrophils. This could be of significant importance.

5) The authors previously described neutrophil MMP-9 to the culprit behind cutaneous vasodilation and hyperpermeability in psoriasis (PMID: 32888954). In the current manuscript they instead state that lactate is critical. These two apparently disparate findings needs to be reconciled or at least discussed.

Minor comments:

- The use of pharmacological inhibitors is risky without all proper controls to demonstrate that their actions are indeed specific. Especially so re in vivo experiments where active

doses of inhibitors are hard to predict/specify. There are no control experiments shown (or mentioned) that e.g., KG-501 does not inhibit non-CREB1 mediated responses.

- Why is CXCR4 not internalized (resulting in lower surface expression) in cells treated with CXCL12?
- Nuclear/cellular morphology and NETs results would be much strengthened by proper quantification
- How were murine neutrophils isolated from blood?
- Viability assays (Fig. 2C) would be much more informative if an early cell death/apoptosis marker was also included.
- Statistical calculations based on n=3 are not meaningful
- The IMQ model of psoriasis was shown to be rather completely dependent on IL17/il23 (PMID: 19380832). In order to convincingly claim that CXCR4/CXCL12 is as critical, ko mice should be used.
- Circadian rhythm have been earlier shown to affect CXCR4 expression, i.e., aging in the blood. Were all blood samples collected at a similar time of the day? When?

Reviewer #2 (Remarks to the Author):

Chen, Bai, Zhu et al describe proinflammatory CXCR4+ neutrophils in psoriasis as relevant immune cells orchestrating skin inflammation. They use state-of-the-art techniques to address a relevant question in a cutaneous inflammation and use elegant experiments to show that neutrophils up regulate CXCR4 in response to proinflammatory stimuli, are recruited to skin via CXCL12 produced by endothelial cells and induce vascular damage by lactate. They furthermore describe CREB1 as transcription factor relevant for their pro inflammatory phenotype. The relevance of their data from psoriasis patients is corroborated with a mouse model for acute psoriasiform inflammation, where inhibition of neutrophils, CXCR4 or its ligand reduced inflammation.

While these observations are of potential interest, the following points need to be addressed in a revised manuscript:

- 1) Patient details should be described in more detail and, given low patient number, be presented for the individual figures as supplements, including age, comorbidities, BMI, and disease duration. In general, patient numbers need to be added to the figure legends for the

individual figures.

2) In Figure 3, patient numbers for sequencing are too low (n=3 and 4 for healthy controls and psoriasis patients, respectively). Also, it should be mentioned if patients with short disease duration have another signature compared to chronic psoriatic patients. It could be that the reported information on neutrophils is especially important in acute inflammation and less so in patients with a long disease duration.

3) Please show representative FACS dot plots for Fig. 3e.

4) While it is nicely shown that CXCR4+ neutrophils modulate vascular permeability in HMEC-1 cells via the lactate-GPR81 axis, it's relevance in psoriatic skin lesions remains to be assessed. Is GPR81 expressed by endothelial cells in psoriatic skin lesions with CXCR4+ neutrophils being in close contact?

5) While the representative picture in Fig 5f shows some p-CREB1/CD15 double positive cells, it is hard to make the conclusions from a single picture. Please quantify p-CREB1/CD15 double positive cells in psoriasis and show co-localization of pCREB1 and CXCR4 in skin lesions similar to blood neutrophils

Reviewer #3 (Remarks to the Author):

CXCR4 is known as a key modulator of neutrophil trafficking between tissues, peripheral blood, and the bone marrow and a subset of neutrophils with high CXCR4 surface expression was recently shown to play an important role in inflammation-related lung injury. In the current study, Chen and colleagues demonstrate a pro-inflammatory role of CXCR4-expressing neutrophils in psoriatic skin inflammation and investigate the underlying mechanisms that drive this pathological neutrophil phenotype and that promote CXCR4 expression. The manuscript contains interesting and novel data, e.g. along the connection between CXCR4 expression and the transcription factor CREB1, and is a valid improvement over current knowledge, strengthening our insight into the plasticity of neutrophils during inflammation.

I have some specific questions and comments that could further improve the manuscript:

- The figure legends and the text in general need more experimental details to understand

what is exactly shown and how it could be interpreted. For example, group sizes (n per group) should always be stated. A few specific examples of missing information to follow:

Fig. 1G: How many samples were tested?

Fig. 2B: What's the color scale? Z-score?

How was NET formation assessed in Fig. 2E?

Fig. 2H: How many samples were tested?

Fig. 3A: Should the heading read CXCR4^{hi} vs CXCR4^{lo}?

Fig. 3B: Are the top most significant genes shown in the heatmap or an arbitrary selection of genes?

Fig. 3C: Can you show normalized enrichment scores for differentially regulated pathways, in order to see the direction of the effects?

- Suppl. Fig. 1C: The authors claim that CD31⁺ vascular endothelial cells are the main source of the CXCR4 ligand CXCL12 in inflamed skin. In Suppl. Fig. 1C it looks like basically all CXCL12-expressing cells are CD31 positive. This is a bit surprising, because CXCL12 expression was also reported in other skin resident cell types such as keratinocytes and dermal fibroblasts. Can the authors quantify how many % of CXCL12⁺ cells are CD31⁺ and how many independent skin samples they actually investigated?

- The upregulation of surface CXCR4 after CXCL12 exposure is also a bit puzzling, since CXCR4, like other chemokine receptors, is internalized following ligand binding and receptor activation. Or isn't it? Can the authors please clarify and discuss?

- As far as I understood only spontaneous ROS production and NET formation was assessed in the manuscript, which is known to correspond to neutrophil activation. Were the NETs in Suppl. Fig. 2B quantified? Have the authors also looked into induced ROS/NET formation by e.g. PMA or calcium ionophore?

- Chronic inflammation and collateral tissue damage is often considered the dark side of immune function essential for defence against pathogens. In line with that, a subpopulation of CXCR4^{HI} neutrophils constitute the first line of defence against bacteria by rapidly migrating towards invading pathogens and being able to more efficiently phagocytose them

(PMID 27609642). However, in the current manuscript the authors show that CXCR4HI neutrophils have a reduced ability to phagocytose. How can these data be reconciled?

- The neutrophil survival time in culture seems pretty long (Fig. 2C). After 24h roughly 90% of neutrophils are still intact. The authors state themselves that neutrophils have a short lifetime and survive less than 24h in vivo in the circulation.

Minor comments:

There are a number of typos and small flaws in the text and the figures, e.g. „amplify“ in the abstract (line 33), „slightly“ (line 239), „subcutaneously“ (line 257), „occludin“ throughout Fig. 4, „MFI“ and „MIF“ being used for mean fluorescence intensity etc.

Point by point responses to the Reviewers' comments

Response to Reviewers

We thank the reviewers for these thorough and constructive comments, which have enabled us to greatly improve our manuscript. As described below, we have now addressed each of the critiques and suggestions from the reviewer. Reviewer's comments are in black font, and our point-by-point responses to these comments are shown in blue.

REVIEWER COMMENTS

Reviewer #1 (Remarks to the Author):

Heterogeneity of neutrophils, and the separation of neutrophils in distinct subsets, is a fascinating new concept in innate immunity although as of yet there is lack of clear consensus re how to properly identify and define different subsets. Multiple different 'types' of neutrophils have been described in recent years, but as far as I am aware only two truly distinct subtype markers (that are either on or off independent of maturity/maturation stage), CD177 and OLFM4. In contrast to such bona fide subset markers, CXCR4 is a marker that may be induced in all (?) neutrophils. The manuscript describes the abundance of CXCR5hi neutrophils in psoriasis and claim that these cells are equipped with some peculiar features (e.g., hyperactivated glycolysis) that drive skin inflammation in this disease. The underlying observation, that psoriasis patients have an abundance of non-standard neutrophils CXCR4hi, but also high in a variety of additional markers) in circulation, is exciting. However, the presented data fail to convince that a) CXCR4 per se is of importance or even a suitable marker, and b) that the CXCR4hi cells are truly distinct functionally.

Response: Thank you so much for these insightful comments. Following efforts are made to address the mentioned issues.

(1) We agree that CD177 and OLFM4 represent distinct neutrophil subtype markers as previously reported ^{1, 2}, and we also introduced this briefly in the Introduction section of our manuscript. In our study, we previously assessed the frequency of OLFM4⁺ or CD177⁺ neutrophils in healthy controls and psoriasis patients and observed no significant difference between the two groups (Supplementary Fig. 16). Therefore, we did not further explore the functions of CD177⁺ or OLFM4⁺ neutrophils in psoriasis. However, we have observed prominent OLFM4⁺ neutrophils in skin lesions of generalized pustular psoriasis ³ which is another subtype of psoriasis with massive neutrophil infiltration, indicating a possible role of OLFM4⁺ neutrophils in pustular psoriasis, but this is currently under further investigation by our group.

The specific granule glycoprotein olfactomedin-4 (OLFM4) marks a subset (1-70%) of human neutrophils and OLFM4^{hi} proportion has been shown to correlate with septic shock severity ⁴, although OLFM4 presence is not regulated at the level of transcription ¹. A higher frequency of circulating CD177⁺ neutrophils has been reported in patients with bacterial infections ^{2, 5} and a variety of autoimmune diseases including SLE ^{6, 7}. CD177⁺ neutrophils have been shown to play a critical role in inflammation by releasing IL-22 and neutrophil extracellular traps (NETs) ^{8, 9}. However, it has also been shown that there are no morphological differences in nuclear conformation or granules between CD177⁺ and CD177⁻ neutrophils ⁸. Therefore, the immune regulatory roles of CD177⁺ or OLFM4⁺ neutrophils warrant further investigation.

Revised Supplementary Fig. 16. The frequency of CD177⁺ and OLFM4⁺ neutrophils in psoriasis patients and healthy controls. (a-b) Gating strategy of CD177 or OLFM4 fluorescence intensity on peripheral neutrophils (left) and the proportions of peripheral CD177⁺ (a) or OLFM4⁺ (b) neutrophils (right) from healthy controls and psoriasis patients. Whole blood was used. A point represents a sample. Analyses: unpaired Student’s t-test. ns, not significant. HC, healthy control; Pso, psoriasis patients.

(2) As to the “types” of neutrophils that also asked in Q3 below, although neutrophils are conspicuously poor in RNA content, application of high-dimensional single-cell transcriptomic approaches to neutrophils has provided insights into their transcriptional heterogeneity during maturation in health and during inflammatory responses or malignant progression, such as TGFβ1⁺CCR5⁺ neutrophil, Fth1^{hi} neutrophils, and various tumor-associated neutrophil (TAN) populations ^{10, 11, 12, 13}. These findings demonstrate that neutrophils in circulation exhibit a heterogeneous mixture of cells with diverse phenotypic and functional states ¹⁴. Another single-cell-based study on mouse neutrophils, defined 8 neutrophil populations, each with a distinct molecular signature, demonstrating that neutrophils gradually acquire microbicidal capability as they traverse the transcriptional landscape, representing conserved mechanisms for fine-tuning regulation of neutrophil response, driven by both known and yet uncharacterized transcription factors. Notably, CXCR4 is noted as a marker gene for one of the

neutrophil clusters in this study¹⁰. This is consistent with our findings shown here on CXCR4^{hi} neutrophils in psoriasis. CD177, as shown in this recent study¹⁰, is expressed in G2-G4 murine neutrophils, whereas CXCR4 is expressed in G5 neutrophils, indicating that CD177⁺ neutrophils may represent a more immature neutrophil subset. The possible relationship between CD177⁺ and CXCR4⁺ neutrophils will need to be addressed in future studies.

Several previously published studies have reported on the role of CXCR4^{hi} neutrophils in inflammation including allergic asthma^{15,16} and Alzheimer's disease¹⁷. For instance, exposure to a low dose of LPS skews neutrophils to upregulate CXCR4 in the lung, which then release NETs to mediate allergic asthma¹⁵. Toward a more comprehensive, mechanistic understanding of CXCR4^{hi} neutrophils, we sought to characterize the phenotypic and functional properties of CXCR4^{hi} and CXCR4^{lo} neutrophils in psoriasis patients and healthy controls. We have revealed the functionally distinct neutrophil subpopulations using patient samples and *in vivo* mouse models, RNA-seq analysis, and neutrophil-like cell lines such as differentiated HL60, etc. In brief, CXCR4^{hi} and CXCR4^{lo} neutrophils showed differences in lifespan, ROS production, NETs formation, phagocytic ability, degranulation, expression of activation markers and proinflammatory genes, and enhanced glycolysis (revised Fig. 2-5). We also identified that the crucial transcription factor determining CXCR4^{hi} neutrophils was CREB1 (revised Fig. 6). Collectively, our observations suggest that CXCR4^{hi} neutrophils represent a pro-inflammatory population in the initiation and progression of psoriasis pathology. Strategies targeting the CXCR4^{hi} neutrophil-CREB1-NETs pathway may effectively inhibit skin inflammation and provide new ideas for treatments.

Moreover, neutrophils may adopt distinct phenotypic and functional properties across different tissues¹⁸. These functions enable, for example, support of B cell maturation in the spleen or vascular regeneration in the lung^{18,19}. One recent study revealed that immunotherapy for cancers expands a distinct neutrophil state with an IFN-stimulated gene signature²⁰, highlighting the dynamic nature of neutrophil phenotypes. Therefore, we further show particularly high responsiveness of CXCR4^{hi} neutrophils toward various inflammatory stimuli, including TNF- α , IL-25, and CXCL12 (revised Fig. 5). It is logical to predict that this neutrophil heterogeneity emanates from local, rapid reprogramming of neutrophils under tissue-derived signals. The CXCL12/CXCR4 axis has been extensively studied in various contexts. In the lung, CXCL12-producing vessels play a critical role in the retention of neutrophils in specific perivascular areas, where they undergo reprogramming to support vascular growth¹⁸. In addition, CXCL12 signals act as classical niche factors that induce chromatin remodeling, promoting

nuclear compaction and facilitating the navigation of neutrophils in complex three-dimensional environments ²¹. However, the parameters influencing neutrophil heterogeneity warrant further investigation.

A major challenge in studying neutrophil subsets is the lack of genetic and molecular tracers to accurately distinguish between heterogeneous populations beyond surface markers ²². In contrast to T cell subsets, which can be categorized based on their origin, transcriptional profile, and genetic drivers ²³, neutrophil heterogeneity remains poorly understood due to a lack of molecular resolution, and the absence of reliable markers makes it difficult to formally classify them into distinct subsets ²⁴. Additionally, these states exist as a transcriptional continuum rather than independent clusters, further complicating their classification. As discussed recently ²², novel barcoding tools coupled with single-cell RNA sequencing or genetic studies *in vitro* and *in vivo* may be required to fully explore the global architecture of the neutrophil compartment, the specific tissue niches that enable functional reprogramming, and potential clinical applications. Therefore, we have substantially expanded on our discussion of neutrophil phenotypes and functions in the Discussion section and revised the Introduction section. We feel that these additions have substantially strengthened our manuscript.

(3) To fully address these comments below, we have performed a series of additional experiments including imaging flow cytometry, a series of experiments on CXCR4 expression kinetics and internalization, additional RNA-sequencing with an increased number of patient samples, mouse experiments, and additional flow cytometry experiments, etc.. We have collected more psoriasis patient samples from the clinic to increase the number of participants (at least n = 6-8) for all the functional studies and RNA-seq analysis. Increased sample size for increased power and validation in data is shown and detailed in corresponding Figure legends.

Specific comments:

1) The isolation procedure on which much data rely is problematic and insufficiently described (as is unfortunately a lot of the experimental details). Flow cytometry data on the isolated (by CXCR4-magnetic beads) subtypes is completely lacking, as is information on purity of isolated cells. Are the isolated CXCR4^{hi} cells used for functional analyses still bound to the antibody-magnetic bead? If so, this fact alone could well explain at least some of the increased activation seen in this subtype. Antibody-bead isolation is likely perfectly fine for analyzing cell content, but not for functional studies of viable cells.

Response: We thank the reviewer for pointing this out. This has helped us to further improve the quality of our manuscript. As suggested, we have now detailed all experimental procedures in the Method section and/or Supplementary Methods, including showing representative gating strategies for all FCM experiments (revised Supplementary Fig. 1a, 1b, 1e, 2a, 2c, 3d, 4e, 4f, 7a, 7b, 9a, 9b, 10a, 12a, 13f, 16a, 16b, 17a, and 17b), and adding details regarding isotype controls, neutrophil isolation by magnetic beads, and the purity of isolated cells (revised Supplementary Fig. 17).

Considering that time from venipuncture to cell isolation may impact neutrophil-related experiment results ²⁵, such as neutrophil density ²⁶ and CXCR4 expression ²¹, we employed a commonly used magnetic cell separation method (nanobeads 20-100nm; Miltenyi Biotec), to isolate CXCR4^{hi} neutrophils for the *ex vivo* functional studies. Magnetic bead-based cell separation is gentle cell isolation and is commonly used in the field of immunology and for the separation of neutrophil subpopulations, including CD177⁺ neutrophils ^{8, 27}, PR3⁺, or NB1⁺ neutrophils ²⁸. Using this approach, we have isolated neutrophils within 1.5 hours from blood collection to prevent artificial activation and apoptosis, in contrast to flow-based separation methods which take over 3 hours to collect enough CXCR4^{hi} neutrophils (1×10^7 cells) from a single blood sample, for functional studies. As detailed and underlined in the Method section, the antibody-bead was only bound to cells for 15 min for isolation and immediately washed off, we consider it unlikely to affect the function of the isolated neutrophils, as shown in multiple other studies.

To fully address this question, we have also isolated psoriatic CXCR4^{hi} neutrophils using magnetic separation (n = 4) and compared them against neutrophils isolated by flow separation (n = 6) using RNA sequencing. Compared to psoriatic CXCR4^{lo} neutrophils, psoriatic CXCR4^{hi} neutrophils highly expressed 1534 genes (via magnetic sorting) or 1699 genes (via flow isolation). And the differentially expressed genes (DEGs) between psoriatic CXCR4^{hi} neutrophils via two methods were only 108 upregulated and 217 downregulated (Figure R1a). Those DEGs in two groups demonstrated no significant differences, including changes in biological processes or cellular pathways (Figure R1a-f). Furthermore, similar results were obtained for expression of activation markers and inflammatory genes as detected by qRT-PCR (Figure R1g), demonstrating that there is no significant difference between magnetic bead and flow-based isolation methods.

Figure R1. The comparison of neutrophil properties isolated via magnetic sorting and flow isolation method. (a) Bar chart showing the number of differentially expressed genes (DEGs) in the psoriatic CXCR4^{hi} vs. CXCR4^{lo} neutrophils via magnetic sorting (n = 4) and flow isolation method (n = 6). (b) Heat map of DEGs in the psoriatic CXCR4^{hi} vs. CXCR4^{lo} neutrophils by two sorting methods. (c-f) KEGG and GO enrichment analysis of the main difference pathways enriched by magnetic sorting (c, e) and flow isolation (d, f). (g) Expression levels of inflammatory molecules, metabolic molecules, and transcription factors in isolated neutrophils were compared by qRT-PCR for both sorting methods. Mean \pm SD (n = 6 biologically independent samples/group). Analyses: two-way ANOVA with Tukey's post hoc test. GO, Gene ontology; KEGG, Kyoto Encyclopedia of Genes and Genomes; Pso, psoriasis patients.

Moreover, as illustrated on the website of Miltenyi biotec, specific MACS MicroBeads are biodegradable and non-toxic and available for the positive selection of numerous cell types and cell subsets. Due to the low labeling concentrations and their small size, MACS MicroBeads do not lead to the activation of the target cells. They are conjugated to highly specific antibodies against a certain cell surface antigen. And magnetic beads connected through anti-APC can naturally fall off without activating the target cells. Importantly, scanning electron microscopy showed no visible labeling on the cell surface or alteration of the cells' appearance after isolation with MACS MicroBeads and MACS Columns. One recent study compared the influences of isolation technology on neutrophil phenotype and function, which recommended using immunomagnetic separation of neutrophils for studying neutrophil polarization, phagocytosis, ROS production, degranulation, and NETosis²⁹. Other studies also proved that this positive magnetic selection would not influence the functions of neutrophils and T cells^{30, 31}, for instance, the polarization of neutrophils or the oxidative burst in neutrophils.

2) More specifically:

- Is the cut-off (re CXCR4) consistent with the gates shown in other data sets (e.g., Fig. 1)?
- Are there any non-neutrophil contaminants (enriched in either of the fractions)? (The importance of being “pure” neutrophils)
- Why was this isolation method not used for the bulk RNAseq (explained in suppl fig. 3)

Response: Thank you for these comments and suggestions. As suggested, we have now detailed all the experimental procedures including representative gating strategies for all FCM experiments (revised Supplementary Fig. 1a, 1b, 1e, 2a, 2c, 3d, 4e, 4f, 7a, 7b, 9a, 9b, 10a, 12a, 13f, 16a, 16b, 17a, and 17b), and now show representative data of isolation efficiencies in the Method section and/or Supplementary Figures (revised Supplementary Fig.17).

(1) As shown in Supplementary Figures, the cut-off (re CXCR4) was consistent with the gating thresholds shown in other data for the same settings (e.g., Fig. 1), and we have added a detailed description of this in the Method section and corresponding figure legends in our revised manuscript.

(2) Regarding the potential of non-neutrophil contaminants, we set the purity of isolated neutrophils, including CXCR4^{hi} neutrophils, at >90% (n = 6, revised Supplementary Fig. 17). However, we cannot completely exclude the possibility that occasional, small,

non-autofluorescent monocytes/macrophages were included among the neutrophils. We have added this as one limitation in the Discussion section with changes underlined.

(3) For the bulk RNA sequencing of CXCR4^{hi} and CXCR4^{lo} neutrophils (2×10^6 cells), flow sorting was used to ensure the purity of CXCR4^{hi} neutrophils, which took within 30 min for each sample (2×10^6 cells). However, considering that time from venipuncture to cell isolation may impact neutrophil-related experiment results ²⁵, such as neutrophil density ²⁶ and CXCR4 expression ²¹, we employed a commonly used magnetic cell separation method to isolate CXCR4^{hi} neutrophils for the *ex vivo* functional studies. As discussed in our response to Q1 above, the additional RNA-seq analysis and qRT-PCR results show that the magnetic bead isolation for functional experiments reproduced the similar findings as from the flow cytometry-based method (Figure R1). Moreover, we have also collected more blood samples for isolating CXCR4^{hi} and CXCR4^{lo} neutrophils for RNA-seq analysis, and replaced the graphs shown previously in Fig. 3.

3) CXCR4 cannot be key to the various effects seen by psoriasis CXCR4^{hi} neutrophils. The pronounced difference in effect by Pso CXCR4^{hi} and HC CXCR4^{hi} cells clearly demonstrates that there is something else going on in the former. For example, >10x more genes are upregulated in CXCR4^{hi} vs CXCR4^{lo} neutrophils from psoriasis patients, as compared to the situation in HC. There are multiple markers (and effects) shown that does not follow CXCR4 expression. Focusing on CXCR4 as the pivotal player is misleading since, according to the data presented, the subset might as well be described as CREB1^{hi}, CD101^{hi}, or CD10^{hi}, especially the latter seems almost an ON/OFF marker (suppl fig. 2). Especially for CD10, an isotype control should be shown –are the CXCR4^{lo} samples (suppl fig. 2a) in fact CD10-? If so, they might be regarded as immature, making the situation even more complex to decipher (since they would be overrepresented in HC blood).

Response: We appreciate this question and will attempt to address this as best as we can. We have substantially revised the Figures and Results in our updated manuscript, and we have also added to our Discussion section details regarding the functions and upstream regulation of CXCR4^{hi} neutrophils.

(1) As we have discussed above, rapidly growing and emerging evidence has demonstrated that neutrophils exhibit heterogeneous phenotypes under certain physiologic and pathologic conditions ^{32, 33}. To date, a wide range of neutrophil phenotypes have been described. This includes normal-density neutrophils (NDNs),

low-density neutrophils (LDNs), N1/N2, reverse-migrated neutrophils, aged neutrophils, CD177⁺neutrophils, OLFM4⁺neutrophils, CD62L^{dim} neutrophils ³⁴, and TGFβ1⁺CCR5⁺ neutrophil ^{13, 35}, amongst others. These neutrophil subtypes likely reflect differences in density, surface markers, and maturity states of neutrophils, but consensus criteria are still lacking. In particular, the diversity of plasticity of neutrophils appears even greater in inflammation and cancers as shown in recently published high-resolution single-cell atlases ^{36, 37, 38}. These illustrate the notion that, already in circulation, neutrophils are a mixture of cells in different phenotypic and functional states ¹⁴. Moreover, neutrophils may adopt distinct phenotypic and functional properties across different tissues ¹⁸. Therefore, understanding the mechanisms by which microenvironmental factors regulate neutrophils to promote their pro-inflammatory functions is crucial, and may provide opportunities for more specific targeting of inflammation-associated neutrophils.

CXCR4⁺neutrophils have been identified in several previous studies. For example, “aged” neutrophils upregulate CXCR4 and represent an overly active subset exhibiting enhanced αMβ2 integrin activation and NETs formation under inflammatory conditions, which is driven by the microbiota via Toll-like receptor ³⁹. Single-cell studies from mouse lungs, that a low dose of LPS upregulates their expression of CXCR4, and CXCR4^{hi} neutrophils release NETs to mediate allergic asthma, indicating that environmental risk factors shape recruited neutrophils in the lung to promote the initiation of allergic asthma ¹⁵. Our data are consistent with these findings and expand on our understanding of the role of CXCR4^{hi} neutrophils in skin inflammation, and further suggest that the activation responses of CXCR4^{hi} neutrophils not only include ROS formation and NETs formation, but also changes in surface receptor compositions (e.g., CD10, CD44, CXCR2), intracellular signaling cascades (e.g., CREB1, glycolysis-associated genes), and pro-inflammatory effects on cutaneous vessels via releasing lactate and MMP-9 (revised Fig. 2, 3). We consider CXCR4 to be a marker of neutrophil priming to additional stimuli - as this is the marker that best distinguishes those cells.

(2) The differences between psoriatic (Pso) CXCR4^{hi} vs. healthy control (HC) CXCR4^{hi} cells are not unexpected and are similar to what has been shown for other neutrophils subsets, such as CD177⁺ neutrophils, where significant changes associated with defense response and inflammatory immune responses were seen in this neutrophil subset between HC and inflammatory bowel disease patients⁸. Similarly, in patients with biliary atresia and patients with intrahepatic cholestasis, CD177⁺ neutrophils showed increased and decreased expression of 4654 and 2590 genes respectively compared to HC CD177⁺ neutrophils ⁹. Lastly, OLFM4⁺ neutrophil subsets also demonstrated

marked differences in proteomic expression between healthy controls and septic patients ⁴. Therefore, the immune microenvironment will influence and educate neutrophil phenotypes, as we observed in psoriatic skin (revised Fig. 5). Our data support the notion that neutrophils are heterogeneous, plastic, and adaptable to context-specific cues to exert particular functions ⁴⁰.

(3) As suggested, we have added an isotype control for the staining of CD10 and other activation markers (revised Supplementary Fig3), which showed that psoriatic CXCR4^{hi} neutrophils have a higher surface protein expression of CD10 than CXCR4^{lo} neutrophils and controls (revised Fig. 2b), and healthy CXCR4^{lo} neutrophils expressed a lower level of CD10, but not negative for CD10 (revised Supplementary Fig. 2a). CD10, also known as common acute lymphoblastic leukemia antigen, can be used as a marker to distinguish mature from immature neutrophils ⁴¹. It has been shown that the absolute numbers of CD10⁺ and CD10⁻ neutrophils of healthy subjects are about $2.5 \times 10^6/\text{ml}$ and $0.3 \times 10^6/\text{ml}$ blood samples, respectively ⁴². However, it is also reported that the peripheral blood of psoriasis patients contains higher numbers of both mature CD10^{pos} and immature CD10^{neg} neutrophils ⁴². They also identified a subpopulation of neutrophils with hypersegmented nuclear morphology, indicative of aged neutrophils that lack CD10, suggesting that CD10 expression is transitory during the developmental process in neutrophils, and maybe especially so in inflammation ⁴². We have revised the description of the Results more accurately.

Revised Supplementary Fig. 2a. (a) Gating strategy and representative flow cytometry histogram of key immune markers between peripheral CXCR4^{lo} and CXCR4^{hi} neutrophils from healthy controls and psoriasis patients. Whole blood was used for analysis.

4) CXCR4 is a chemokine receptor, heavily expressed by various lymphocytes, but also on the surface of some, presumably aged and/or activated neutrophils. Most neutrophil receptors, the surface expression of which can be substantially increased, are stored within mobilizable granules and does not require de novo transcription/protein

synthesis in order to be upregulated. I lack information on the mechanism underlying CXCR4 upregulation:

- is it stored inside granules in CXCR4^{lo} cells?
- Is *de novo* protein synthesis required?
- What is the kinetic of CXCR4 upregulation (the shortest time shown is 2 hrs) and would is this limited time sufficient for the gene and protein to be expressed?

Response: Thank you so much for these critical comments. We agree with the reviewer on the importance of these points and have addressed those in our revised manuscript to the best of our capabilities. Neutrophil activation and degranulation rapidly increase the cell surface levels of multiple proteins to reflect shifting functional needs⁴³. As we showed, the mRNA and total protein expression of CXCR4 was dramatically high in psoriatic neutrophils than that in healthy controls (revised Fig. 1e, revised Supplementary Fig. 1d). However, at present, the kinetics of CXCR4 expression on neutrophils is still largely unknown. Here we have performed a series of additional experiments to explore the mechanism underlying CXCR4 upregulation on psoriatic neutrophils and added one new Figure (revised Fig.5).

(1) To determine if *de novo* protein synthesis is required for CXCR4 in neutrophils, we evaluated mRNA and protein expression of CXCR4 in human neutrophils in response to TNF- α , IL-25, and CXCL12 for indicated times (2, 4, 8, 12, 24 h). Real-time PCR results showed that the mRNA expression of CXCR4 in human neutrophils was increased at 2 h reaching the peak at 24 h with stimulation of TNF- α , IL-25, or CXCL12 (revised Fig. 5f). Similarly, the circulating serotonin-induced upregulation of CXCR4 mRNA in neutrophils was seen at 4 h⁴⁴. It has also been reported that CXCL12 activates CREB⁴⁵, which is the critical transcriptional factor for CXCR4 expression in neutrophils as illustrated in our manuscript (revised Fig. 6). Flow cytometry results show the time-dependent protein expression of CXCR4 on the cell membrane of human neutrophils upon TNF- α , IL-25, or CXCL12 stimulation (revised Fig. 5g). However, the protein expression of CXCR4 on neutrophils in the CXCL12-treated group was decreased after 8 h of stimulation (revised Fig. 5g), probably due to CXCR4 internalization. CXCR4 is a member of the GPCR superfamily that binds to CXCL12, initiating signaling that is subsequently terminated in part by internalization and lysosomal degradation of CXCR4⁴⁶. Therefore, cell immunofluorescence was employed to observe the intracellular trafficking of CXCR4 in CXCL12-treated neutrophils, which displayed high levels of colocalization between CXCR4 and LAMP1 since 4 h, but not in vehicle-treated neutrophils (revised Fig. 5h). To further ascertain whether upregulation of CXCR4 protein expression was dependent on

intracellular protein transport, we tested the effects of brefeldin A, which reversibly blocks protein translocation from the endoplasmic reticulum to the Golgi apparatus. This result showed that the upregulation of CXCR4 on neutrophils was almost completely abrogated by pre-treatment with brefeldin A in the IL-25-, TNF- α -, or CXCL12-treated groups (revised Supplementary Fig. 7g).

In addition, our additional experiments that explored the role of CREB1 in regulating CXCR4^{hi} neutrophils, as we addressed below in Q7, confirmed the recruitment of CREB1 to the promoter regions of CXCR4 and PADI4 in dHL-60 cells (neutrophil-like cells), and that the CREB-CBP complex contributed to CXCR4 expression and CXCR4^{hi} neutrophils in skin inflammation (in revised Fig. 6).

(2) To explore if CXCR4 is stored inside neutrophil granules and mobilized to the surface upon stimulation, we employed imaging flow cytometry to quantify the intensity and distribution of CXCR4 in psoriatic CXCR4^{lo} neutrophils as previously reported⁴⁷. In the data presented (see revised Fig. 5i), we utilized established granule markers: LCN2 for specific granules, and CD63 for azurophil granules. CXCR4^{lo} neutrophils were permeabilized and stained with antibodies against CD63 and LCN2, along with an anti-CXCR4 antibody, and then subjected the neutrophils to imaging flow cytometry to assess for heterotypic granule fusion, i.e., the degree of colocalization of these markers and CXCR4. This was calculated using a built-in colocalization analysis of the IDEAS software⁴⁷. We found that CXCR4 was not co-located with neutrophil granule markers in CXCR4^{lo} neutrophils (revised Fig. 5i, Supplementary Fig. 8a). Another cell immunofluorescence showed consistent results that CXCR4 protein expression did not overlap with neutrophil granule markers such as LCN2 and MMP-9 (revised Supplementary Fig. 8b). Therefore, CXCR4 is not stored inside neutrophil granules.

To address if CXCR4 can be upregulated to the surface of CXCR4^{hi} neutrophils or CXCR4^{lo} neutrophils upon degranulation, similar to the CD177 expression pattern⁴⁸. Neutrophils were analyzed for CXCR4 expression before and after degranulation triggered by short (30 min) treatment with TNF- α (100 ng/ml, n = 6), as previously reported². These results show that TNF- α stimulation does not induce an increase in the mean fluorescence intensity of CXCR4 on neutrophils (revised Supplementary Fig. 8c). This is consistent with previous studies that have profiled human neutrophil granules and secretory vesicles and did not identify CXCR4 amongst 1292 unique proteins identified^{49,50}.

Therefore, we show that CXCR4 upregulation depends on *de novo* mRNA and protein synthesis and on intracellular protein transport. We have added a new Figure 5 in our revised manuscript to illustrate this, substantially revised the Result section, and added these experimental details to the Method/supplementary Method section or corresponding Figure legends. We have also expanded our discussion to address this. All the changes are underlined.

5) Although mentioned in the introduction, the possible impact of LDNs to the results is neither addressed nor discussed. LDNs (it is not unlikely that they are rather abundant in psoriasis patients) would be present in all experiments using whole blood, but not in the experiments using isolated neutrophils. This could be of significant importance.

Response: We agree, and the reviewer raises an important point. We have removed the sentences about low-density neutrophils (LDNs) from the Introduction section to avoid ambiguity and we have added to the discussion information on neutrophil heterogeneity with a focus on the established and known roles of LDNs in psoriasis (page 18; para 1). In fact, one recent study has reported that no significant differences in the proportion of aged neutrophils (CXCR4⁺CD62L^{low}) are observed between LDNs and HDNs⁵¹, suggesting that the classification of the aged/non-aged neutrophils is different from the density classification, and indicating that aged neutrophils are a unique subset.

Emerging evidence has shown that neutrophils exhibit heterogeneous phenotypes under certain physiologic and pathologic conditions^{32, 33} likely reflecting differences in neutrophil density, surface markers, and maturity, although consensus criteria are still lacking. This includes low-density neutrophils (LDNs), N1/N2, reverse-migrated neutrophils, aged neutrophils, CD177⁺neutrophils, OLFM4⁺neutrophils, CD62L^{dim} neutrophils³⁴, and TGFβ1⁺CCR5⁺ neutrophil^{13, 35}. Among these, LDGs are a recently identified subpopulation of neutrophils that are retained in the upper layer containing PBMCs after Ficoll-Paque sedimentation (determined by CD14^{lo}CD15^{hi}CD10^{hi}) (Figure R2a), with different functional properties compared with normal-density neutrophils (NDNs) and all neutrophils (LDNs+NDNs)^{52, 53, 54}. Moreover, “spontaneous” density shift of neutrophils *in vitro* has also been reported due to the delayed processing of blood samples, which may contribute to the wide variety of LDN numbers found between different experimental setups²⁶. In fact, LDNs to date are a heterogeneous group composed of both mature neutrophils and immature neutrophils at various stages of differentiation, and function (pro-inflammatory or suppressive)^{38, 55, 56}. This diversity and plasticity in neutrophil function have been demonstrated in multiple -omic studies^{11, 14, 18, 38, 53}.

Data regarding LDNs in psoriasis vary across studies. Previous studies have shown that the number of circulating LDNs (1.3-fold) and NDNs (2.0-fold) are all higher in psoriasis patients than those in healthy controls ^{57, 58}, and LDNs are more efficient at generating NETs ⁵⁹. Recently, a distinct population of immature CD66⁺CD16⁺ LDNs was characterized in acute generalized pustular psoriasis ⁶⁰. Increased numbers of LDNs at baseline have been reported in some psoriatic arthritis patients but no specific clinical features were identified in correlation with high numbers of LDGs at baseline ⁶¹. In our experiments, we analyzed the neutrophil phenotype from surface markers and maturity but did not assess neutrophil density. It is likely that LDNs would be present in some of the flow analysis experiments, however, we feel that it would be unlikely to affect our findings and conclusions as our study was performed on whole-blood to minimize potential biases related to isolation procedures and took the proinflammatory cytokine environment into account. We have found that the CXCR4^{hi} neutrophils identified in our study appear to have some overlap with mature LDNs, when comparing our data with a previously published study on psoriasis ⁵⁸ on account of similar functional characteristics, including enhanced leukocyte activation, increasing vascular permeability, pro-inflammatory effects, NF-kappa B signaling, NETs formation, enhanced phagocytosis, etc. (Figure R2b). Further studies, outside the scope of our current manuscript, will be needed to fully address the relationship between CXCR4^{hi} and LDNs, but to address this we have added a whole paragraph in the Discussion section.

Figure R2. The comparison between LDNs and CXCR4^{hi} neutrophils in psoriasis. (a) The location of LDNs and isolated neutrophils in the Ficol-Paque-processed blood samples. LDNs

are retained in the upper layer containing PBMCs after Ficoll-Paque sedimentation (then mostly determined by CD14^{lo}CD15^{hi}CD10^{hi}). (b) The comparison of CXCR4^{hi} neutrophils identified in our study and mature LDNs, a previously published study on psoriasis⁵⁸.

6) The authors previously described neutrophil MMP-9 to be the culprit behind cutaneous vasodilation and hyperpermeability in psoriasis (PMID: 32888954). In the current manuscript they instead state that lactate is critical. These two apparently disparate findings need to be reconciled or at least discussed.

Response: Thank you so much for this suggestion. The interaction between neutrophils and vascular endothelial cells (EC) and their contributions to skin inflammation has been an important concern for us^{62, 63, 64}. To address this, we have evaluated MMP-9 release from CXCR4^{hi} and CXCR4^{lo} neutrophils of psoriasis patients and healthy controls via ELISA and Western blotting of the neutrophil supernatant. This demonstrated that psoriatic CXCR4^{hi} neutrophils release more MMP-9 than CXCR4^{lo} neutrophils and healthy controls (revised Fig. 2i, j).

Revised Fig. 2i, j. The MMP-9 level in the supernatant of neutrophils. CXCR4^{hi} and CXCR4^{lo} neutrophils were isolated from psoriasis patients and healthy controls (n = 6) and cultured *in vitro* for 24 h, then the supernatant of each group was collected and processed for ELISA assay (i) and Western blot (j). Data are mean ± SD (n = 6). *** P < 0.001, ns, not significant. Two-way ANOVA with Tukey's post hoc test was used. HC, healthy control; Pso, patients with psoriasis.

Furthermore, we found that either inhibiting MMP-9⁶² or blocking the receptor for lactate (Fig. 4h) reverses endothelial hyperpermeability and decreases immune cell infiltration. Based on these additional data, we consider that both MMP-9 and lactate exert pro-inflammatory effects on ECs through different mechanisms, leading to cutaneous vasodilation and hyperpermeability. Accumulating studies have explored the multiple steps for neutrophil recruitment and diapedesis following infections and injury, which is driven by complex interactions between neutrophils and various components of the blood vessel wall^{65, 66}. For instance, neutrophils and NETs⁶⁷, as well as MMP-9^{68, 69} can degrade glycocalyx on the endothelial cell surface to expose adhesion

molecules and increase endothelial permeability. As we have recently revealed, endothelial glycocalyx destruction, a major feature of EC dysfunction in psoriasis, is a driving force during the process of immune cell extravasation ⁶³. Similarly, MMP-9 produced by monocytes in giant cell arteritis may degrade type IV collagen in medium-sized arteries to promote the entry of CD4⁺T cells and monocytes into the vessel wall to exert inflammatory responses ⁷⁰. It is also reported that MMP-9 facilitates the proteolytic processing of the extracellular portion of platelet endothelial cell adhesion molecule-1 (PECAM-1), thus inducing leukocyte migration across this junctional molecule ⁷¹.

Recently, lactate has begun to be recognized as an active molecule capable of modulating immune responses via its receptors or transporters. For instance, lactate produced by neutrophils following LPS stimulation acts on the lactate GPR81 receptor that is expressed by endothelial cells to reduce VE-cadherin expression, thus increasing vascular permeability ⁷². In addition, lactate accumulation in inflamed tissue contributes to the upregulation of the lactate transporter SLC5A12 by human CD4⁺ T cells, which mediates lactate uptake into CD4⁺ T cells, reshaping their effector phenotype to produce increased IL-17 ⁷³. Developments in genomic, imaging, and sequencing technologies are beginning to enable more in-depth investigations into the pathophysiology of vascular inflammation, making now a good time to re-examine the roles of neutrophils in these processes.

Therefore, there is substantial evidence to support our findings that neutrophils exert their pro-inflammatory effects on ECs via different molecules and mechanisms. We have added the MMP-9 data in revised Fig. 2, added the experimental protocols in the Method section, and further addressed this in our revised Discussion section, and also revised the Graphic Abstract (revised Fig. 9) to better illustrate the crosstalk between neutrophils and ECs.

Minor comments:

7) The use of pharmacological inhibitors is risky without all proper controls to demonstrate that their actions are indeed specific. Especially so re in vivo experiments where active doses of inhibitors are hard to predict/specify. There are no control experiments shown (or mentioned) that e.g., KG-501 does not inhibit non-CREB1 mediated responses.

Response: We apologize for this not being clear in our original submission. We have now extensively updated and enhanced information on the use of inhibitors in our revised manuscript (as underlined in Methods, Results, and Figure Legends).

(1) For each inhibitor used in our experiments, the buffer to dissolve the corresponding inhibitor was used at the same volume as the control. For example, the CREB1 inhibitor (KG-501) was dissolved and stored in DMSO, and the CXCR4 inhibitor (AMD3100 hydrate) in phosphate-buffered saline (PBS) solution, thus DMSO or PBS, matching the inhibitors, was used as the control in cell and/or mouse experiments, as indicated in Methods, Figures, and Figure legends.

(2) As to the CREB1 inhibitor, KG-501 and 666-15 were mostly used in experimental settings^{74, 75}, but may have some mechanism differences. Upon stimulation CREB is activated by phosphorylation on Ser-133 then interacts with its coactivator protein, CREB-binding protein (CBP) to initiate transcription of CREB-responsive genes⁷⁶. KG-501 directly targets the KIX domain of CBP, resulting in a disrupted CREB-CBP complex, inhibiting CREB-target gene induction. Another CREB1 inhibitor 666-15 can effectively inhibit CREB phosphorylation and related target gene transcription but is rather weak to inhibit CREB-CBP interaction. Therefore, we pre-treated human neutrophils with CREB inhibitors 1 h earlier, and then TNF- α and CXCL12 to induce CXCR4 expression. We then compared the dose-response curves for the two different inhibitors, which showed that KG-501 (at 300 μ M) was the more potent inhibitor (revised Supplementary Fig. 10a), compared to the inhibitory effects of 666-15 (Figure R3). Importantly, this dose of KG-501 was not cytotoxic, nor did it have general adverse effects on cell viability (revised Supplementary Fig. 10b). Moreover, we found that silencing of CREB by shRNA and/or treatment with a CREB inhibitor (KG-501) reverts the overexpression and activation of CXCR4 and PAD4, NETs formation, and hyperactivated glycolysis, as well as maturation and activation marker including CD10, CD101, and degranulation marker CD63 (in revised Fig. 6h-j).

Figure R3. Verification of 666-15 inhibition efficiency. Neutrophils from healthy controls were pre-treated with 666-15 for 1 h at indicated concentrations, followed by CXCL12 and TNF- α stimulation for 2h. The CXCR4 expression on neutrophils were analyzed.

Based on this, we further analyzed the binding of activated CREB to its molecular partner CBP to illustrate the mechanisms underlying CXCR4 expression. Cell IF showed that CBP was constitutively expressed in all samples and co-localized with

phosphorylation of CREB in TNF- α and CXCL12-treated neutrophils, however, undetected in normal controls (revised Fig. 6f). Western blot data showed phosphorylation of CREB1 and increased CBP expression in TNF- α and CXCL12-treated neutrophils (revised Fig. 6g). Therefore, we consider that the CREB-CBP complex contributes to CXCR4 expression and CXCR4^{hi} neutrophils in skin inflammation, which can be disrupted appropriately by KG-501. Moreover, chromatin immunoprecipitation (ChIP) assay confirmed the recruitment of CREB1 to the promoter regions of CXCR4 and PADI4 in dHL-60 cells (revised Fig. 6l). To further determine this, we cloned the promoter region of CXCR4 (2.0 kb upstream of its transcription start site) into a luciferase construct and generated a deletion construct that lacked the predicted N-CREB1 binding site. Treatment with CXCL12 or TNF- α resulted in a significant increase in CXCR4 promoter expression, which was found to be inhibited by the deletion of the CREB1 binding site (revised Fig. 6m), demonstrating the direct transcriptional regulation of CXCR4^{hi} genes by CREB1 in neutrophils.

(3) In our *in vivo* experiment, 10mg/kg of AMD3100 hydrate (Sigma-Aldrich), a selective CXCR4 antagonist was administered intraperitoneally every day to inhibit CXCR4 function. AMD3100 (plerixafor), a CXCR4 antagonist, has opened a variety of avenues for potential therapeutic approaches in different refractory diseases, and all the pre-clinical studies support the clinical testing of the monotherapy and combination therapies in humans with safety^{77, 78}. The dose of AMD3100 and the length of treatment were chosen based on previous studies^{79, 80, 81, 82}. Moreover, in our initial experiments, AMD3100 was injected intraperitoneally at various concentrations (0.1, 1, or 10 mg/kg, in 100 μ l), which showed that AMD3100 had activity for inhibition of IMQ-induced skin inflammation, as well as lowering CXCR4^{hi} neutrophils over a broad range of doses, reaching the peak at 10 mg/kg (revised Supplementary Fig. 13). After 6 days, the hematoxylin and eosin (H&E) staining of mouse kidney and liver didn't show any abnormal phenotypes (revised Supplementary Fig. 13g). Further investigation of whether this long period of CXCR4 inhibition can induce side effects will be essential. We have included these explanations in the Method and Result section.

In parallel, anti-CXCL12 was injected intraperitoneally at various concentrations (0.1, 0.5, or 1 mg/kg, in 100 μ l), and the isotype antibody was used as a control. This showed that neutralizing CXCL12 also inhibited the IMQ-induced skin inflammation and infiltration of CXCR4^{hi} neutrophils over a broad range of doses, reaching the peak at 1 mg/kg (revised Supplementary Fig. 14). Notably, no side effects were observed and no damage to organs, including the kidney and liver, was detected (Supplementary Fig. 14f)

We have added these data in the revised Figures and supplementary figures and added the experimental details in the Method section to improve the quality of our data and manuscript.

8) Why is CXCR4 not internalized (resulting in lower surface expression) in cells treated with CXCL12?

Response: Thank you for this comment. This question was also raised by Reviewer 3. We have now added new experiments to address this important issue. As mentioned above in our response to reviewer's comments in Q4, we treated healthy neutrophils with TNF- α , IL-25, or CXCL12 for the indicated time (2, 4, 8, 12, 24 h) and examined the mRNA and protein expression of CXCR4 in neutrophils. Real-time PCR results showed that the mRNA expression of CXCR4 in human neutrophils was increased at 2h with stimulation of either TNF- α , IL-25, or CXCL12, reaching peak level at 24h (revised Fig. 5f, as shown in response to Q4). Furthermore, flow cytometry analysis showed time-dependent protein overexpression of CXCR4 on neutrophil cell surface upon TNF- α , IL-25, or CXCL12 stimulation at early timepoints (<8 h), although lower in the CXCL12-treated group beyond 8h (revised Fig. 5g). We believe that the changes in CXCR4 protein expression on the membrane in response to CXCL12 are likely caused by CXCR4 internalization. Therefore, we further employed cell immunofluorescence to observe the intracellular trafficking of CXCR4 in neutrophils treated with CXCL12 or vehicle for 12 h. Following stimulation and fixation, cell IF showed high levels of colocalization between CXCR4 and lysosomal protein LAMP1, but not in vehicle-treated neutrophils (revised Fig. 5h). These data demonstrate CXCL12 elicits an intracellular itinerary that facilitates the degradation of CXCR4 in terminal endocytic compartments.

Receptor internalization occurs during inflammation/activation, impacting surface receptor levels, and relies on endocytic mechanisms, generally involving caveolae and/or clathrin-mediated endocytosis and redistribution mechanics^{83, 84}. As examples, CXCR1, 2, and 4 have been described to undergo receptor internalization, modifying the interaction and activation potential of different cells^{85, 86}. It is reported that the CXCR4-CXCL12 axis and internalization, as well as lysosomal degradation of CXCR4, are regulated by complicated posttranslational modifications, including phosphorylation and ubiquitination⁴⁶. However, the mechanisms that regulate cell surface expression, membrane trafficking, and CXCR4 recycling in neutrophils are largely unknown and warrant further in-depth investigation. We have expanded on our discussion to better address this.

9) Nuclear/cellular morphology and NETs results would be much strengthened by proper quantification.

Response: We appreciate the reviewer's comment. We have now quantified the Nuclear/cellular morphology (revised Fig. 2a) and NETs (revised Figure 2e), the method of which has been detailed and underlined in the supplementary Method.

10) How were murine neutrophils isolated from blood?

Response: We apologize for not being clear in regard to this in our original version. Here we performed isolation of peripheral blood neutrophils using a Neutrophil Isolation Kit (Miltenyi Biotec) according to the manufacturer's instructions. Ly6G is a neutrophil-specific cell surface marker in mice, therefore, these Ly6G-based positive selection strategies can result in high yield, high purity, and high viability of neutrophils^{87, 88}. Detailed information for the isolation of neutrophils and the purity of the cells was provided in the Methods section and Supplementary Fig. 17. As previously shown, mouse neutrophils isolated by above protocols can be used for examining various cellular functions *ex vivo* and for neutrophil adoptive transfer experiments⁸⁸. These details have been added and underlined in our revised supplementary Method section.

11) Viability assays (Fig. 2C) would be much more informative if an early cell death/apoptosis marker was also included.

Response: We agree with the reviewer. Therefore, we have collected more psoriasis patient blood samples (n = 6) and evaluated early and late apoptosis markers of CXCR4^{hi} neutrophils via AnnexinV/7-AAD assay, which showed that the percentage of early apoptotic cells was higher in psoriatic CXCR4^{hi} neutrophils than that in healthy controls, both at 0h and 24 h; however, no significant difference in late apoptosis was observed between psoriatic CXCR4^{hi} neutrophils and healthy controls (revised Fig. 2c, Supplementary Fig. 2b). We have added these data to our revised Results and Figures, and also added experimental details and gating strategy in the Method section/supplementary methods.

Revised Fig. 2c. (c) Assessment of CXCR4^{hi} neutrophil survival by Annexin V and 7-AAD staining after 24 hours of culture. Analyses: two-way ANOVA with Tukey's posthoc test was used in c. * $P < 0.05$, ** $P < 0.01$, *** $P < 0.001$, ns, not significant. HC, healthy control; MFI, mean fluorescence intensity; Pso, psoriasis patients.

12) Statistical calculations based on $n=3$ are not meaningful

Response: Thank you for this suggestion. We have collected more patient samples and increased the sample size in Figures 1h-i, 2c-d, 2g-h, 2k, 3, 4, 5b, 5g, 5i, 6c, 6j, 7, 8 and performed all suggested *in vivo* and *in vitro* experiments to improve this.

13) The IMQ model of psoriasis was shown to be rather completely dependent on IL17/il23 (PMID: 19380832). In order to convincingly claim that CXCR4/CXCL12 is as critical, ko mice should be used.

Response: Thanks for your suggestion. We agree that topical application of imiquimod (IMQ), a TLR7/8 ligand and potent immune regulator, mimics human psoriasis mainly depending on IL-17/IL-23 axis¹⁸. However, massive neutrophil infiltration and neutrophil-related signaling can be also detected in the lesional skin of IMQ mice⁸⁹, suggesting that the IMQ mouse model is suitable to study the role of neutrophils in skin inflammation^{90, 91, 92, 93, 94}. Therefore, depleting neutrophils or inhibiting neutrophil bioactive molecules leads to alleviation of IMQ-induced skin inflammation^{93, 95, 96}.

We like to stress that the purpose of our study is not to show that the neutrophil CXCR4/CXCL12 axis is more important in psoriasis than the IL-17/IL-23 axis, the importance of which has been shown in multiple studies and clinical trials^{97, 98, 99}. Instead, we are characterizing an inflammatory circuit involving neutrophils to determine their role, and that of CXCR4/CXCL12, in psoriasis pathogenesis. Our additional data demonstrated that the expression of both CXCL12 and CXCR4 was upregulated in inflamed skin of IMQ-induced skin inflammation (revised Fig. 7a), with

CXCR4 being primarily found on infiltrating neutrophils (revised Fig. 7b). Serum CXCL12 level was also elevated in IMQ mice compared to control (revised Fig. 7c). Treatment with the CXCR4 antagonist AMD3100 or CXCL12 neutralizing antibody potently suppresses skin inflammation in IMQ mice, associated with reduced vascular remodeling and inflammatory cell infiltration (revised Fig. 8).

Therefore, our findings on the role of CXCR4/CXCL12 are clear from the data presented in our manuscript. We also performed a variety of interventions centered on CREB1-CXCR4^{hi} neutrophils. KG-501 inhibited CREB1 activation in psoriatic neutrophils, AMD3100 or CXCL12 neutralizing antibody blocked the CXCL12-CXCR4 axis and accumulation of CXCR4^{hi} neutrophils, and DNaseI removed the formed NETs as we previously reported⁹³; all of these agents prevented skin inflammation to varying degrees, with potential clinical applications. Undoubtedly, conditional knockout mice are needed to further confirm the importance of the CREB1-CXCR4^{hi} neutrophils axis, which is the direction for our future research. We want to stress that this knockout is not readily available (we have reached out to several groups without success), and generating a new knockout mouse line would end up delaying our publication by at least 1 year. It would be unlikely that the use of KO mice would add much more to our paper than what we have already demonstrated with the specific inhibitor and monoclonal targeting antibodies. Instead, and we hope that the reviewer will find this acceptable, we have added this to our Discussion as one of the limitations of our data/findings in our revised manuscript.

Revised Fig. 7a-c. (a) Relative mRNA expressions of CXCR4 and CXCL12 in mice tissues. (b) Proportion of Ly6G⁺CXCR4^{hi} neutrophils in murine skin was determined by flow cytometry. (c) Serum level of CXCL12 in control and IMQ-treated mice. Mean \pm SD (n = 6 biologically independent samples/group). Analyses: unpaired Student's t-test in **a**, **b**, and **c**. * $P < 0.05$, ** $P < 0.01$, *** $P < 0.001$, ns, not significant; IMQ, imiquimod.

14) Circadian rhythm have been earlier shown to affect CXCR4 expression, i.e., aging in the blood. Were all blood samples collected at a similar time of the day? When?

Response: We agree with the reviewer that circadian rhythm may affect CXCR4 expression in neutrophils and we considered this when implementing this project. All blood samples in this study were collected at 7:00 - 8:00 am and immediately processed for cell isolation and serum collection. Then isolated neutrophils were processed immediately for the next experimental steps and serum was frozen at -80°C. Moreover, all selected participants in this study reported both regular work schedules and sleep-wake patterns in the preceding 4 weeks. As neutrophil stimulation assays were performed at different times of the day, the data regarding percentages of CXCR4^{hi} neutrophils or CXCR4 expression were normalized by calculating the ratio between the values of samples containing stimulus and the values of samples with corresponding controls in each test. We have added and underlined these details in our revised Method section to increase the clarity of this study design.

Recent data suggest that the trafficking and recruitment of leukocytes are rhythmic during light-dark cycles, both under physiological conditions and during inflammation¹⁰⁰. Previous studies have shown that circulating neutrophil counts peak between 20:00 and 2:00 in humans^{101, 102}. It is also reported that in humans and in mice, the proportion of CXCR4 labeled aged neutrophils is relatively high in the evening and relatively low in the morning, and no difference in the proportion of aged/young cells was observed in samples obtained at 7 am^{103, 104}. Therefore, CXCR4 expression is relatively low at 7:00 - 8:00 am in the morning compared to different times of the day, and we collected all blood samples at 7:00 - 8:00 am and immediately processed them for cell isolation and serum collection, which would not affect the overall conclusions in our study. This is also the reason why we did not treat human neutrophils for prolonged periods of time in the *in vitro* settings, as CXCR4 expression will increase along with time.

Therefore, we have added text to address this in our revised Method and Discussion section.

Reviewer #2 (Remarks to the Author):

Chen, Bai, Zhu et al describe proinflammatory CXCR4+ neutrophils in psoriasis as relevant immune cells orchestrating skin inflammation. They use state-of-the-art techniques to address a relevant question in a cutaneous inflammation and use elegant experiments to show that neutrophils up regulate CXCR4 in response to proinflammatory stimuli, are recruited to skin via CXCL12 produced by endothelial cells and induce vascular damage by lactate. They furthermore describe CREB1 as transcription factor relevant for their pro inflammatory phenotype. The relevance of their data from psoriasis patients is corroborated with a mouse model for acute psoriasiform inflammation, where inhibition of neutrophils, CXCR4 or its ligand reduced inflammation.

Response: We appreciate these positive comments and helpful suggestions. We have collected additional psoriasis patient samples for RNA-sequencing and all the other functional experiments that were noted with a limited sample size of $n = 3$. Now the sample size is at minimum $n = 6$ or more in each experiment. We also performed additional experiments including multi-staining of psoriatic skin lesions and provided additional details including patient demographics, sample size, representative FACS dot plots, and quantification of immunofluorescence and Western blot to increase the clarity of this manuscript.

While these observations are of potential interest, the following points need to be addressed in a revised manuscript:

1) Patient details should be described in more detail and, given low patient number, be presented for the individual figures as supplements, including age, comorbidities, BMI, and disease duration. In general, patient numbers need to be added to the Figure legends for the individual figures.

Response: We agree with the reviewer. We have now added the patient details for the blood samples used in this work (see the Methods section). The median age of psoriasis patients was 32.2 yrs, whereas healthy controls were 30 yrs. Other available clinical information including age, comorbidities, BMI, and disease duration are indicated in revised Table S1.

2) In Fig.3, patient numbers for sequencing are too low ($n=3$ and 4 for healthy controls and psoriasis patients, respectively). Also, it should be mentioned if patients with short disease duration have another signature compared to chronic psoriatic patients. It could

be that the reported information on neutrophils is especially important in acute inflammation and less so in patients with a long disease duration.

Response: We thank the reviewer for these helpful suggestions. To address this, we have recruited additional psoriasis patients and healthy volunteers, and collected blood samples for isolating CXCR4^{hi} and CXCR4^{lo} neutrophils for RNA-seq analysis. The inclusion and exclusion criteria are now clearly detailed and underlined in the Method section. We re-analyzed the RNA-sequencing of CXCR4^{hi} and CXCR4^{lo} neutrophils (n = 6 for psoriasis patients and n = 7 healthy controls) (see revised Fig. 3), and replaced the graphs shown previously in Fig. 3. Patient demographics, including age, comorbidities, BMI, and disease duration are shown in revised Table S1.

We agree with the reviewer's suggestion that exploring the relationship of CXCR4^{hi} neutrophils in psoriasis patients against disease duration might add clinical and translational value. However, it is hampered by the fact that the onset of inflammatory disease is often difficult to determine and delays for these patients to gain access to our specialty dermatology clinics. Also, to our knowledge, there is no firm definition of "short duration" of psoriasis, and we'd need a very large cohort of patients to adequately address this. In our study (in Fig. 1c, d), 21 psoriasis patients were included with a disease duration of 0.5-16 years. To further substantiate our finding, we recruited additional patients and divided them into three groups based on disease duration: less than 5 year, 5-10 years, and longer than 10 years, and then evaluated the frequency of CXCR4^{hi} neutrophils. Our data suggest that the ratio of CXCR4^{hi} neutrophils is not associated with disease duration (revised Supplementary Fig. 1c), but positively correlated with psoriasis severity as measured by PASI (shown in revised Fig. 1d).

Based on these additional findings, we have added sentences in the Result and Discussion section related to the effect of disease duration on CXCR4^{hi} neutrophils. All the experimental details were added in Method section and corresponding Figure legends.

Revised Supplementary Fig. 1c. (c) Correlation of the percentage of psoriatic CXCR4^{hi} neutrophils with disease durations (n = 25). Analyses: The Spearman method (left) and one-way ANOVA with Tukey's posthoc test (right). ns, not significant.

3) Please show representative FACS dot plots for Fig. 3e.

Response: We have shown representative FACS dot plots for Fig. 3e in revised Supplementary Fig. 4e, as well as representative gating strategies for all the other FCM experiments (revised Supplementary Fig. 1a, 1b, 1e, 2a-c, 3d, 4e, 4f, 7a, b, 9a, b, 10a, 12a, 13f, 16a-b, and 17a, b).

4) While it is nicely shown that CXCR4⁺ neutrophils modulate vascular permeability in HMEC-1 cells via the lactate-GPR81 axis, its relevance in psoriatic skin lesions remains to be assessed. Is GPR81 expressed by endothelial cells in psoriatic skin lesions with CXCR4⁺ neutrophils being in close contact?

Response: Thank you so much for this suggestion. We have co-stained psoriatic skin with CD31, GPR81, and CXCR4 to better show the relevance of the lactate-GPR81 axis. The staining results demonstrated that GPR81 was mainly expressed by CD31-positive endothelial cells in psoriatic skin (left) with CXCR4^{hi} neutrophils being in close proximity (right) (revised Fig. 4i). This is in agreement with our findings that CXCR4^{hi} neutrophils modulate vascular permeability in HMEC-1 cells via the lactate-GPR81 axis. We have revised our Figures and Result to reflect these findings.

Revised Fig. 4i. (i) Confocal images of CD31 and GPR81 in psoriatic lesions (n = 6) with CD15⁺CXCR4^{hi} neutrophils adjacent to GPR81⁺ vascular endothelial cells. Scale bar = 50μm, 20μm.

5) While the representative picture in Fig 5f shows some p-CREB1/CD15 double positive cells, it is hard to make the conclusions from a single picture. Please quantify p-CREB1/CD15 double positive cells in psoriasis and show co-localization of pCREB1 and CXCR4 in skin lesions similar to blood neutrophils.

Response: Thank you so much for this suggestion. To address this, we have co-stained CD15, CXCR4, and p-CREB1 in lesional skin of psoriasis patients and quantified these (revised Supplementary Fig. 9c). This result demonstrated that p-CREB1 was increased and co-localized with CD15, a neutrophil marker, as well as CXCR4, in inflamed psoriatic skin (revised Fig. 6f).

Revised Supplementary Fig. 9c, d. (c) Representative immunofluorescence staining of CD15, CXCR4, and p-CREB1 in inflamed psoriatic skin (n = 6). Scale bar = 50 µm, 10 µm. (d) Quantitation of p-CREB1/CD15 and p-CREB1/CXCR4 double positive cells in psoriatic lesions. Mean ± SD (n = 6 biologically independent samples/group). Analyses: one-way ANOVA with Tukey's post hoc test. HC, healthy control; Pso, psoriasis patients.

Reviewer #3 (Remarks to the Author):

CXCR4 is known as a key modulator of neutrophil trafficking between tissues, peripheral blood, and the bone marrow and a subset of neutrophils with high CXCR4 surface expression was recently shown to play an important role in inflammation-related lung injury. In the current study, Chen and colleagues demonstrate a pro-inflammatory role of CXCR4-expressing neutrophils in psoriatic skin inflammation and investigate the underlying mechanisms that drive this pathological neutrophil phenotype and that promote CXCR4 expression. The manuscript contains interesting and novel data, e.g. along the connection between CXCR4 expression and the transcription factor CREB1, and is a valid improvement over current knowledge, strengthening our insight into the plasticity of neutrophils during inflammation.

Response: We thank the reviewer for these helpful suggestions. To address comments below, we have performed additional experiments including scRNA-seq analysis and multi-staining of psoriatic lesions, expression kinetics and internalization of CXCR4 in response to CXCL12 and other cytokines, NET formation, phagocytosis experiments, and early and late apoptosis analysis. Moreover, we have provided details in the revised Figure legends and Method section and corrected inappropriate labels and sentences. These additional experiments show that:

- (1) Endothelial cells, fibroblasts, and pericytes are the main sources of CXCL12 in skin lesions of psoriasis (revised Fig. 5d, Supplementary Fig. 7c, d)
- (2) The dynamic changes of CXCR4 protein expression on the membrane of CXCL12-induced neutrophils may be caused by CXCR4 internalization (revised Fig. 5f-h)
- (3) Psoriatic CXCR4^{hi} neutrophils are more prone to release NETs upon PMA than healthy CXCR4^{hi} neutrophils, or CXCR4^{lo} neutrophils (revised Fig. 2e, Supplementary Fig. 3a, b)
- (4) CXCR4^{hi} neutrophils exhibit a higher phagocytic potential compared to CXCR4^{lo} neutrophils in both psoriasis patients and healthy controls (revised Fig. 2f, g)
- (5) CXCR4^{hi} neutrophils show a higher rate of early apoptosis than CXCR4^{lo} neutrophils, which is further enhanced in the psoriasis group (revised Fig. 2c, Supplementary Fig. 2b).

We also have added the reported new findings about CXCR4 in lung injury¹⁰⁵ and neutrophil heterogeneity and plasticity to our Discussion section. All changes are underlined.

I have some specific questions and comments that could further improve the manuscript:

1) The Figure legends and the text in general need more experimental details to understand what is exactly shown and how it could be interpreted. For example, group sizes (n per group) should always be stated. A few specific examples of missing information to follow:

Fig. 1G: How many samples were tested?

Fig. 2B: What's the color scale? Z-score?

How was NET formation assessed in Fig. 2E?

Fig. 2H: How many samples were tested?

Fig. 3A: Should the heading read CXCR4^{hi} vs CXCR4^{lo}?

Fig. 3B: Are the top most significant genes shown in the heatmap or an arbitrary selection of genes?

Fig. 3C: Can you show normalized enrichment scores for differentially regulated pathways, in order to see the direction of the effects?

Response: We thank the reviewer for bringing these points to our attention. In the revised manuscript, we have addressed these concerns as shown below:

Fig. 1G: the sample size is n = 30 fields from 10 psoriasis patient samples;

Fig. 2B: The color scale is log-transformed mean fluorescence intensity, as described in the legend;

The method for assessing NET formation was detailed in the supplementary Method section.

Fig. 2H: n = 6;

Fig. 3A: we have revised the heading to read CXCR4^{hi} vs CXCR4^{lo};

Fig. 3B: The top-most significant genes are shown in the heatmap;

Fig. 3C: Normalized enrichment scores were shown for differentially regulated pathways in Supplementary Fig. 4b.

Moreover, we have increased the sample size in RNA-seq and functional experiments, and more experimental details were described in the corresponding Figure legends and Method section. Due to the limited space, we have removed some detailed methods to the Supplementary files. All the changes are underlined.

2) Suppl. Fig. 1C: The authors claim that CD31⁺ vascular endothelial cells are the main source of the CXCR4 ligand CXCL12 in inflamed skin. In Suppl. Fig. 1C it looks like basically all CXCL12-expressing cells are CD31 positive. This is a bit surprising, because CXCL12 expression was also reported in other skin resident cell types such as keratinocytes and dermal fibroblasts. Can the authors quantify how many % of

CXCL12+ cells are CD31+ and how many independent skin samples they actually investigated?

Response: Thank you for this comment. To address this, we have re-analyzed publicly available scRNA seq data from psoriatic lesional and healthy controls, as previously reported ¹⁰⁶. This demonstrated that CXCL12+ cells are increased in psoriatic lesions, among which endothelial cells (12.37%), fibroblasts (73.87%), and pericytes (7%) were the main sources of CXCL12 in psoriasis lesions (revised Supplementary Fig. 7c, d). We further find that CXCL12 is increased in VE3 (vessel endothelial cell cluster 3) in psoriatic lesions compared to healthy controls (revised Supplementary Fig. 7c). VE3 was identified as the postcapillary venular cells regulating leukocyte adhesion and migration, with higher frequency in atopic dermatitis and psoriasis skin and with a reduction after treatment ¹⁰⁶. Moreover, we performed additional immunofluorescence co-staining of CXCL12 and the endothelial marker CD31 and fibroblast marker vimentin in psoriatic lesions and healthy skin (n = 6) to quantify CXCL12+ cells. The immunofluorescence staining further showed that CXCL12 was mainly expressed by cutaneous vascular endothelial cells and fibroblasts (revised Fig. 5d), with CXCR4^{hi} neutrophils in close proximity (revised Fig. 5e). Therefore, we have added this data to the Results and addressed this further in our revised Discussion.

Revised Supplementary Fig. 7c, d. CXCL12 is mainly derived from vascular endothelium and fibroblasts in psoriatic lesions. (c-d) Expressions of CXCL12 in each cluster were

analyzed from scRNA-seq analysis (<https://developmental.cellatlas.io/diseased-skin>) and the proportions were determined. The y-axis represents log-normalized expression.

Revised Fig. 5d. (d) Representative immunofluorescence staining of CD31, Vimentin, and CXCL12 in inflamed psoriatic skin (n = 6). Scale bar = 50 μm, 20 μm. (e) Representative immunofluorescence staining of CD31, CXCR4, and CXCL12 in inflamed psoriatic skin (n = 6). Scale bar = 50 μm, 20 μm.

3) The upregulation of surface CXCR4 after CXCL12 exposure is also a bit puzzling, since CXCR4, like other chemokine receptors, is internalized following ligand binding and receptor activation. Or isn't it? Can the authors please clarify and discuss?

Response: Thank you for this helpful comment. This question was also raised by Reviewer 1. To address this, we treated healthy neutrophils with TNF- α , IL-25, or CXCL12 for the indicated time (2, 4, 8, 12, 24 h) and examined the mRNA and protein expression of CXCR4 in neutrophils. Real-time PCR results showed that the mRNA expression of CXCR4 in human neutrophils was increased at 2h with stimulation of either TNF- α , IL-25, or CXCL12, reaching peak level at 24h (revised Fig. 5f). Furthermore, flow cytometry analysis showed time-dependent protein overexpression of CXCR4 on neutrophil cell surface upon TNF- α , IL-25, or CXCL12 stimulation at early timepoints (<8 h), although lower in the CXCL12-treated group beyond 8h (revised Fig. 5g). We believe that the changes in CXCR4 protein expression on the membrane in response to CXCL12 are likely caused by CXCR4 internalization. Therefore, we further employed cell immunofluorescence to observe the intracellular trafficking of CXCR4 in neutrophils treated with CXCL12 or vehicle for 12 h. Following stimulation and fixation, cell IF showed high levels of colocalization between CXCR4 and lysosomal protein LAMP1, but not in vehicle-treated neutrophils (revised Fig. 5h). These data demonstrate CXCL12 elicits an intracellular itinerary that facilitates the degradation of CXCR4 in terminal endocytic compartments. It is reported that the CXCR4-CXCL12 axis and internalization, as well as lysosomal degradation of CXCR4, are regulated by complicated posttranslational modifications, including phosphorylation and ubiquitination⁴⁶. The rare autoimmune disease WHIMS is caused

by impaired desensitization and internalization of CXCR4, leading to enhanced chemotactic responsiveness to CXCL12¹⁰⁷.

Therefore, in this study, we have demonstrated that CXCR4 was upregulated in psoriatic neutrophils, correlating with disease activity. The immune microenvironment that includes TNF- α , IL-25, or CXCL12 may increase the expression of CXCR4 in neutrophils and skew this CXCR4^{hi} neutrophils phenotype, and CXCR4 will be also internalized in response to CXCL12. However, the intracellular trafficking of CXCR4 in neutrophils in psoriasis warrants in-depth investigation. We have added this to our supplementary Figures and have expanded on our discussion to address this.

4) As far as I understood only spontaneous ROS production and NET formation was assessed in the manuscript, which is known to correspond to neutrophil activation. Were the NETs in Suppl. Fig. 2B quantified? Have the authors also looked into induced ROS/NET formation by e.g. PMA or calcium ionophore?

Response: As suggested by the reviewer, we have quantified NETs formation in Suppl. Fig. 3b in previous version, which shows that psoriatic CXCR4^{hi} neutrophils spontaneously release greater than 2- fold the number of NETs compared to psoriatic CXCR4^{lo} neutrophils or neutrophils from healthy controls (revised Fig. 2e). The method for assessing NET formation was detailed in the supplementary Method section.

Additionally, we have performed experiments to measure ROS production by neutrophils under spontaneous and phorbol 12-myristate 13-acetate (PMA)-stimulated conditions using the Amplex Red assay. We found that CXCR4^{hi} neutrophils of psoriasis patients produced higher levels of ROS than CXCR4^{lo} neutrophils or neutrophils from healthy controls, which was further enhanced by PMA stimulation (revised Fig. 2d, Supplementary Fig. 2c). We also evaluated the formation of NETs in CXCR4^{hi}/ CXCR4^{lo} neutrophils in response to PMA. NETs formation was observed to be increased in PMA-stimulated CXCR4^{hi} neutrophils compared with that in CXCR4^{lo} neutrophils and was also dramatically increased in PMA-stimulated CXCR4^{hi} neutrophils from psoriasis patients compared with that from normal controls. In contrast, NETs formation was much lower in unstimulated CXCR4^{lo} neutrophils (revised Fig. 2e, Supplementary Fig. 3a, b). We have revised the Result section, with all changes underlined.

5) Chronic inflammation and collateral tissue damage is often considered the dark side of immune function essential for defence against pathogens. In line with that, a

subpopulation of CXCR4^{HI} neutrophils constitute the first line of defence against bacteria by rapidly migrating towards invading pathogens and being able to more efficiently phagocytose them (PMID 27609642). However, in the current manuscript the authors show that CXCR4^{HI} neutrophils have a reduced ability to phagocytose. How can these data be reconciled?

Response: Thank you for this helpful comment. To address this, we have repeated the NET formation and phagocytosis experiments as previously reported ¹⁰⁸. Briefly, to evaluate the phagocytic behavior of CXCR4^{hi} and CXCR4^{lo} neutrophils from psoriasis patients and healthy controls (n = 6), phagocytosis of *pHrodo*TM Green *E. coli* was analyzed *in vitro* by flow cytometry. The results show that phagocytosis of bioparticles did not significantly differ between CXCR4^{lo} neutrophils of psoriasis patients and healthy controls; however, CXCR4^{hi} neutrophils exhibited a significantly higher phagocytic potential as compared with CXCR4^{lo} neutrophils in both psoriasis and healthy control groups (revised Fig. 2f, g, Supplementary Fig. 3c, d). The phagocytic potential of CXCR4^{hi} and CXCR4^{lo} neutrophils for *E. coli* bioparticles was analyzed and is detailed in our revised Methods section. We have revised our Result and Method section with all changes underlined.

Revised Fig. 2f, g. The phagocytic potential of CXCR4^{hi} and CXCR4^{lo} neutrophils for *E. coli* bioparticles. (f, g) Neutrophils were incubated with *pHrodo*TM Green *E. coli* to measure phagocytosis, and analyzed by immunofluorescence staining (f) and flow cytometry (g). Data are mean ± SD (n = 6 biologically independent samples/group). Analyses: two-way ANOVA with Tukey's posthoc test. * $P < 0.05$, ** $P < 0.01$, *** $P < 0.001$, ns, not significant. HC, healthy control; MFI, mean fluorescence intensity; neu, neutrophils; Pso, psoriasis patients.

6) The neutrophil survival time in culture seems pretty long (Fig. 2C). After 24h roughly 90% of neutrophils are still intact. The authors state themselves that neutrophils have a short lifetime and survive less than 24h *in vivo* in the circulation.

Response: Thank you for this question. The lifespan of human neutrophils is relatively short (less than 24h) but may last up to 5.4 days ^{109, 110}. We also cultured healthy

neutrophils *in vitro* for the indicated time and evaluated their cell viability via AnnexinV-7-AAD analysis. Our AnnexinV-7-AAD results showed that the early apoptosis rate of neutrophils was about 14.86%, 47.04%, 67.36%, and 68.24% at 12 h, 24 h, 48 h, and 72 h, respectively (Figure R4). However, *in vitro* culture and *in vivo* environment are different, and future research is needed to adequately perform *in vivo* tracking and neutrophil survival.

Figure R4. Early and late apoptosis of healthy neutrophils *in vitro*. Flow cytometry analysis for Annexin V-7-AAD binding to healthy neutrophils (n=5) that culture *in vitro* for indicated time.

During inflammation and activation, the survival time of neutrophils may increase ¹¹¹. This ensures the continued presence of primed neutrophils at the site of infection. Activation of neutrophils occurs through a variety of cytokines and growth factors and/or bacterial products ¹¹². There are situations where mature neutrophils (as defined by terminally differentiated cells with segmented nuclei and fully formed granules) are able to proliferate outside the bone marrow in response to molecules such as serum amyloid A ¹¹³ or in the spleen following an infection ¹¹⁴. Moreover, tracking photo-activated neutrophils in mice that entered injured tissue records a neutrophil lifespan as long as 48 h ¹¹⁵. We performed additional experiments to detect the early and late apoptosis of CXCR4^{hi} neutrophils in normal control and psoriasis patients. This showed that the percentage of early apoptotic cells was higher in psoriatic CXCR4^{hi} neutrophils than that in healthy controls, both at 0h and 24 h, however, no significant difference in late apoptosis was observed between psoriatic CXCR4^{hi} neutrophils and healthy controls (revised Fig. 2c). Therefore, we have revised our statements about neutrophil lifespan in our revised manuscript and revised the Result section and Discussion.

Minor comments:

7) There are a number of typos and small flaws in the text and the figures, e.g. „amplify“ in the abstract (line 33), „slightly“ (line 239), „subcutaneously“ (line 257), „occuldin“ throughout Fig. 4, „MFI“ and „MIF“ being used for mean fluorescence intensity etc.

Response: Thank you so much for this reminder. We have carefully checked the spelling errors and substantially revised our manuscript, with all changes underlined.

References

1. Clemmensen SN, *et al.* Olfactomedin 4 defines a subset of human neutrophils. *J Leukoc Biol* **91**, 495-500 (2012).
2. Dahlstrand Rudin A, *et al.* The neutrophil subset defined by CD177 expression is preferentially recruited to gingival crevicular fluid in periodontitis. *J Leukoc Biol* **109**, 349-362 (2021).
3. Shao S, *et al.* Neutrophil exosomes enhance the skin autoinflammation in generalized pustular psoriasis via activating keratinocytes. *FASEB J* **33**, 6813-6828 (2019).
4. Lundquist H, Andersson H, Chew MS, Das J, Turkina MV, Welin A. The Olfactomedin-4-Defined Human Neutrophil Subsets Differ in Proteomic Profile in Healthy Individuals and Patients with Septic Shock. *J Innate Immun*, 1-14 (2022).
5. Gohring K, *et al.* Neutrophil CD177 (NB1 gp, HNA-2a) expression is increased in severe bacterial infections and polycythaemia vera. *Br J Haematol* **126**, 252-254 (2004).
6. Huang YH, Lo MH, Cai XY, Liu SF, Kuo HC. Increase expression of CD177 in Kawasaki disease. *Pediatr Rheumatol Online J* **17**, 13 (2019).
7. Andrisani OM. CREB-mediated transcriptional control. *Crit Rev Eukaryot Gene Expr* **9**, 19-32 (1999).
8. Zhou G, *et al.* CD177(+) neutrophils as functionally activated neutrophils negatively regulate IBD. *Gut* **67**, 1052-1063 (2018).
9. Zhang R, *et al.* CD177(+) cells produce neutrophil extracellular traps that promote biliary atresia. *J Hepatol* **77**, 1299-1310 (2022).
10. Xie X, *et al.* Single-cell transcriptome profiling reveals neutrophil heterogeneity in homeostasis and infection. *Nat Immunol* **21**, 1119-1133 (2020).
11. Xue R, *et al.* Liver tumour immune microenvironment subtypes and neutrophil heterogeneity. *Nature* **612**, 141-147 (2022).
12. Wang K, *et al.* Locally organised and activated Fth1(hi) neutrophils aggravate inflammation of acute lung injury in an IL-10-dependent manner. *Nat Commun* **13**, 7703 (2022).
13. Li J, *et al.* TGFbeta1(+)CCR5(+) neutrophil subset increases in bone marrow and causes age-related osteoporosis in male mice. *Nat Commun* **14**, 159 (2023)
14. Montaldo E, *et al.* Cellular and transcriptional dynamics of human neutrophils at steady state and upon stress. *Nat Immunol* **23**, 1470-1483 (2022).
15. Radermecker C, *et al.* Locally instructed CXCR4(hi) neutrophils trigger environment-driven allergic asthma through the release of neutrophil extracellular traps. *Nat Immunol* **20**, 1444-1455 (2019).
16. Adrover JM, *et al.* A Neutrophil Timer Coordinates Immune Defense and Vascular Protection. *Immunity* **50**, 390-402 e310 (2019).
17. Dong Y, *et al.* Neutrophil hyperactivation correlates with Alzheimer's disease progression. *Ann Neurol* **83**, 387-405 (2018).
18. Ballesteros I, *et al.* Co-option of Neutrophil Fates by Tissue Environments. *Cell* **183**, 1282-1297 e1218 (2020).
19. Puga I, *et al.* B cell-helper neutrophils stimulate the diversification and production of immunoglobulin in the marginal zone of the spleen. *Nat Immunol* **13**, 170-180 (2011).
20. Gungabeesoon J, *et al.* A neutrophil response linked to tumor control in immunotherapy. *Cell* **186**, 1448-1464 e1420 (2023).
21. Cali B, *et al.* Atypical CXCL12 signaling enhances neutrophil migration by modulating nuclear deformability. *Sci Signal* **15**, eabk2552 (2022).
22. Palomino-Segura M, Sicilia J, Ballesteros I, Hidalgo A. Strategies of neutrophil diversification. *Nat Immunol* **24**, 575-584 (2023).
23. Zhu X, Zhu J. CD4 T Helper Cell Subsets and Related Human Immunological Disorders. *Int J Mol Sci* **21**, (2020).
24. Deniset JF, Kubes P. Neutrophil heterogeneity: Bona fide subsets or polarization states? *J Leukoc Biol* **103**, 829-838 (2018).
25. Schonbacher M, *et al.* Time from venipuncture to cell isolation: Impact on granulocyte-reactive antibody testing. *Clin Biochem* **63**, 72-78 (2019).
26. McKenna KC, Beatty KM, Vicetti Miguel R, Bilonick RA. Delayed processing of blood increases the frequency of activated CD11b+ CD15+ granulocytes which inhibit T cell function. *J Immunol Methods* **341**, 68-75 (2009).
27. Deng H, Hu N, Wang C, Chen M, Zhao MH. Interaction between CD177 and platelet endothelial cell adhesion molecule-1 downregulates membrane-bound proteinase-3 (PR3) expression on neutrophils and attenuates neutrophil activation induced by PR3-ANCA. *Arthritis Res Ther* **20**, 213 (2018).
28. von Vietinghoff S, *et al.* NB1 mediates surface expression of the ANCA antigen proteinase 3 on human neutrophils. *Blood* **109**, 4487-4493 (2007).
29. Blanter M, *et al.* Method Matters: Effect of Purification Technology on Neutrophil Phenotype and Function. *Front Immunol* **13**, 820058 (2022).
30. Laghmouchi A, Hoogstraten C, Falkenburg JHF, Jedema I. Long-term in vitro persistence of magnetic properties after magnetic bead-based cell separation of T cells. *Scand J Immunol* **92**, e12924 (2020).

31. Soltys J, *et al.* Isolation of bovine neutrophils with biomagnetic beads: comparison with standard Percoll density gradient isolation methods. *J Immunol Methods* **226**, 71-84 (1999).
32. Jaillon S, Ponzetta A, Di Mitri D, Santoni A, Bonocchi R, Mantovani A. Neutrophil diversity and plasticity in tumour progression and therapy. *Nat Rev Cancer* **20**, 485-503 (2020).
33. Calzetti F, Finotti G, Cassatella MA. Current knowledge on the early stages of human neutropoiesis. *Immunol Rev*, (2022).
34. Tak T, *et al.* Human CD62L(dim) neutrophils identified as a separate subset by proteome profiling and in vivo pulse-chase labeling. *Blood* **129**, 3476-3485 (2017).
35. Margraf A, Perretti M. Immune Cell Plasticity in Inflammation: Insights into Description and Regulation of Immune Cell Phenotypes. *Cells* **11**, (2022).
36. Salcher S, *et al.* High-resolution single-cell atlas reveals diversity and plasticity of tissue-resident neutrophils in non-small cell lung cancer. *Cancer Cell* **40**, 1503-1520 e1508 (2022).
37. Vafadarnejad E, *et al.* Dynamics of Cardiac Neutrophil Diversity in Murine Myocardial Infarction. *Circ Res* **127**, e232-e249 (2020).
38. Mistry P, *et al.* Transcriptomic, epigenetic, and functional analyses implicate neutrophil diversity in the pathogenesis of systemic lupus erythematosus. *Proc Natl Acad Sci U S A* **116**, 25222-25228 (2019).
39. Zhang D, *et al.* Neutrophil ageing is regulated by the microbiome. *Nature* **525**, 528-532 (2015).
40. Ng LG, Ostuni R, Hidalgo A. Heterogeneity of neutrophils. *Nat Rev Immunol* **19**, 255-265 (2019).
41. Marini O, *et al.* Mature CD10(+) and immature CD10(-) neutrophils present in G-CSF-treated donors display opposite effects on T cells. *Blood* **129**, 1343-1356 (2017).
42. Rodriguez-Rosales YA, *et al.* Immunomodulatory aged neutrophils are augmented in blood and skin of psoriasis patients. *J Allergy Clin Immunol* **148**, 1030-1040 (2021).
43. Faurschou M, Borregaard N. Neutrophil granules and secretory vesicles in inflammation. *Microbes Infect* **5**, 1317-1327 (2003).
44. Garcia F, *et al.* CXCR4(hi) effector neutrophils in sickle cell anemia: potential role for elevated circulating serotonin (5-HT) in CXCR4(hi) neutrophil polarization. *Sci Rep* **10**, 14262 (2020).
45. Corcoran KE, Malhotra A, Molina CA, Rameshwar P. Stromal-derived factor-1alpha induces a non-canonical pathway to activate the endocrine-linked Tac1 gene in non-tumorigenic breast cells. *J Mol Endocrinol* **40**, 113-123 (2008).
46. Caballero A, Mahn SA, Ali MS, Rogers MR, Marchese A. Heterologous regulation of CXCR4 lysosomal trafficking. *J Biol Chem* **294**, 8023-8036 (2019).
47. Bjornsdottir H, Welin A, Dahlgren C, Karlsson A, Bylund J. Quantification of heterotypic granule fusion in human neutrophils by imaging flow cytometry. *Data Brief* **6**, 386-393 (2016).
48. Goldschmeding R, *et al.* Further characterization of the NB 1 antigen as a variably expressed 56-62 kD GPI-linked glycoprotein of plasma membranes and specific granules of neutrophils. *Br J Haematol* **81**, 336-345 (1992).
49. Rorvig S, Ostergaard O, Heegaard NH, Borregaard N. Proteome profiling of human neutrophil granule subsets, secretory vesicles, and cell membrane: correlation with transcriptome profiling of neutrophil precursors. *J Leukoc Biol* **94**, 711-721 (2013).
50. Lominadze G, Powell DW, Luerman GC, Link AJ, Ward RA, McLeish KR. Proteomic analysis of human neutrophil granules. *Mol Cell Proteomics* **4**, 1503-1521 (2005).
51. Yang C, *et al.* Aged neutrophils form mitochondria-dependent vital NETs to promote breast cancer lung metastasis. *J Immunother Cancer* **9**, (2021).
52. Lood C, *et al.* Neutrophil extracellular traps enriched in oxidized mitochondrial DNA are interferogenic and contribute to lupus-like disease. *Nat Med* **22**, 146-153 (2016).
53. Tay SH, Celhar T, Fairhurst AM. Low-Density Neutrophils in Systemic Lupus Erythematosus. *Arthritis Rheumatol* **72**, 1587-1595 (2020).
54. Hassani M, *et al.* On the origin of low-density neutrophils. *J Leukoc Biol* **107**, 809-818 (2020).
55. Scapini P, Marini O, Tecchio C, Cassatella MA. Human neutrophils in the saga of cellular heterogeneity: insights and open questions. *Immunol Rev* **273**, 48-60 (2016).
56. Goretti Rica I, *et al.* Neutrophil heterogeneity and emergence of a distinct population of CD11b/CD18-activated low-density neutrophils after trauma. *J Trauma Acute Care Surg* **94**, 187-196 (2023).
57. Lin AM, *et al.* Mast cells and neutrophils release IL-17 through extracellular trap formation in psoriasis. *J Immunol* **187**, 490-500 (2011).
58. Teague HL, *et al.* Neutrophil Subsets, Platelets, and Vascular Disease in Psoriasis. *JACC Basic Transl Sci* **4**, 1-14 (2019).
59. Skrzeczynska-Moncznik J, *et al.* Differences in Staining for Neutrophil Elastase and its Controlling Inhibitor SLPI Reveal Heterogeneity among Neutrophils in Psoriasis. *J Invest Dermatol* **140**, 1371-1378 e1373 (2020).
60. Yu N, Qin H, Yu Y, Li Y, Lu J, Shi Y. A Distinct Immature Low-Density Neutrophil Population Characterizes Acute Generalized Pustular Psoriasis. *J Invest Dermatol* **142**, 2831-2835 e2835 (2022).
61. Cross AL, *et al.* Neutrophil function following treatment of psoriatic arthritis patients with secukinumab: altered cytokine signalling but no impairment of host defence. *Rheumatology (Oxford)*, (2023).

62. Chen J, *et al.* Neutrophils Enhance Cutaneous Vascular Dilation and Permeability to Aggravate Psoriasis by Releasing Matrix Metalloproteinase 9. *J Invest Dermatol* **141**, 787-799 (2021).
63. Li Q, *et al.* An IGFBP7^{high} endothelial cell subset drives T cell extravasation in psoriasis via endothelial glycocalyx degradation. *J Clin Invest.* (2023).
64. Zhu Z, *et al.* Aryl Hydrocarbon Receptor in Cutaneous Vascular Endothelial Cells Restricts Psoriasis Development by Negatively Regulating Neutrophil Recruitment. *J Invest Dermatol* **140**, 1233-1243 e1239 (2020).
65. Joulia R, *et al.* Neutrophil breaching of the blood vessel pericyte layer during diapedesis requires mast cell-derived IL-17A. *Nat Commun* **13**, 7029 (2022).
66. Wang L, Luqmani R, Udalova IA. The role of neutrophils in rheumatic disease-associated vascular inflammation. *Nat Rev Rheumatol* **18**, 158-170 (2022).
67. Qiao X, *et al.* Biological Effects of Intravenous Vitamin C on Neutrophil Extracellular Traps and the Endothelial Glycocalyx in Patients with Sepsis-Induced ARDS. *Nutrients* **14**, (2022).
68. Zhang D, *et al.* Syndecan-1 Shedding by Matrix Metalloproteinase-9 Signaling Regulates Alveolar Epithelial Tight Junction in Lipopolysaccharide-Induced Early Acute Lung Injury. *J Inflamm Res* **14**, 5801-5816 (2021).
69. Gao W, *et al.* Doxycycline can reduce glycocalyx shedding by inhibiting matrix metalloproteinases in patients undergoing cardiopulmonary bypass: A randomized controlled trial. *Microvasc Res* **142**, 104381 (2022).
70. Watanabe R, *et al.* MMP (Matrix Metalloprotease)-9-Producing Monocytes Enable T Cells to Invade the Vessel Wall and Cause Vasculitis. *Circ Res* **123**, 700-715 (2018).
71. Kato H, Kuriyama N, Duarte S, Clavien PA, Busuttill RW, Coito AJ. MMP-9 deficiency shelters endothelial PECAM-1 expression and enhances regeneration of steatotic livers after ischemia and reperfusion injury. *J Hepatol* **60**, 1032-1039 (2014).
72. Khatib-Massalha E, *et al.* Lactate released by inflammatory bone marrow neutrophils induces their mobilization via endothelial GPR81 signaling. *Nat Commun* **11**, 3547 (2020).
73. Pucino V, *et al.* Lactate Buildup at the Site of Chronic Inflammation Promotes Disease by Inducing CD4(+) T Cell Metabolic Rewiring. *Cell Metab* **30**, 1055-1074 e1058 (2019).
74. Stafim da Cunha R, *et al.* Uremic toxins activate CREB/ATF1 in endothelial cells related to chronic kidney disease. *Biochem Pharmacol* **198**, 114984 (2022).
75. Zheng W, *et al.* cAMP-response element binding protein mediates podocyte injury in diabetic nephropathy by targeting lncRNA DLX6-AS1. *Metabolism* **129**, 155155 (2022).
76. Wen AY, Sakamoto KM, Miller LS. The role of the transcription factor CREB in immune function. *J Immunol* **185**, 6413-6419 (2010).
77. Wang J, Tannous BA, Poznansky MC, Chen H. CXCR4 antagonist AMD3100 (plerixafor): From an impurity to a therapeutic agent. *Pharmacol Res* **159**, 105010 (2020).
78. Biasci D, *et al.* CXCR4 inhibition in human pancreatic and colorectal cancers induces an integrated immune response. *Proc Natl Acad Sci U S A* **117**, 28960-28970 (2020).
79. Sun Q, *et al.* C-X-C-Chemokine-Receptor-Type-4 Inhibitor AMD3100 Attenuates Pulmonary Inflammation and Fibrosis in Silicotic Mice. *J Inflamm Res* **15**, 5827-5843 (2022).
80. Liu P, *et al.* CXCL12/CXCR4 axis as a key mediator in atrial fibrillation via bioinformatics analysis and functional identification. *Cell Death Dis* **12**, 813 (2021).
81. Lukacs NW, Berlin A, Schols D, Skerlj RT, Bridger GJ. AMD3100, a CxCR4 antagonist, attenuates allergic lung inflammation and airway hyperreactivity. *Am J Pathol* **160**, 1353-1360 (2002).
82. Matthys P, *et al.* AMD3100, a potent and specific antagonist of the stromal cell-derived factor-1 chemokine receptor CXCR4, inhibits autoimmune joint inflammation in IFN-gamma receptor-deficient mice. *J Immunol* **167**, 4686-4692 (2001).
83. Zhu XD, *et al.* Caveolae-dependent endocytosis is required for class A macrophage scavenger receptor-mediated apoptosis in macrophages. *J Biol Chem* **286**, 8231-8239 (2011).
84. Kumari S, Mg S, Mayor S. Endocytosis unplugged: multiple ways to enter the cell. *Cell Res* **20**, 256-275 (2010).
85. Rose JJ, Foley JF, Murphy PM, Venkatesan S. On the mechanism and significance of ligand-induced internalization of human neutrophil chemokine receptors CXCR1 and CXCR2. *J Biol Chem* **279**, 24372-24386 (2004).
86. Forster R, *et al.* Intracellular and surface expression of the HIV-1 coreceptor CXCR4/fusin on various leukocyte subsets: rapid internalization and recycling upon activation. *J Immunol* **160**, 1522-1531 (1998).
87. Ubags NDJ, Suratt BT. Isolation and Characterization of Mouse Neutrophils. *Methods Mol Biol* **1809**, 45-57 (2018).
88. Swamydas M, Luo Y, Dorf ME, Lionakis MS. Isolation of Mouse Neutrophils. *Curr Protoc Immunol* **110**, 3 20 21-23 20 15 (2015).
89. Swindell WR, *et al.* Imiquimod has strain-dependent effects in mice and does not uniquely model human psoriasis. *Genome Med* **9**, 24 (2017).
90. Hardman CS, *et al.* CD1a promotes systemic manifestations of skin inflammation. *Nat Commun* **13**, 7535

- (2022).
91. Adachi A, *et al.* Estradiol suppresses psoriatic inflammation in mice by regulating neutrophil and macrophage functions. *J Allergy Clin Immunol* **150**, 909-919 e908 (2022).
 92. Sumida H, *et al.* Interplay between CXCR2 and BLT1 facilitates neutrophil infiltration and resultant keratinocyte activation in a murine model of imiquimod-induced psoriasis. *J Immunol* **192**, 4361-4369 (2014).
 93. Shao S, *et al.* Neutrophil Extracellular Traps Promote Inflammatory Responses in Psoriasis via Activating Epidermal TLR4/IL-36R Crosstalk. *Front Immunol* **10**, 746 (2019).
 94. Costa S, *et al.* Neutrophils inhibit gammadelta T cell functions in the imiquimod-induced mouse model of psoriasis. *Front Immunol* **13**, 1049079 (2022).
 95. Neu SD, Strzepa A, Martin D, Sorci-Thomas MG, Pritchard KA, Jr., Dittel BN. Myeloperoxidase Inhibition Ameliorates Plaque Psoriasis in Mice. *Antioxidants (Basel)* **10**, (2021).
 96. Zhukov AS, Khairutdinov VR, Samtsov AV, Krasavin M, Garabadzhiu AV. Preclinical efficacy investigation of human neutrophil elastase inhibitor sivelestat in animal model of psoriasis. *Skin Health Dis* **2**, e90 (2022).
 97. Ghoreschi K, Balato A, Enerback C, Sabat R. Therapeutics targeting the IL-23 and IL-17 pathway in psoriasis. *Lancet* **397**, 754-766 (2021).
 98. Majumder S, McGeachy MJ. IL-17 in the Pathogenesis of Disease: Good Intentions Gone Awry. *Annu Rev Immunol* **39**, 537-556 (2021).
 99. Kim J, Krueger JG. Highly Effective New Treatments for Psoriasis Target the IL-23/Type 17 T Cell Autoimmune Axis. *Annu Rev Med* **68**, 255-269 (2017).
 100. Zhong Y, Yu X, Li X, Zhou H, Wang Y. Augmented early aged neutrophil infiltration contributes to late remodeling post myocardial infarction. *Microvasc Res* **139**, 104268 (2022).
 101. Sennels HP, Jorgensen HL, Hansen AL, Goetze JP, Fahrenkrug J. Diurnal variation of hematology parameters in healthy young males: the Bispebjerg study of diurnal variations. *Scand J Clin Lab Invest* **71**, 532-541 (2011).
 102. Haus E, Smolensky MH. Biologic rhythms in the immune system. *Chronobiol Int* **16**, 581-622 (1999).
 103. Ella K, Csepanyi-Komi R, Kaldi K. Circadian regulation of human peripheral neutrophils. *Brain Behav Immun* **57**, 209-221 (2016).
 104. Casanova-Acebes M, *et al.* Rhythmic modulation of the hematopoietic niche through neutrophil clearance. *Cell* **153**, 1025-1035 (2013).
 105. Song D, *et al.* PTP1B inhibitors protect against acute lung injury and regulate CXCR4 signaling in neutrophils. *JCI Insight* **7**, (2022).
 106. Reynolds G, *et al.* Developmental cell programs are co-opted in inflammatory skin disease. *Science* **371**, (2021).
 107. Dale DC, Dick E, Kelley M, Makaryan V, Connelly J, Bolyard AA. Family studies of warts, hypogammaglobulinemia, immunodeficiency, myelokathexis syndrome. *Curr Opin Hematol* **27**, 11-17 (2020).
 108. Uhl B, *et al.* Aged neutrophils contribute to the first line of defense in the acute inflammatory response. *Blood* **128**, 2327-2337 (2016).
 109. Pillay J, *et al.* In vivo labeling with 2H2O reveals a human neutrophil lifespan of 5.4 days. *Blood* **116**, 625-627 (2010).
 110. Lahoz-Beneytez J, *et al.* Human neutrophil kinetics: modeling of stable isotope labeling data supports short blood neutrophil half-lives. *Blood* **127**, 3431-3438 (2016).
 111. Summers C, Rankin SM, Condliffe AM, Singh N, Peters AM, Chilvers ER. Neutrophil kinetics in health and disease. *Trends Immunol* **31**, 318-324 (2010).
 112. Kim MH, *et al.* Neutrophil survival and c-kit(+)-progenitor proliferation in Staphylococcus aureus-infected skin wounds promote resolution. *Blood* **117**, 3343-3352 (2011).
 113. De Santo C, *et al.* Invariant NKT cells modulate the suppressive activity of IL-10-secreting neutrophils differentiated with serum amyloid A. *Nat Immunol* **11**, 1039-1046 (2010).
 114. Deniset JF, Surewaard BG, Lee WY, Kubes P. Splenic Ly6G(high) mature and Ly6G(int) immature neutrophils contribute to eradication of S. pneumoniae. *J Exp Med* **214**, 1333-1350 (2017).
 115. Wang J, Hossain M, Thanabalasuriar A, Gunzer M, Meininger C, Kubes P. Visualizing the function and fate of neutrophils in sterile injury and repair. *Science* **358**, 111-116 (2017).

REVIEWER COMMENTS

Reviewer #1 (Remarks to the Author):

Clearly, the authors have put in impressive effort in supplementing the manuscript with new data/sections/supplements. Unfortunately, these additions make the manuscript even harder to follow and clear information re (in my opinion) critical things is lost. I am still not convinced that CXCR4hi neutrophils are really a subtype with distinct functional features, or that CXCR4 is even the best marker to define the activated neutrophils found in psoriasis.

Reviewer #2 (Remarks to the Author):

The authors sufficiently addressed all criticisms and have substantially improved the revised manuscript. I have no further remarks and recommend to accept the manuscript for publication.

Reviewer #3 (Remarks to the Author):

After reading the revised version I have still a few questions and remarks:

1. Revised Figs. 5d & 5e: Since the overlay between and green (Vimentin+CXCL12+ or CXCR4+CXCL12+, respectively) results in yellow color, yellow is not the best choice for showing expression of CD31. In fact, it is hardly possible to make out in these images if there is strong CXCL12 expression in CD31+ cells.
2. Can you give an explanation why 3 data sets (CXCL12-expression, cell death in CXCR4LO vs CXCR4HI cells and phagocytosis are so fundamentally different in the revised version as compared to the original version of the manuscript?
3. Typo: Line 290: CXCR12-treated group.

Reviewer #4 (Remarks to the Author):

The authors did an exceptional job in responding to Reviewer #1's concerns. However, the reviewer has issue with the title of the manuscript, which states that CXCR4+ PMN-mediated skin inflammation occurs via glycolysis. Although the authors have shown that the CXCR4+ PMNs are associated with increased glycolytic activity and markers, nowhere in the manuscript was it shown that CXCR4+ PMN glycolysis was mechanistically relevant to skin inflammation. This was somewhat shown in an in vitro assay by blocking lactate, but not in vivo using glycolytic deficient PMNs. Therefore, the title should be modified to appropriately reflect the conclusions of the data. For example, CREB1-driven CXCR4^{hi} neutrophils orchestrate skin inflammation.

Point-by-point response

Reviewer #1

Clearly, the authors have put in impressive effort in supplementing the manuscript with new data/sections/supplements. Unfortunately, these additions make the manuscript even harder to follow and clear information re (in my opinion) critical things is lost. I am still not convinced that CXCR4^{hi} neutrophils are really a subtype with distinct functional features, or that CXCR4 is even the best marker to define the activated neutrophils found in psoriasis.

Response: Thank you so much for your comments. We have taken additional steps to try to address your concerns and to further strengthen our conclusions, we have added new explanations and clarifications as noted below.

Neutrophil classification has traditionally relied on morphology or gradient separation, which while simple and robust do not capture the full repertoire of the neutrophil compartment. Some neutrophil subpopulations overlap, making nomenclature confusing and contributing to controversies regarding neutrophil function and ontogeny. In addition, the exact function of some neutrophil subpopulations and the molecular bases of heterogeneity are varied and remain elusive^{1,2}. Single-cell sequencing studies have found that neutrophils exhibit a variety of gene expression patterns depending on their developmental stage, activation status, and tissue microenvironment^{1,3}. For example, only neutrophils show tissue specificity among immune cells infiltrating the lungs, and the pulmonary microenvironment leads to their maturation and aging⁴. These findings suggest that specific micro-environmental stress may trigger neutrophil heterogeneity. The inflammatory immune microenvironment in psoriasis is both complex and variable, which provides the possibility for contributing to neutrophil heterogeneity. The limitation of our study arises from the current technical inability of single-cell RNA-sequencing to adequately capture infiltrating neutrophils in psoriatic lesions to decipher their reprogramming due to their frailty and cell death. Moreover, the mechanisms underlying neutrophil heterogeneity in pathological states, such as in trained immunity⁵, require additional multi-omics analyses and verification of the mechanism(s) in future studies.

Single-cell analyses from mouse lungs also show that exposure to low LPS dose upregulates their expression of CXCR4, and CXCR4^{hi} neutrophils release NETs to mediate allergic asthma, indicating that environmental risk factors may shape neutrophil recruitment in the lung to promote the initiation of allergic asthma⁶. CXCR4⁺ neutrophils have further been shown to exhibit over-activation, up-regulation of CD11b, enhanced secretion, phagocytosis, and NET production^{6,7}. Though we haven't observed a separate and distinct clustering marked with CXCR4 on psoriatic neutrophils using flow cytometry, we observed a significant increase of CXCR4 on neutrophils having enriched pro-inflammatory functions. Our results, therefore, highlight the importance of CXCR4^{hi} neutrophils *in vivo* and *in vitro*, in inducing vascular remodeling and permeability, facilitating infiltration of immune cells into tissues, as well as promoting a heightened inflammatory state. Moreover, our data highlight the pathogenic and potential therapeutic value of a novel neutrophil CXCR4/CXCL12 axis in inflammatory skin diseases.

To address your concerns, we have substantially revised the manuscript, revised the Result section to increase readability, and replaced words such as "distinct" and "unique" to describe this neutrophil phenotype in our revised manuscript. All changes are underlined. We hope that the reviewer will find our changes and revision adequate and satisfactory.

Reviewer #2

The authors sufficiently addressed all criticisms and have substantially improved the revised manuscript. I have no further remarks and recommend to accept the manuscript for publication.

Response: We really appreciate your suggestions and comments, which are very valuable and helpful for revising and improving our manuscript.

Reviewer #3

After reading the revised version I have still a few questions and remarks:

1. Revised Figs. 5d & 5e: Since the overlay between and green (Vimentin+CXCL12+ or CXCR4+CXCL12+, respectively) results in yellow color, yellow is not the best choice for showing expression of CD31. In fact, it is hardly possible to make out in these images if there is strong CXCL12 expression in CD31+ cells.

Response: Thank you so much for these helpful comments. To address this, we have changed the color of each marker and we also changed the presentation strategy to try to make the result more accessible to the readers, which is shown below in our second revised Figure 5d and 5e. In Figure 5d, CD31 is marked in red and vimentin is shown in green, both of which co-localize with CXCL12 (yellow), indicating that CXCL12 is derived from fibroblasts and endothelial cells in psoriatic skin. In Figure 5e, CD31 and CXCL12 are marked with red and yellow, respectively, and CXCR4 is shown in green, demonstrating that vascular endothelial cells express CXCL12, with CXCR4^{hi} cells in close proximity.

Revised Figure 5d, 5e. (d) Representative immunofluorescence staining of CD31, Vimentin, and CXCL12 in inflamed psoriatic skin (n = 6). Scale bar = 50 µm, 20 µm. (e) Representative immunofluorescence staining of CD31, CXCR4, and CXCL12 in inflamed psoriatic skin (n = 6). Scale bar = 50 µm, 20 µm.

In addition to Figure 5d, we provide additional data to demonstrate that CD31 expresses CXCL12, as shown in Figure R1. Importantly, as Supplementary Fig. 7c and 7d show, scRNA seq analysis of publicly available data from psoriatic lesions and healthy controls show that CXCL12 is mainly derived from endothelial cells, fibroblasts, and pericytes in psoriasis lesions, which is consistent with our multiple-color immunofluorescence results.

Figure R1. Representative immunofluorescence staining of CD31 (red) and CXCL12 (yellow) in inflamed psoriatic skin. Scale bar = 50 μ m, 20 μ m.

2. Can you give an explanation why 3 data sets (CXCL12-expression, cell death in CXCR4LO vs CXCR4HI cells and phagocytosis) are so fundamentally different in the revised version as compared to the original version of the manuscript?

Response: Thank you for this comment. We apologize for this not being clear. For the three data sets in the first revised version, we have addressed this comment point-by-point (see response below). The description in the Result section of the original submission is shown in *italic* to increase clarity, and related figures and method details in the two versions are shown for comparison.

1) CXCL12-expression:

In our initial submission, there was limited emphasis or focus on CXCL12 expression, with our data limited to IF co-staining on skin lesions to detect CXCL12 expression (see Figure above from the original submission, top left). In response to the comments from reviewer #3, we explored and expanded on this in our Revised manuscript via analyses of publicly available scRNA-seq data (see results from the analysis below and in our revised manuscript) and adding multi-color IF to quantify CXCL12⁺ cells in psoriatic lesions (see new panel above, top right).

In our initial (Original) submission, we stated that: *“Tissue IF showed co-localization of the endothelial marker CD31 with CXCL12 in inflamed dermis (Supplementary Fig. 1c), indicating that vascular endothelial cells were the main source of CXCL12 in inflamed psoriatic skin.”* In our revised submission, we stated that: *“We then re-analyzed publicly available scRNA seq data from psoriatic lesions and healthy controls, as previously reported. This demonstrated that CXCL12⁺ cells are increased in psoriatic lesions, among which endothelial cells (12.37%), fibroblasts (73.87%), and pericytes (7%) were the main sources of CXCL12 in psoriasis lesions (Supplementary Fig. 7c, d). Multiple-color immunofluorescence also revealed that CXCL12 immunoreactivity co-localized prominently with endothelial cells marked by CD31 and fibroblasts marked by*

vimentin (Fig. 5d) and demonstrated the colocalization of CXCL12 and CD31⁺ endothelial cells with CXCR4^{hi} neutrophils in close proximity in psoriatic lesions (Fig. 5e).”

To make the comparisons clear between the initial and revised version clear, we have a side-by-side comparison of the images from our initial (Original) submission and the first Revised submission (see figure above). Please note that there was one error in the first Revised figure that had to do with a mistake in exporting the images from the ZEN software (Zeiss), which may have made the image confusing (giving the co-localization a purple color instead of a red one). We apologize for this being unclear. Based on your comments, we not only corrected the error, but also changed the color of each marker in the second Revised submission to avoid causing ambiguity.

Method: In the original submission, immunofluorescence co-staining was performed using the following antibodies: CD31 (ab199012, Abcam) and CXCL12 (ab155090, Abcam). In the first Revised submission, multiplex fluorescence staining was employed with relatively complicated steps, as detailed in the Supplementary Method section, using the following antibodies: CD31 (ab199012, Abcam), Vimentin (ab16700, Abcam), and CXCL12 (17402-1-AP, Proteintech). We used another CXCL12 antibody in the revision due to the quick shipment, but overall, the staining pattern is the same. This information was provided in our revised submission.

New data provided for this specific aspect is the single-cell data analyses provided below (and see associated supplementary figure legend).

Revised Supplementary Fig. 7c, d. ScRNA seq analysis of publicly available data from psoriatic lesions and healthy controls demonstrated that CXCL12⁺ cells were increased in psoriatic lesions, and CXCL12 was

mainly derived from endothelial cells (12.37%), fibroblasts (73.87%), and pericytes (7%) in psoriasis lesions, which was consistent with our multiple-color immunofluorescence results.

We want to emphasize that the findings are consistent between our initial (Original) version and the first and second submitted Revised version.

2) Cell death in CXCR4^{LO} vs CXCR4^{HI} cells:

In the first Revision, as suggested by Reviewer#1 and Reviewer#2, we increased the sample size and repeated the experiments, obtained similar/same results, and added a detailed description of cell viability and apoptosis in the revised supplementary materials.

In our initial (Original) submission, we stated that: “To further validate the phenotype of CXCR4^{hi} neutrophils, neutrophil survival rate after 24 hours of culture *in vitro* was evaluated via 7-AAD staining. This showed a decreased survival rate of psoriatic CXCR4^{hi} neutrophils compared to the other two groups (Fig. 2c).”

In our first Revised submission we stated: “To further validate the phenotype of CXCR4^{hi} neutrophils, neutrophil survival rate after 24 hours (h) of culture *in vitro* was evaluated via Annexin V-7-AAD staining. This showed an increased early apoptotic rate of psoriatic CXCR4^{hi} neutrophils compared to the other two groups (Fig. 2c, Supplementary Fig. 2b).”

We apologize for the confusion, but the difference here is in regard to the different presentation of these results. In our original submission, the Y-axis showed the cell survival rate (7-AAD staining negative); while in the revised submission, the Y-axis showed the early (AnnexinV+/7-AAD-) and late apoptosis (AnnexinV+/7-AAD+). For instance, the cell survival rate of psoriatic CXCR4^{hi} neutrophils at 24 h was about 69.5% (Original); and the early apoptosis rate of psoriatic CXCR4^{hi} neutrophils at 24 h is 26.5%, late apoptosis rate is 4.42% (Revised). The survival cells are those that have not undergone apoptosis. Thus, these results are consistent between our initial submission and our revised version submitted.

Method: In the Original submission, neutrophil survival rate *in vitro* was evaluated via 7-AAD staining. In the Revised submission, cell viability and apoptosis assays were performed using the Annexin V-PE/7-AAD apoptosis kit (AP104, MultiSciences), following the manufacturer's instructions. Four cellular populations were distinguished: viable cells (Annexin V-PE and 7-AAD double-negative), early apoptotic cells (Annexin V-PE positive and 7-AAD negative), late apoptotic cells (Annexin V-PE and 7-AAD double-positive) and

necrotic cells/cellular debris (Annexin V-PE negative and 7-AAD positive), as described in the methods of our revised submission.

These findings are consistent between our initial (Original) version and the first and second submitted Revised version.

3) Phagocytosis:

To address the comment from Reviewer #3 about the phagocytic capacity of CXCR4^{hi} neutrophils, we repeated the phagocytosis experiments according to the reference⁷ provided by Reviewer #3 and bought the pHrodo Green *E coli* (2 mg/mL; P35366, Thermo Fisher Scientific) in accordance with the reference (we had previously used *Escherichia coli* DH5α bacteria labeled with Alexa 488 that was constructed in-house for our initial data). We also increased the patient samples from 3 to 6 and verified the result by immunofluorescence and flow cytometry.

In our initial (Original) submission, we stated that: “In addition, phagocytic capacity, as measured by the ability to take up FITC-labeled *Escherichia coli*, was lower in CXCR4^{hi} compared to CXCR4^{lo} neutrophils (Fig. 2f).”

In the Revised submission, we stated: “In addition, phagocytic capacity, as measured by the ability to take up FITC-labeled *Escherichia coli* (Fig. 2f, g, Supplementary Fig. 3c, d), and degranulation capacity, as measured by CD63 expression (Fig. 2h), were all higher in CXCR4^{hi} compared to CXCR4^{lo} neutrophils.” This finding is also supported by previous publication⁷.

Thus, there was a change between our initial and our first revised experiments. We suspect that the likely reason for this difference in phagocytosis is the source and concentration of *Escherichia coli*, used in these experiments. In our initial (Original) submission, *Escherichia coli* DH5α bacteria labeled with Alexa 488 was used, and bacteria/cell ratio was 2:1. In the Revised submission, pHrodo Green *E coli* (P35366, Thermo Fisher Scientific) was used at 2 mg/mL, according to previous publications suggested by Reviewer #3, and which methods we then followed^{7, 8}. We also added a detailed description of phagocytosis to our revised Method sections in the revised manuscript (Page 25).

We apologize that this was not clearly explained in detail in the response letter (NCOMMS-22-44320A) to reviewers. We are grateful that the reviewers pointed us in the right direction, and the data are now more robust with an increased sample size as well as a validated experimental approach. We want to emphasize that this change does not affect the overall findings about the role of CXCR4^{hi} neutrophils

in skin inflammation, as we mainly focus on the contribution of this neutrophil subset to vascular remodeling and inflammatory responses in sterile skin inflammation. We thank the reviewer for these thorough and constructive comments, which have enabled us to greatly improve our manuscript.

3. Typo: Line 290: CXCR12-treated group.

Response: Thank you so much for this reminder. We have corrected this mistake and carefully checked the spelling errors in our manuscript, with all changes underlined.

Reviewer #4

The authors did an exceptional job in responding to Reviewer #1's concerns. However, the reviewer has issue with the title of the manuscript, which states that CXCR4+ PMN-mediated skin inflammation occurs via glycolysis. Although the authors have shown that the CXCR4+ PMNs are associated with increased glycolytic activity and markers, nowhere in the manuscript was it shown that CXCR4+ PMN glycolysis was mechanistically relevant to skin inflammation. This was somewhat shown in an in vitro assay by blocking lactate, but not in vivo using glycolytic deficient PMNs. Therefore, the title should be modified to appropriately reflect the conclusions of the data. For example, CREB1-driven CXCR4^{hi} neutrophils orchestrate skin inflammation.

Response: We would like to express our heartfelt appreciation for the valuable and constructive comments you provided on our article. Thank you very much for your encouragement. As you suggested, we have modified the title to "CREB1-driven CXCR4^{hi} neutrophils orchestrate skin inflammation".

References

1. Xie X, et al. Single-cell transcriptome profiling reveals neutrophil heterogeneity in homeostasis and infection. *Nat Immunol* 21, 1119-1133 (2020).
2. Palomino-Segura M, Sicilia J, Ballesteros I, Hidalgo A. Strategies of neutrophil diversification. *Nat Immunol* 24, 575-584 (2023).
3. Khojraty TE, et al. Distinct transcription factor networks control neutrophil-driven inflammation. *Nat Immunol* 22, 1093-1106 (2021).
4. Ballesteros I, et al. Co-option of Neutrophil Fates by Tissue Environments. *Cell* 183, 1282-1297 e1218 (2020).
5. Kalafati L, Hatzioannou A, Hajishengallis G, Chavakis T. The role of neutrophils in trained immunity. *Immunol Rev* 314, 142-157 (2023).
6. Radermecker C, et al. Locally instructed CXCR4(hi) neutrophils trigger environment-driven allergic asthma through the release of neutrophil extracellular traps. *Nat Immunol* 20, 1444-1455 (2019).
7. Uhl B, et al. Aged neutrophils contribute to the first line of defense in the acute inflammatory response. *Blood* 128, 2327-2337 (2016).
8. Impellizzeri D, Egholm C, Valaperti A, Distler O, Boyman O. Patients with systemic sclerosis show phenotypic and functional defects in neutrophils. *Allergy* 77, 1274-1284 (2022).

REVIEWERS' COMMENTS

Reviewer #3 (Remarks to the Author):

For the former version of the immunofluorescence imaging in Figs. 5d & 5e, the authors had chosen the colors red (CXCL12), green (vimentin) and yellow (CD31), making it impossible to see those three markers in one image, because red + green = yellow.

For the new version, the authors have simply swapped two colors, now red = CD31, green = vimentin, yellow = CXCL12.

Unfortunately it is still not possible to make out on one image which cells express the majority of CXCL12, endothelial cells or fibroblasts.

I do believe the authors that CD31+ cells express CXCL12, but the choice of colors is still puzzling me.

Point-by-point response

Reviewer #3 (Remarks to the Author):

For the former version of the immunofluorescence imaging in Figs. 5d & 5e, the authors had chosen the colors red (CXCL12), green (vimentin) and yellow (CD31), making it impossible to see those three markers in one image, because red + green = yellow.

For the new version, the authors have simply swapped two colors, now red = CD31, green = vimentin, yellow = CXCL12.

Unfortunately, it is still not possible to make out on one image which cells express the majority of CXCL12, endothelial cells or fibroblasts.

I do believe the authors that CD31+ cells express CXCL12, but the choice of colors is still puzzling me.

Response: Thank you so much for this reminding. According to your suggestions, we have changed the color of each marker again to avoid confusing readers. In revised Figure 5d, CD31 is marked in yellow and vimentin is shown in green, both of which co-localize with CXCL12 (purple). In revised Figure 5e, CD31 and CXCL12 are marked with yellow and purple, respectively, and CXCR4 is shown in green.

Revised Figure 5d, 5e. (d) Representative immunofluorescence staining of CD31 (yellow), Vimentin (green), and CXCL12 (purple) in inflamed psoriatic skin (n = 6). Scale bar = 50 µm, 20 µm. (e) Representative immunofluorescence staining of CD31 (yellow), CXCR4 (green), and CXCL12 (purple) in inflamed psoriatic skin (n = 6). Scale bar = 50 µm, 20 µm.